# Federated Generalised Variational Inference: A Robust Probabilistic Federated Learning Framework

Terje Mildner [1]   Oliver Hamelijnck [1]   Paris Giampouras [1]   Theodoros Damoulas [1,2]

## Abstract

We introduce FEDGVI, a probabilistic Federated Learning (FL) framework that is robust to both prior and likelihood misspecification. FEDGVI addresses limitations in both frequentist and Bayesian FL by providing unbiased predictions under model misspecification, with calibrated uncertainty quantification. Our approach generalises previous FL approaches, specifically Partitioned Variational Inference (Ashman et al., 2022), by allowing robust and conjugate updates, decreasing computational complexity at the clients. We offer theoretical analysis in terms of fixed-point convergence, optimality of the cavity distribution, and provable robustness to likelihood misspecification. Further, we empirically demonstrate the effectiveness of FEDGVI in terms of improved robustness and predictive performance on multiple synthetic and real world classification data sets.

## 1. Introduction

Federated learning (FL) is a framework for the collaborative training of a global model by a collection of clients, without requiring proprietary data to be shared with a central server or other participating clients (McMahan et al., 2017). This decentralised approach allows FL to be used on applications with strict data privacy constraints, such as in finance or healthcare (Kairouz et al., 2021). However, due to the sensitive nature and complexity of these domains, both privacy and robustness to model misspecification are paramount.

The frequentist formulation of FL aims to minimise a global loss function by aggregating local gradients from clients. Early works include Federated Averaging (FEDAVG, McMahan et al., 2017) which iterates between training clients locally and averaging updates on the server. This has sparked a large body of research on issues such as communication efficiency, data privacy, and data heterogeneity across clients (Hamer et al., 2020; Malinovsky et al., 2020; Reddi et al., 2021; Chen et al., 2022; Tenison et al., 2023; Tziotis et al., 2023; Li et al., 2024; Demidovich et al., 2025). There has been some work addressing robustness to adversarial clients (Allouah et al., 2024; Bao et al., 2024) and data and system heterogeneity (Chen et al., 2022; Zhao et al., 2023; Heikkilä et al., 2023). However, these only provide point estimates, and do not allow principled uncertainty quantification, as required in many FL applications (Jonker et al., 2024).

In contrast, Bayesian FL approaches aim to update beliefs of a global model with data partitioned across clients. This largely builds on distributed inference methods such as the Bayesian Committee Machine (Tresp, 2000), parallel MCMC (Ahn et al., 2014; Mesquita et al., 2020), or Divide&Conquer SMC (Chan et al., 2023). Expectation Propagation (Minka, 2001; Vehtari et al., 2020) is naturally applicable to the distributed setting where local sites are iteratively refined. This requires computing the cavity distribution that removes local sites from the current approximation. Partitioned Variational Inference (PVI, Bui et al., 2018; Ashman et al., 2022) takes this idea and proposes a distributed variational inference algorithm, which has been extended through MCMC (Guo et al., 2023) and Stochastic Gradient Langevin Dynamics (SGLD) (Mekkaoui et al., 2021). Whilst these approaches quantify uncertainty, they are susceptible to model misspecification which can lead to inaccurate, overconfident predictions (Bernardo & Smith, 2000; Bissiri et al., 2016; Knoblauch et al., 2022).

Current approaches to FL are inherently non-robust to model misspecification which leads to compromised performance and uncalibrated uncertainty quantification. We address these challenges by departing from the traditional Bayesian paradigm and propose a distributed Generalised Variational Inference framework that allows us to deal with model misspecification. In summary, our contributions are:

- We introduce Federated Generalised Variational Inference (FEDGVI), a family of robust probabilistic algorithms for federated learning.

[1]University of Warwick, Department of Computer Science, Coventry, United Kingdom [2]University of Warwick, Department of Statistics, Coventry, United Kingdom. Correspondence to: Terje Mildner <Terje.Mildner@warwick.ac.uk>.

*Proceedings of the 42$^{nd}$ International Conference on Machine Learning*, Vancouver, Canada. PMLR 267, 2025. Copyright 2025 by the author(s).

- We prove that FEDGVI is robust to likelihood misspecification (Theorem 4.12).

- We demonstrate that FEDGVI generalises standard approaches such as PVI and FEDAVG (Remarks 4.1 and 4.2) and theoretically justify the use of the cavity distribution (Theorem 4.9).

- We prove that, under suitable conditions, FEDGVI converges to Generalised Bayesian posteriors (Lemma 4.6 and Proposition 4.10) that are computationally tractable.

- We evaluate FEDGVI on a range of synthetic and real-world datasets, across multiple models, demonstrating improved robustness and predictive performance.

In Section 2 we define model misspecification and recall methods that mitigate it in the non–distributed setting. Section 3 introduces our framework, which builds on these concepts and extends them to the federated setting. We analyse the theoretical properties of FEDGVI in Section 4, including provable robustness. Finally, Section 5 studies the empirical performance and gains of FEDGVI with multiple models and real world datasets such as Bayesian Neural Networks on MNIST and FASHIONMNIST. [1]

## 1.1. Related Work

**Robust Frequentist Federated Learning** In the frequentist setting, building on the seminal paper of McMahan et al. (2017), many approaches have aimed at mitigating challenges in FL, such as robustness to adversarial servers through secure aggregation (Chen et al., 2022), to stragglers (Tziotis et al., 2023), heterogenous data in out–of–distribution generalisation (Tenison et al., 2023), heterogeneous and asynchronous clients (Fraboni et al., 2023), or finding weaknesses in communications (Zhu et al., 2019; Zhao et al., 2023). More recently, work on robust server aggregations achieves robustness against Byzantine clients that aim to deteriorate model performance (Allouah et al., 2024; Bao et al., 2024). However these do not allow principled uncertainty quantification.

**Federated Bayesian Inference** Federated and distributed Bayesian methods aim to approximate the posterior as if it had been computed with the data of all clients available at a central server. Early work on distributed Bayesian inference includes Bayesian opinion pools (Genest, 1984; Carvalho et al., 2023), and the Bayesian Committee machine (Tresp, 2000), which aim to find a consensus among a collection of Bayesian beliefs. Works that aim to operationalise this in the distributed setting, where data is split IID across clients,

include Expectation Propagation (Minka, 2001; Opper & Winther, 2005; Hasenclever et al., 2017; Vehtari et al., 2020), and consensus based Monte Carlo (Scott et al., 2016). In the Federated setting this assumption is often violated, as data is not split homogeneously and IID across participating devices. From this perspective, most approaches to Bayesian FL can be categorised into finding an approximate posterior through variational inference (Corinzia et al., 2021; Ashman et al., 2022; Kassab & Simeone, 2022; Heikkilä et al., 2023; Hassan et al., 2024; Vedadi et al., 2024; Swaroop et al., 2025), Markov Chain Monte Carlo (Al-Shedivat et al., 2021; Mekkaoui et al., 2021; Kotelevskii et al., 2022; Guo et al., 2023; Hasan et al., 2024), Gaussian Processes (Achituve et al., 2021), or directly learning a Bayesian neural network (Yurochkin et al., 2019; Zhang et al., 2022). Personalised or hierarchical Bayesian FL (Kotelevskii et al., 2022; Zhang et al., 2022; Kim & Hospedales, 2023; Hassan et al., 2023; 2024; Vedadi et al., 2024) allows for additional expressibility of client posteriors, especially under heterogeneity. However, none of these are inherently robust to contamination and model misspecification.

**Robust Bayesian Inference** Although the existing Bayesian FL methods address some of the challenges of federated learning, such as communication constraints and data heterogeneity, they still aim to approximate the Bayesian posterior, which in itself is a flawed objective under model misspecification (Walker, 2013; Berk, 1966; Bernardo & Smith, 2000). In the global, non-federated case, several methods have been proposed to combat misspecification in the Bayesian setting (Grünwald, 2012), with the most promising direction being Generalised Bayesian Inference (Hooker & Vidyashankar, 2014; Bissiri et al., 2016; Ghosh & Basu, 2016a; Jewson et al., 2018; Miller, 2021; Alquier, 2021; Knoblauch et al., 2022; Matsubara et al., 2022). In this work we capitalise on this front and bring robustness to model misspecification in the federated setting.

## 2. Preliminaries

### 2.1. Notation and Model Misspecification

Let $(\Omega, \mathcal{F}, P_0)$ be a probability space where $P_0$ is the data generating process, generating the observable random variables $X_1, ..., X_n \equiv X_1^n$ taking values in the measurable space $(\Xi, \mathcal{X})$. Further, let $Y_1^n$ be observable random variables depending on $X_1^n$ respectively, taking values in $(\Upsilon, \mathcal{Y})$. Denote their realisations $\{X_i = x_i, Y_i = y_i\}_{i=1}^n$, which are assumed to be partitioned across $M$ clients $\{\mathbf{x}_m, \mathbf{y}_m\}_{m=1}^M$ each of size $n_m$. Consider hypothesis measures $P_{\boldsymbol{\theta}}$ where $\boldsymbol{\theta}$ takes values in $(\Theta, \mathcal{T})$, a measurable space, admitting densities $p_{\boldsymbol{\theta}}$. We study elements of $\mathcal{P}(\Theta)$, the set of all probability measures on $(\Theta, \mathcal{T})$, starting with prior $\Pi$ and updated to $Q$, dominated by some common measure $\mu$, and

---

[1]Code to reproduce experiments can be found at https://github.com/Terje-M/FedGVI.

---

**Algorithm 1** FEDGVI SERVER

---

1: **Input:** $\pi(\boldsymbol{\theta})$, $\mathcal{Q}$, $D_s$
2: **Define:** $\ell_m^{(0)}(\boldsymbol{\theta}) = 0$, $\ell_s^{(0)}(\boldsymbol{\theta}) = 0$, $q_s^{(0)}(\boldsymbol{\theta}) = \pi(\boldsymbol{\theta})$
3: **for** $t = 1, ..., T$ **do**
4:     **for** $m = 1, ..., M$ in parallel **do**
5:        $\Delta_m^{(t)}(\boldsymbol{\theta}) \leftarrow$ CLIENT$(q_s^{(t-1)}(\boldsymbol{\theta}), \mathcal{Q}, m)$
6:     **end for**
7:     Set $\ell_s^{(t)}(\boldsymbol{\theta}) \leftarrow \ell_s^{(t-1)}(\boldsymbol{\theta}) + \sum_{m=1}^{M} \Delta_m^{(t)}(\boldsymbol{\theta})$
8:     Optimise $q_s^{(t)}(\boldsymbol{\theta})$ according to Equation (7)
9: **end for**

---

**Algorithm 2** FEDGVI CLIENT

---

1: **Input:** $q_s^{(t-1)}(\boldsymbol{\theta})$, $\mathcal{Q}$, $\{\mathbf{x}_m, \mathbf{y}_m\}$, $L_m$, $\ell_m^{(t-1)}(\boldsymbol{\theta})$, $D$
2: Optimise $q^{\backslash m}(\boldsymbol{\theta})$ according to Equation (3)
3: Optimise $q_m^{(t)}(\boldsymbol{\theta})$ according to Equation (4)
4: Set $\Delta_m^{(t)}(\boldsymbol{\theta})$ according to Equation (5)
5: Set $\ell_m^{(t)}(\boldsymbol{\theta}) \leftarrow \ell_m^{(t-1)}(\boldsymbol{\theta}) + \Delta_m^{(t)}(\boldsymbol{\theta})$
6: **return:** Communicate $\Delta_m^{(t)}(\boldsymbol{\theta})$ to SERVER

---

admitting densities $\pi$ and $q$ respectively. Naive Bayes updates $\pi(\boldsymbol{\theta})$ to $q_B(\boldsymbol{\theta})$ through

$$q_B(\boldsymbol{\theta}) = \pi(\boldsymbol{\theta}) \prod_{m=1}^{M} p_{\boldsymbol{\theta}}(\mathbf{y}_m; \mathbf{x}_m) / Z \quad (1)$$

where $Z = \int_\Theta \prod_{m=1}^{M} p_{\boldsymbol{\theta}}(\mathbf{y}_m; \mathbf{x}_m) \Pi(d\boldsymbol{\theta})$ is the marginal likelihood. Since we do not suppose that the prior $\Pi$, nor the likelihood $P_{\boldsymbol{\theta}}$ are well specified, i.e. $P_0 \notin \mathcal{P}(\Theta)$, we are in the $\mathcal{M}$–open setting (Bernardo & Smith, 2000), the model misspecified, and the Bayesian posterior inappropriate.

### 2.2. Model Misspecification

There are several different ways we can think about model misspecification under the $\mathcal{M}$–open assumption.

**Prior Misspecification** The traditional Bayesian paradigm assumes that the prior encodes the best available judgement about $\boldsymbol{\theta}$, which beyond simple settings, is never realised (Berger, 1985; Knoblauch et al., 2018). Such misspecification is common; e.g. it is standard to use zero–mean Gaussian distributions on the weights of Bayesian Neural networks. This can have dire effects, for instance Diaconis & Freedman (1986) demonstrate that multimodal priors in a location model can cause the posterior to not accumulate around $P_0$, even when the DGP is well specified, i.e. when $P_0 \in \mathcal{P}(\Theta)$.

**Likelihood Misspecification** One such example is where the hypothesis of interest is contaminated , and an $\varepsilon$ fraction of the data (input and/or output variables) has some unknown data source. Formalising this we follow the definition of Huber (1964):

**Definition 2.1** (Huber contamination). *Given an $\varepsilon \in (0, \frac{1}{2})$ and the uncontaminated distribution $P_{\boldsymbol{\theta}}$ of inliers and some contaminating distribution $G$ of outliers, then $P_0$ is said to be an $\varepsilon$-corrupted version of $P_{\boldsymbol{\theta}}$; $P_0 := (1 - \varepsilon)P_{\boldsymbol{\theta}} + \varepsilon G$.*

### 2.3. Robust Bayesian Methods

**Generalised Bayesian Inference (GBI)** Instead of linking the parameter and data through likelihoods, Bissiri et al.

(2016) and Miller (2021) formalised a coherent Bayesian framework using loss functions leading to Gibbs posteriors (Alquier et al., 2016). This was further utilised to deal with likelihood misspecification through robust losses, e.g Knoblauch et al. (2018). Let $L : \Theta \times \Xi \times \Upsilon \to \mathbb{R}$ be such a loss, then the GBI posterior is given by:

$$q_{\text{GBI}}(\boldsymbol{\theta}) = \pi(\boldsymbol{\theta}) \exp\left\{-\beta \sum_{m=1}^{M} L(\mathbf{y}_m; \boldsymbol{\theta}, \mathbf{x}_m)\right\} / Z \quad (2)$$

with $Z = \int_\Theta \exp\{-\beta \sum_{m=1}^{M} L(\mathbf{y}_m; \boldsymbol{\theta}, \mathbf{x}_m)\} \Pi(d\boldsymbol{\theta})$. Here, $\beta \in \mathbb{R}_{>0}$ is a learning rate parameter that determines how much weight we place on the observed data, similar to power posteriors in VI (Grünwald, 2012; Kallioinen et al., 2024). This recovers $q_B(\boldsymbol{\theta})$ when the loss is the negative log–likelihood and $\beta = 1$.

**Generalised Variational Inference (GVI)** In Knoblauch et al. (2022) GBI is generalised within a variational framework that explicitly accounts for prior and likelihood misspecification. Let $D : \mathcal{P}(\Theta) \times \mathcal{P}(\Theta) \to \mathbb{R}_+$ be a divergence then the GVI posteriors are defined as:

$$q_{\text{GVI}}(\boldsymbol{\theta}) = \arg\min_{q \in \mathcal{Q}} \left\{ \mathbb{E}_{q(\boldsymbol{\theta})} \left[ L(\mathbf{y}_1^M; \boldsymbol{\theta}, \mathbf{x}_1^M) \right] + D(q : \pi) \right\}$$

where $\mathcal{Q} \subset \mathcal{P}(\Theta)$, making inference tractable. This allows for targeting a larger subspace of posteriors, and through different divergences the effect of the prior can be controlled.

## 3. Federated Generalised Variational Inference

### 3.1. Methodology

In this section, we present the proposed federated learning framework, named FEDGVI, that explicitly addresses likelihood and prior misspecification. We aim to learn a robust approximate posterior $q_s(\boldsymbol{\theta})$ using partitioned observations across M clients. FEDGVI iterates consist of two steps: a) sending of the current approximate posterior to each client, which is updated through a robust variational objective, and b) aggregating the updates on the server, resulting in a robust approximate posterior; summarised in Algorithms 1 and 2.

**Initialisation** We set the initial server posterior as the prior, $q_s^{(0)}(\boldsymbol{\theta}) = \pi(\boldsymbol{\theta})$, and the local and server loss ap-

proximations to be zero, $\ell_m^{(0)}(\boldsymbol{\theta}) = 0$ and $\ell_s^{(0)}(\boldsymbol{\theta}) = 0$ respectively; $m$ denotes a specific client and $s$ the server.

**Until Convergence** For $t = 1, 2, ..., T$, we synchronously compute updates locally at each client, and accumulate these at the server to form the new global posterior $q_s^{(t)}(\boldsymbol{\theta})$.

**Client** The client receives the current approximate posterior from the server. This will be used as the prior from which a client can compute an updated posterior using their local data. First, however the information of the client's data must be removed by computing the cavity distribution. The cavity distribution acts as the local prior incorporating all previous information from all other clients and is given by:

$$q^{\backslash m}(\boldsymbol{\theta}) \propto \frac{q_s^{(t-1)}(\boldsymbol{\theta})}{\exp\{-\ell_m^{(t-1)}(\boldsymbol{\theta})\}} \qquad (3)$$

The client then computes a robust local approximate posterior with it's local data set $\{\mathbf{x}_m, \mathbf{y}_m\}$ and it's loss function $L_m^{(t)}(\cdot)$, which is regularised by the divergence, $D$, and cavity distribution

$$q_m^{(t)}(\boldsymbol{\theta}) = \underset{q \in \mathcal{Q}}{\arg\min} \, \mathbb{E}_{q(\boldsymbol{\theta})}\left[ L_m^{(t)}(\mathbf{y}_m; \boldsymbol{\theta}, \mathbf{x}_m) \right] + D(q : q^{\backslash m}). \qquad (4)$$

This GVI style objective allows the client to be robust to both likelihood misspecification as well as prior misspecification arising due to the cavity. To update the global posterior at the server, the client computes the negative log ratio of the local and global posteriors. In line with existing Bayesian FL (Ashman et al., 2022; Guo et al., 2023), we use a damping parameter $\tau_m \in (0, 1]$, which is analogous to a learning rate as in frequentist FL, to compute the update:

$$\Delta_m^{(t)}(\boldsymbol{\theta}) = -\tau_m \log \frac{q_m^{(t)}(\boldsymbol{\theta})}{q_s^{(t-1)}(\boldsymbol{\theta})} \qquad (5)$$

The client stores $\ell_m^{(t)}(\boldsymbol{\theta}) := \ell_m^{(t-1)}(\boldsymbol{\theta}) + \Delta_m^{(t)}(\boldsymbol{\theta})$ and communicates $\Delta_m^{(t)}(\boldsymbol{\theta})$ to the server.

**Server** The loss at the server is updated based on the received client updates,

$$\ell_s^{(t)}(\boldsymbol{\theta}) = \ell_s^{(t-1)}(\boldsymbol{\theta}) + \sum_{m=1}^{M} \Delta_m^{(t)}(\boldsymbol{\theta}) \qquad (6)$$

By only incorporating clients' updates that have changed we can trivially allow for batched and asynchronous scheduling of clients. The updated loss is then used to compute the new server posterior though a GVI optimisation procedure:

$$q_s^{(t)}(\boldsymbol{\theta}) = \underset{q \in \mathcal{Q}}{\arg\min} \, \mathbb{E}_{q(\boldsymbol{\theta})}\left[ \ell_s^{(t)}(\boldsymbol{\theta}) \right] + D_s(q : \pi) \qquad (7)$$

This posterior and loss are passed back to the clients for further refinement at the next iteration until convergence.

### 3.1.1. HYPERPARAMETERS

Ashman et al. (2022) set the damping parameter to $\tau \propto \frac{1}{M}$ throughout their experiments. This turns out, see Proposition 4.3, to be a reasonable choice when $\tau = \frac{1}{M}$ in combination with $D_s = D_{KL}$ since this causes the posterior at the server to be a logarithmic opinion pool induced by an externally Bayesian pooling operator (Genest et al., 1986), ensuring stable convergence. Other hyperparameters arising from the choice of losses and divergences are dependent on the expected amount of model misspecification.

### 3.2. Robustness to Likelihood Misspecification

Within our framework we are free to choose the client side losses. We consider the Density–Power divergence based loss (Ghosh & Basu, 2016b), often referred to as $\beta$–divergence loss $\mathcal{L}_\beta$, the $\gamma$–divergence based losses (Hung et al., 2018), $\mathcal{L}_\gamma$, as well as a score matching loss, $\mathcal{L}_{SM}$, based on the Hyvärinen divergence (Hyvärinen, 2005; Altamirano et al., 2023). In the classification setting, we consider the generalised cross–entropy loss

$$\mathcal{L}_{GCE}^{(\delta)}(y_i; \boldsymbol{\theta}, x_i) = \frac{(1 - p_{\boldsymbol{\theta}}(y = y_i; x_i)^\delta)}{\delta} \qquad (8)$$

for some $\delta \in (0, 1]$ (Zhang & Sabuncu, 2018). These losses are robust to misspecification because they have a finite supremum (see Definition 4.11). It is important to highlight that GVI and FEDGVI may underperform when using robust losses in the case of correct likelihood specification; see Knoblauch et al. (2022). We can use a Sequential Monte Carlo sampler to estimate the $\beta$ or $\gamma$ hyperparameters in $\mathcal{L}_\beta$ and $\mathcal{L}_\gamma$ (Yonekura & Sugasawa, 2023) or use cross validation to select optimal parameters (Altamirano et al., 2024).

### 3.3. Robustness to Prior Misspecification

We mainly consider the weighted Kullback–Leiber divergence, $\frac{1}{w}D_{KL}$, (Kullback & Leibler, 1951)

$$\frac{1}{w}D_{KL}(q : \pi) := \frac{1}{w} \mathbb{E}_{q(\boldsymbol{\theta})}\left[ \log \frac{q(\boldsymbol{\theta})}{\pi(\boldsymbol{\theta})} \right],$$

and the Alpha–Rényi divergence, $D_{AR}^{(\alpha)}$,

$$D_{AR}^{(\alpha)}(q : \pi) := \frac{1}{\alpha(\alpha - 1)} \log \left( \mathbb{E}_{\pi(\boldsymbol{\theta})}\left[ \left( \frac{q(\boldsymbol{\theta})}{\pi(\boldsymbol{\theta})} \right)^\alpha \right] \right).$$

As examined in Knoblauch et al. (2022), $D_{AR}^{(\alpha)}$ allows for different prior regularisation depending on how much we trust the prior by placing different weights on it. In future work it would be simple to explore other divergences such as the $f$–divergences, $D_f$, (Amari, 2016; Alquier, 2021). Similarly to the losses, we can perform cross validation to select the $\alpha$ parameter, however as demonstrated in the ablation study (Figure 6) FedGVI performs favourably under a range of $\alpha$ (and $\delta$) values.

# 4. Theoretical Results

We now present a theoretical analysis of FEDGVI. We begin by examining the relationship of FEDGVI with other FL algorithms while recovering some of them as special cases, we study the damping parameter, and examine the convergence behaviour of FEDGVI. Then, we turn our attention on robustness to likelihood misspecification, where we first study FEDGVI as distributed GBI, from which we derive a theorem on the necessity of the cavity distribution. Finally, we derive a result for computationally tractable and conjugate FEDGVI, enabling us to present the main theorem on bias–robustness of FEDGVI.

Since it is an open problem where global GVI posteriors converge to under arbitrary divergences, we often have to restrict ourselves to consider the server divergence to be the Kullback–Leibler divergence. This ensures that the posterior at the server will have the structure of a GBI posterior,

$$q_s^{(T)}(\boldsymbol{\theta}) \propto \exp\left\{-\sum_{m=1}^M \ell_m^{(T)}(\boldsymbol{\theta})\right\} \pi(\boldsymbol{\theta})$$

where we incorporate prior robustness and tractability through the approximate losses.

## 4.1. Recovering Existing Methods as a Special Case

By choosing specific divergences, loss functions, and variational families, we can recover existing methods as special cases of our framework, which we summarise in Figure 1:

*Remark* 4.1. Choosing the Kullback–Leibler divergence and the negative log–likelihood as a loss function recovers the PVI algorithm of Ashman et al. (2022).

*Remark* 4.2. When $D = D_s = 0$, and $\mathcal{Q} = \{\delta_{\hat{\boldsymbol{\theta}}}(\boldsymbol{\theta}) : \hat{\boldsymbol{\theta}} \in \Theta\}$, with $\delta_{\hat{\boldsymbol{\theta}}}$ being the Dirac–delta measure at some element $\hat{\boldsymbol{\theta}}$, we recover FEDAVG of McMahan et al. (2017).

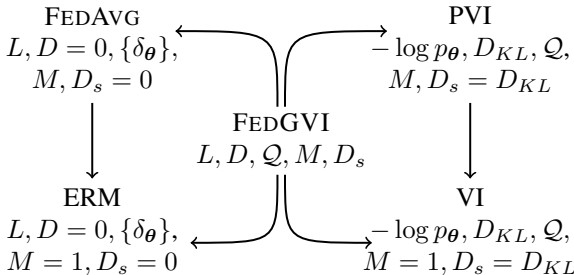

Figure 1: We illustrate the relationship of FEDGVI—characterised by the loss $L$, the client divergence $D$, the variational family $\mathcal{Q}$, the number of clients $M$, and the divergence at the server $D_s$—to Partitioned Variational Inference (PVI), Variational Inference (VI), Federated Averaging (FEDAVG), and Empirical Risk Minimisation (ERM).

## 4.2. Damping as a Bayesian Logarithmic Opinion Pool

Choosing the damping parameter to be $\tau = 1/M$ results in a logarithmic opinion pool. In fact choosing damping parameters such that all of them sum to unity also forms a valid logarithmic opinion pool (Genest et al., 1986).

**Proposition 4.3.** *Assume $D_s = D_{KL}$, and that $\sum_m \tau_m = 1$ where $\tau_m \geq 0 \,\forall m$, then the posterior at the server is an externally Bayesian logarithmic opinion pool of the form*

$$q_s^{(t)}(\boldsymbol{\theta}) = \frac{\prod_{m=1}^M \big(q_m^{(t)}(\boldsymbol{\theta})\big)^{\tau_m}}{\int_\Theta \prod_{m=1}^M \big(q_m^{(t)}(\boldsymbol{\theta})\big)^{\tau_m} d\boldsymbol{\theta}}, \;\; \boldsymbol{\theta} - a.e.$$

See Appendix B.2 for the proof. This results provides a theoretical justification on the previously heuristic use of the damping parameter (as used in PVI, Ashman et al., 2022). Specifically it ensures that this selection of $\tau$ leads to a valid distribution and results in more stable convergence.

## 4.3. Fixed Points of FEDGVI

In this section we study the properties of FEDGVI posteriors when these converge to some fixed point. Specifically, we generalise the fixed point result of PVI (Ashman et al., 2022, Property 2.3) to arbitrary losses.

**Proposition 4.4.** *Let $D_s = D_{KL}$, $D = \frac{1}{w}D_{KL}$, $w > 0$, and $\mathcal{Q} \subset \mathcal{P}(\Theta)$, then if $q_s^*(\boldsymbol{\theta}) = \pi(\boldsymbol{\theta}) \exp\{-\ell_s^*(\boldsymbol{\theta})\}/Z_{q^*}$ such that $\forall m \in [M]$, $\Delta_m^*(\boldsymbol{\theta}) = 0$, then $q_s^*(\boldsymbol{\theta})$ is a local minimiser of the following GVI objective:*

$$\mathbb{E}_{q(\boldsymbol{\theta})}\left[\sum_{m=1}^M L_m(\mathbf{y}_m; \boldsymbol{\theta}, \mathbf{x}_m)\right] + \frac{1}{w}D_{KL}(q : \pi) \quad (9)$$

*Remark* 4.5. If the loss in Equation (9) is convex, then a fixed point of FEDGVI is a global minimum of GVI.

This illustrates that if FEDGVI converges, then the posterior is a (local) minimiser of the GVI objective. We refer to such distributions as fixed points. This recovers Kassab & Simeone (2022, Theorem 1) (which deals with the restricted case of $\mathcal{Q} = \mathcal{P}(\Theta)$) with a novel proof; see Appendix B.3.

## 4.4. Generalised Bayesian Inference

As a consequence of Proposition 4.4 and Remark 4.1, FEDGVI will recover the GBI posterior when $\mathcal{Q} = \mathcal{P}(\Theta)$.

**Lemma 4.6.** *Assuming $\mathcal{Q} = \mathcal{P}(\Theta)$, $D = \frac{1}{\beta}D_{KL}$ with $\beta > 0$, $D_s = D_{KL}$, and $\tau = 1$, then FEDGVI will recover the GBI posterior after the first iteration.*

$$\begin{aligned} q_s^{(1)}(\boldsymbol{\theta}) &= q_{GBI}(\boldsymbol{\theta}|\{\mathbf{x}_m, \mathbf{y}_m\}_{m=1}^M) \\ &= \exp\{-\beta\sum_{m=1}^M L(\mathbf{y}_m; \boldsymbol{\theta}, \mathbf{x}_m)\}\pi(\boldsymbol{\theta})/Z \end{aligned}$$

*This posterior is invariant under subsequent iterations of* FEDGVI*, having reached a fixed point.*

*Moreover, for a damping rate $\tau = 1/M$, the posterior at the server converges pointwise a.e. in $\Theta$ to the GBI posterior,*

$$q_s^{(T)}(\boldsymbol{\theta}) \overset{T \to \infty}{\longrightarrow} q_{GBI}(\boldsymbol{\theta}|\{\mathbf{x}_m, \mathbf{y}_m\}_{m=1}^M), \; \boldsymbol{\theta} - a.e.$$

This result, proven in Appendix B.4, is the first step towards likelihood robustness. If we were able to find the GBI posterior efficiently with some robust loss, then the posterior would be robust and computable. Here however, the loss may not vary over different iterations of FEDGVI as in Equation (4) and the normaliser may be intractable.

### 4.5. The Cavity Distribution is Necessary

By further investigating the relationship of FEDGVI with the GBI posterior, we can extend Lemma 4.6 and derive a Theorem under which we are required to use the cavity distribution to regularise the client update. This is in contrast to both PVI, where it's use is heuristically justified, and to other Bayesian FL approaches where the previous posterior is used instead. For this we recall two natural assumptions that any such distribution must satisfy in a federated setting.

**Assumption 4.7.** No client can have access to the data set of another client.

**Assumption 4.8.** Each client generates their update equivalently to other clients.

These assumptions combined with Lemma 4.6 lead us to the necessity of the cavity distribution.

**Theorem 4.9.** *Let the assumptions be as in Lemma 4.6 with $\tau = 1$, and assume that the Assumptions 4.7 and 4.8 are satisfied, then $(1.)$ holds if and only if $(2.)$ holds.*

1. FEDGVI *recovers the generalised Bayesian posterior $q_{\mathrm{GBI}}(\boldsymbol{\theta})$ which is invariant under further FEDGVI updates.*

2. *The cavity regularises the client optimisation problem.*

This provides a principled justification for the use of the cavity distribution, as defined in Equation (3), in FEDGVI. We provide the proof in Appendix B.5.

### 4.6. Conjugate Client Updates

Before we present our main result on provable robustness to likelihood misspecification, we first show that we can find a GBI posterior under specific losses in a computationally tractable manner. Assuming that the data generating process has some exponential family distribution, where $\mathbf{y} \sim p_{\boldsymbol{\theta}}(\mathbf{y})$,

$$p_{\boldsymbol{\theta}}(\mathbf{y}) = \exp\{\boldsymbol{\eta}(\boldsymbol{\theta})^\top \boldsymbol{\phi}(\mathbf{y}) - A(\boldsymbol{\eta}(\boldsymbol{\theta})) + h(\mathbf{y})\},$$

such that this is differentiable in $\mathbf{y}$, by using the weighted score matching loss of Altamirano et al. (2023), $\mathcal{L}_{SM}^w$, then client updates, using the weighted KL divergence locally, are available in closed form. If we further assume that our

model is Gaussian, or has the form of a squared exponential, and that the natural parameters of the DGP are $\boldsymbol{\eta}(\boldsymbol{\theta}) = \boldsymbol{\theta}$, then the client approximation will have a conjugate form.

**Proposition 4.10.** *Assume that the hypothesis $p_{\boldsymbol{\theta}}(\mathbf{y})$ has differentiable, exponential family distribution with $\boldsymbol{\eta}(\boldsymbol{\theta}) = \boldsymbol{\theta}$, $L_m^{(t)} = \mathcal{L}_{SM}^{w_m^t}$, and $D = \frac{1}{\beta} D_{KL}$, and the variational family $\mathcal{Q}$ is the multivariate Gaussians, then the local posteriors at the clients are conjugate Gaussians. Moreover, Equation (7) will have closed form if $D_s$ has closed form between Gaussian distributions.*

See Appendix B.6 for the proof. The loss may now depend on the client and iteration $t$. Most exponential family distributions satisfy the conditions of the proposition, and there are several divergences that allow closed form expressions between Gaussians, such as the Alpha–Rényi, or the $\alpha, \beta, \gamma$–divergences of Cichocki & Amari (2010). Further, this enables the use of intractable likelihood models.

### 4.7. Provable Robustness to Outliers

For a robust loss function at the clients, and using the weighted KL divergence at the clients and the KL divergence at the server, guarantees that after $T$ iterations, the posterior computed at the server will also be robust to outliers. This means we can achieve robustness at the server by leveraging the robust losses that were derived for GVI. In this, we mean robustness as defined by Ghosh & Basu (2016a) and further developed in Matsubara et al. (2022). We define the empirical DGP of a client as $\mathbb{P}_{n_m} := \frac{1}{n_m} \sum_{i=1}^{n_m} \delta_{\mathbf{x}_i}$, and of the entire data set as $\mathbb{P}_n := \frac{1}{n} \sum_{m=1}^M n_m \mathbb{P}_{n_m}$. When this is contaminated by some $\varepsilon$ fraction of data centred at some adversarially chosen data point $z \in \Xi$, the misspecified DGP is defined as $\mathbb{P}_{n,\varepsilon,z} := (1-\varepsilon)\mathbb{P}_n + \varepsilon \delta_z$.

**Definition 4.11.** We say that a loss $L_m^{(t)}(\boldsymbol{\theta}; \mathbb{P}_{n_m,\varepsilon,z})$, w.r.t. some prior distribution $\pi(\boldsymbol{\theta})$, is robust to outliers, if the following hold:

1. $\displaystyle \sup_{z \in \Xi} \left| \frac{d}{d\varepsilon} L_m^{(t)}(\boldsymbol{\theta}; \mathbb{P}_{n_m,\varepsilon,z})\big|_{\varepsilon=0} \right| \leq \gamma_{(m)}^{(t)}(\boldsymbol{\theta}),$

2. $\displaystyle \sup_{\boldsymbol{\theta} \in \Theta} \pi(\boldsymbol{\theta})\gamma_{(m)}^{(t)}(\boldsymbol{\theta}) < \infty, \text{ and}$

3. $\displaystyle \int_\Theta \pi(\boldsymbol{\theta})\gamma_{(m)}^{(t)}(\boldsymbol{\theta})\mu(d\boldsymbol{\theta}) < \infty$

These conditions ensure that the influence of arbitrary contamination on the local posterior is not arbitrarily bad. In particular the auxiliary function $\gamma_m^{(t)}$ ensures that the influence of an adversarial data point $z$ on the posterior over infinitesimal contaminations, $\frac{d}{d\epsilon}q_m^{(t)}(\theta; \mathbb{P}_{n_m,\epsilon,z})|_{\epsilon=0}$, are finite over all $\theta$ and $z$. Condition 2 ensures the loss increases slowly enough for the local posterior to concentrate around the data, and condition 3 ensures the resulting posterior will be normalisable.

**Theorem 4.12.** *Let $D_s = D_{KL}$, $D = \frac{1}{w}D_{KL}$, $\mathcal{Q} = \mathcal{P}(\Theta)$, further assume that the prior is upper bounded and the loss is lower bounded, then if $\forall t \in [T]$ and $\forall m \in [M]$ $L_m^{(t)}(\boldsymbol{\theta}; \mathbb{P}_{n_m,\varepsilon,z})$ is robust, then the posterior generated by* FEDGVI *will be robust to outliers.*

The proof is in Appendix B.7. This result together with Proposition 4.10 is significant as we have robustness under intractable optimisation, *and* we can choose a provably robust, conjugate loss to generate robust FEDGVI posteriors, which are then computationally efficient to compute.

## 5. Experiments

We evaluate FEDGVI against several other methods, specifically PVI (Ashman et al., 2022), FEDAVG (McMahan et al., 2017), the nonparametric DSVGD (Kassab & Simeone, 2022), the distributed MCMC based DSGLD (Ahn et al., 2014), federated MCMC based FEDPA (Al-Shedivat et al., 2021), and the one shot BCM based approach $\beta$–PREDBAYES (Hasan et al., 2024). We provide further details about experiments in Appendix D.

### 5.1. 1D Clutter Problem

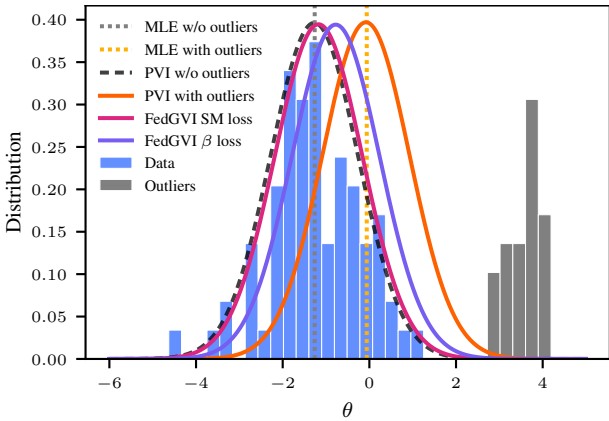

Figure 2: Robustness to outliers can be achieved through varying losses with FEDGVI, while traditional Bayesian methods fail.

We first examine the effect of misspecified likelihoods through the well known clutter problem (Minka, 2001). We generate 100 observations from a Gaussian location model that is contaminated through Definition 2.1 with $\varepsilon = 0.25$ Gaussian noise. The aim is to infer the location parameter $\boldsymbol{\theta}$ of the uncontaminated data. We compare FEDGVI with both $\mathcal{L}_\beta$ and $\mathcal{L}_{SM}$ vs PVI with and without misspecification. We also provide the corresponding MLE results. See Figure 2. Under misspecification both the MLE and PVI fail to recover the true $\boldsymbol{\theta}$, whereas FEDGVI can easily handle different levels of contamination.

### 5.2. Influence Function

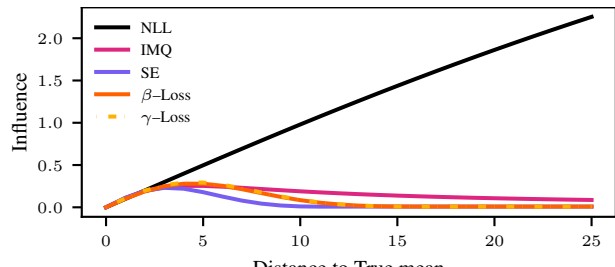

Figure 3: We plot the influence of a single outlier on the server posterior. PVI is not robust to likelihood misspecification through outliers, because it uses the negative log–likelihood (NLL).

To demonstrate robustness to likelihood misspecification as in Theorem 4.12, we consider the influence of a single outlier at one of seven clients on the server posterior. Figure 3 demonstrates that the negative log likelihood is not robust in the federated setting, whereas different robust divergence based losses allow only limited influence of outliers on the posterior. We plot this as the divergence between the posterior, had we observed the outlier value at the true mean, against the posteriors that have the outlier be farther from the true mean, using the Fisher–Rao distance (Nielsen, 2023).

### 5.3. 2D Misspecified Logistic Regression

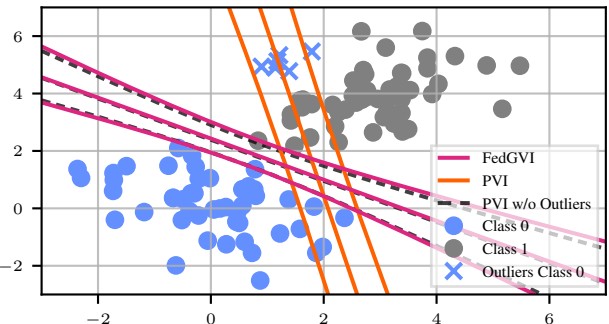

Figure 4: Logistic Regression decision boundaries (0.2, 0.5, 0.8) for PVI without outliers, PVI with misspecification, and FEDGVI with misspecification. The synthetic data set is split homogeneously across 5 clients where PVI negatively skews the decision boundary, while FEDGVI does not.

We next consider a 2D logistic regression example where we generate 100 linearly separable samples from a Gaussian mixture distribution. We inject outliers generated by a third Gaussian distribution and assign them to one of the classes so that the data is no longer linearly separable. We compare FEDGVI with $\mathcal{L}_\beta^{(0.7)}$ and $D_{AR}^{(1.5)}$ against PVI, both with 5 clients. Again, the target is given by PVI only trained on the uncontaminated data. As expected PVI is severely impacted

by outliers, whereas FEDGVI is robust to them and closely recovers the target posterior.

## 5.4. Real-World Cover Type Dataset

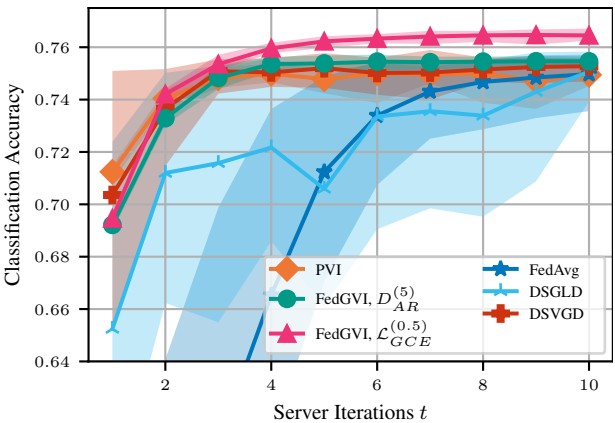

Figure 5: Results on the COVERTYPE data set. We place a Gaussian distribution over the weights and average over 10 different train/test splits; see Appendix D for details.

In this experiment we follow the experimental setup of Kassab & Simeone (2022) and average accuracy over 10 random 80/20 train-test splits, where the training data is split homogeneously across 2 clients. We do not add any label contamination. The results are plotted in Figure 5. The non-robust methods all eventually achieve similar accuracy, however FEDGVI is able to outperform all competing methods, which we argue is due to FEDGVI putting less weight on data points that are less likely to belong to the class.

## 5.5. Bayesian Neural Networks on MNIST and FASHIONMNIST

Table 1: Classification accuracy (highest in bold) on uncontaminated test data after training on 10% contaminated MNIST data. We report the best performance across all server iterations.

| MODEL | ACCURACY + STD. | |
| --- | --- | --- |
| | 10 CLIENTS | 3 CLIENTS |
| FEDAVG | 96.64± 0.07 | 96.34 ± 0.20 |
| FEDPA | 94.25± 0.39 | 95.31± 0.35 |
| $\beta$–PREDBAYES | 94.90± 0.08 | 96.73± 0.08 |
| PVI | 95.56± 0.18 | 96.68± 0.07 |
| FEDGVI $D_{AR}$ | 96.36± 0.09 | 97.13 ± 0.13 |
| FEDGVI $L_{GCE}$ | 97.06± 0.03 | 98.04 ± 0.07 |
| FEDGVI $D_{AR}+L_{GCE}$ | **97.50± 0.07** | **98.13± 0.08** |
| VI (1 CLIENT) | (96.96± 0.17) | |
| GVI (1 CLIENT) | (**98.13± 0.07**) | |

We create label contamination by adding noise to the train-

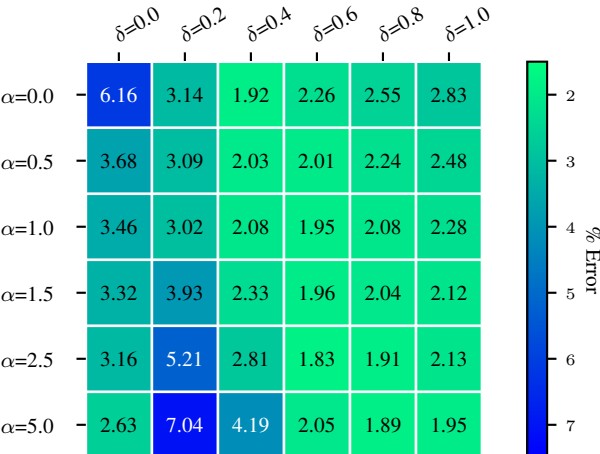

Figure 6: An ablation study on the hyperparameters of FEDGVI with $\mathcal{L}_{GCE}^{(\delta)}$ and $D_{AR}^{(\alpha)}$. We plot the maximum results achieved as percentage errors on uncontaminated test data after training 5 clients on 10% contaminated data.

Table 2: Classification accuracy (highest in bold) on uncontaminated test data after training on different amounts of contaminated FASHIONMNIST data. For FEDGVI we have fixed $\alpha = 2.5$ for the $\alpha-$Rényi divergence. Each Method has data split homogeneously across 3 Clients. We report the best performance during all server iterations.

| MODEL | CONTAMINATION | | | |
| --- | --- | --- | --- | --- |
| | 0% | 10% | 20% | 40% |
| FEDAVG | 85.7±0.5 | 79.0±1.9 | 71.2±1.5 | 49.0±6.5 |
| FEDPA | 88.1±0.3 | 87.4±0.2 | 86.5±0.2 | 85.4±0.5 |
| $\beta$–PREDBAYES | 87.6±0.1 | 87.2±0.1 | 86.8±0.1 | 85.8±0.1 |
| PVI | 86.2±0.2 | 85.1±0.1 | 84.4±0.1 | 82.8±0.1 |
| FEDGVI $\delta = 0.0$ | 87.1±0.1 | 86.2±0.2 | 85.6±0.1 | 83.8±0.1 |
| FEDGVI $\delta = 0.4$ | 88.7±0.2 | **88.6±0.1** | 87.0±0.4 | 78.1±0.4 |
| FEDGVI $\delta = 0.5$ | **89.0±0.2** | 88.6±0.2 | **88.4±0.2** | 85.1±0.7 |
| FEDGVI $\delta = 0.8$ | 88.6±0.0 | 88.4±0.1 | 88.0±0.0 | **87.2±0.1** |
| FEDGVI $\delta = 1.0$ | 88.1±0.1 | 87.8±0.1 | 87.5±0.2 | 86.0±0.3 |

ing set while leaving the test set unchanged and evaluate performance in this. For MNIST, we add 10% of class dependent label noise, see Figure 7 and Table 1. We further carry out an ablation study on the hyperparameter selection in FEDGVI with the Alpha–Rényi divergence and the generalised cross entropy loss, see Figure 6. This demonstrates that FEDGVI performs well under a variety of different loss and divergence parameters. Note that $\alpha = 1$ recovers the KL divergence, $\alpha = 0$ the reverse KL divergence, i.e. $D_{AR}^{(0)}(q : \pi) = D_{RKL}(q : \pi) = D_{KL}(\pi : q)$, and that $\delta = 0$ recovers the negative log–likelihood.

For FASHIONMNIST, in Table 2, we vary the amount of random label contamination, showcasing performance drops under different amounts of misspecification. We use an MLP, for FEDGVI and PVI with 1 hidden layer of 200

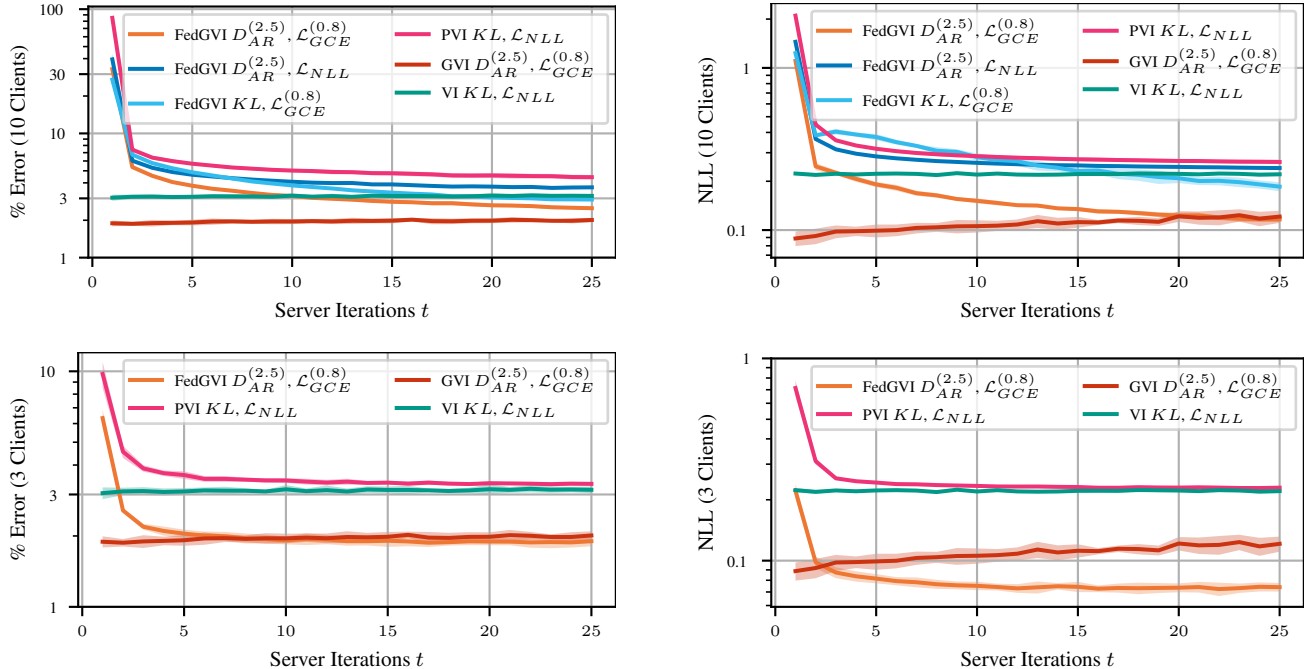

Figure 7: Accuracy (% Error) and Negative Log Likelihood (NLL) results when running fully connected BNNs, with a Mean–Field Gaussian distribution, on the MNIST data set with FEDGVI. The training data set is contaminated by 10% random label flipping, fixed across all repetitions. We average over five runs with random, homogeneous client splits.

neurons; for FEDAVG, FEDPA, and $\beta$–PREDBAYES, two hidden layers with 100 neurons in each. Data is distributed homogeneously across clients, using 5 different, randomly chosen seeds. We demonstrate that under model misspecification, FEDGVI significantly outperforms competing FL methods. Furthermore, FEDGVI incurs no additional computational complexity when compared to PVI. This is due to the KL and Alpha-Rényi divergences having closed form solutions between Multivariate Gaussians with complexity of $\mathcal{O}(1)$ in each other, and as we require $\mathcal{O}(1)$ additional, constant operations to get the GCE from the NLL.

We provide further experiments in Appendix D on the runtime of FEDGVI against PVI, learning rate selection, stability of posteriors under small perturbations in the robust loss parameters, and showing that using a single hidden layer NN for the competing methods would either negatively, or not significantly, affect their performance.

## 6. Conclusions and Future Work

We have introduced FEDGVI, a novel probabilistic approach to federated learning that is provably robust to model misspecification, and allows for faster, conjugate client updates. The theoretical analysis of FEDGVI demonstrates it's appealing properties; we easily recover existing methods as restricted cases, and characterise the convergence behaviour

at fixed points of FEDGVI as solving a global GVI optimisation problem, extending existing theory. Our result on provable robustness to outliers through FEDGVI allows for closed form, conjugate posteriors that are computationally efficient, and robust to model misspecification. In deriving this, we have also shown that the cavity distribution is necessary as predictions would otherwise be overly confident and biased. The robustness of FEDGVI was further demonstrated empirically on multiple synthetic and real–world data sets, showing outperformance of existing FL methods across model architectures and misspecification levels.

An interesting future direction is to extend FedGVI within personalised FL settings (Kotelevskii et al., 2022) and hierarchical Bayesian FL through latent variables (Kim & Hospedales, 2023) as well as through the use of a structured posterior approximation (Hassan et al., 2024), in order to incorporate client level variations. Incorporating the hierarchical model structures and additional inductive biases from such settings, while maintaining conjugacy and favourable computational complexity, remain as open challenges. In future work, we further aim to address the robust Bayesian nonparametric setting of FL through FEDGVI, as well as investigate other types of robustness, including to adversarial and Byzantine attacks, by for instance using a robust aggregator in Equation (6), and addressing the open problem of provable robustness to prior misspecification in GVI.

## Acknowledgements

OH, TM and TD acknowledge support from a UKRI Turing AI Acceleration Fellowship [EP/V02678X/1] and a Turing Impact Award from the Alan Turing Institute. For the purpose of open access, the authors have applied a Creative Commons Attribution (CC-BY) license to any Author Accepted Manuscript version arising from this submission. The authors acknowledge the University of Warwick Research Technology Platform for assistance in the research described in this paper.

## Impact Statement

This paper presents work on robust federated learning, a framework that aims to not only advance the field of machine learning, but also to develop methods that ensure the privacy of data sources, whilst aiming to achieve optimal performance even under contamination of the data. This approach, however, may discard low probability, tail events that could represent minority groups. Hence, the trade off between robustness and inclusivity is a fundamental ethical challenge for decision makers.

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

# Supplementary Material for:
# Federated Generalised Variational Inference:
# A Robust Probabilistic Federated Learning Framework

The appendix is structured as follows: Appendix A summarises the notation used throughout the paper and in the proofs. In Appendix B we present complete proofs of all theorems, propositions and lemmas given in the main paper. Appendix C clarifies the requirements of Definition 4.11, the GBI learning rate, and places FEDGVI in the broader GVI literature. Lastly, Appendix D gives additional details about the implementation of FEDGVI and additional experiments.

## A. Notation

In this section, we give definitions of the symbols used throughout the paper and the appendix.

$P_0$   The abstract and unknown probability measure, also called data generating process, acting on some abstract measurable space $(\Omega, \mathcal{F})$ which gives rise to the data

$\{x_i, y_i\}_{i=1}^n$   Entire data set of all clients, also written as $\{x_1^n, y_1^n\}$, for $x_i \in \Xi$ and $y_i | x_i \in \Upsilon$

$\{\mathbf{x}_m, \mathbf{y}_m\}_{m=1}^M$   The entire set of data points split across $M$ clients labelled $m \in [M] := \{1, 2, ..., M\}$

$\Xi$   The data space, which is assumed to have Polish topology

$\Upsilon$   The output space, which can be categorical such as in classification where $\Upsilon = [C]$, or real valued as in regression $\Upsilon = \mathbb{R}^C, C \in \mathbb{N}$

$\Theta$   In the parametric setting this is the parameter space $\boldsymbol{\theta} \in \Theta$, assumed to admit Polish topology

$\mathcal{P}(\Theta)$   The space of probability measures over the measurable space $(\Theta, \mathcal{T})$. We refer to distributions in this space, where we mean distribution functions given rise to by measures in this space. Note that these need not be continuous, and could only be defined almost everywhere in $\boldsymbol{\theta}$.

$\mathcal{Q}$   A variational family of distributions such that $\mathcal{Q} \subset \mathcal{P}(\Theta)$ and, in terms of distributions, $\mathcal{Q} = \{q(\boldsymbol{\theta}|\boldsymbol{\kappa}) \in \mathcal{P}(\Theta) : \boldsymbol{\kappa} \in \boldsymbol{K}\}$, where $\boldsymbol{K}$ is a set of variational parameters

$\pi(\boldsymbol{\theta})$   The prior distribution, given rise to by the prior measure $\Pi$ on $(\Theta, \mathcal{T})$

$L_m^{(t)}(\mathbf{y}_m; \boldsymbol{\theta}, \mathbf{x}_m)$   The local loss of client $m$, at iteration $t \in [T]$, on the local data set $\{\mathbf{x}_m, \mathbf{y}_m\}$, not necessarily the same across clients nor iterations, and associated with the parameters $\boldsymbol{\theta} \in \Theta$

$\ell_m^{(t)}(\boldsymbol{\theta})$   Local loss approximation of $L_m(\mathbf{y}_m; \boldsymbol{\theta}, \mathbf{x}_m)$ and the impact of the data of client $m$ on the posterior at the server

$\Delta_m^{(t)}(\boldsymbol{\theta})$   Local update, Equation (5), that represents the change in the approximate posteriors, and the de facto change in the local loss approximation. It has associated damping parameter $\tau$.

$\ell_s^{(t)}(\boldsymbol{\theta})$   Global loss approximation of all clients aggregated at the server

$q_m^{(t)}(\boldsymbol{\theta})$   Local posterior computed through Equation (4)

$q_s^{(t)}(\boldsymbol{\theta})$   Global approximate posterior after server–side optimisation step, Equation (7)

$P(L, D, \mathcal{Q})$   The Rule of Three (Knoblauch et al., 2022) that defines a global GVI objective

$D$   Any statistical divergence $D : \mathcal{P}(\Theta) \times \mathcal{P}(\Theta) \to \mathbb{R}_{\geq 0}$ (for a detailed definition see Nielsen, 2020); $D_s$ denotes the divergence at the server.

$\mathbb{E}_{q(\boldsymbol{\theta})}$   The expectation with respect to $q(\boldsymbol{\theta})$

## B. Proofs of Theorems, Propositions, and Lemmas

Here, we provide the full proofs of the theorems stated in the paper. Throughout, we assume that all the losses, distributions and approximate losses, are measurable with respect to some dominating measure $\mu(d\boldsymbol{\theta})$. This can be the Lebesgue measure in finite dimensional spaces, or more generally the Haar measure. For infinite dimensional measure spaces, which are of interest in the study of Bayesian inverse problems and nonparametrics, we could assume $\mu(d\boldsymbol{\theta})$ to be a Gaussian measure as

in Pinski et al. (2015).

## B.1. Equivalence Between The KL Divergence and Weighted KL Divergence

First, we present a well known auxiliary lemma that will be used throughout the proofs. It states that the weighted KL divergence is equivalent to using a tempered or weighted likelihood in the optimisation procedure, and hence lead to equivalent inference problems (Knoblauch et al., 2022; Bissiri et al., 2016). So without loss of generality, we can push the weighting term of the KL divergence inside the loss, by defining the loss to be $L = w \cdot L$, which does not change the optimisation procedure. We show this result for $f$–divergences, which we define as in Ali & Silvey (1966) and Amari (2016).

**Lemma B.1.** *For $w > 0$ the posteriors computed by the weighted $f$–divergence, $D = \frac{1}{w} D_f$ and loss $L$, and the posterior through the $f$–divergence $D = D_f$ and weighted loss $w \cdot L$ are equivalent, i.e.,*

$$P(L, \frac{1}{w} D_f, \mathcal{Q}) = P(w \cdot L, D_f, \mathcal{Q})$$

**Proof**

$$P(L, \frac{1}{w} D_f, \mathcal{Q}) = \underset{q \in \mathcal{Q}}{\arg\min} \left\{ \mathbb{E}_{q(\boldsymbol{\theta})} \left[ L(\mathbf{y}; \boldsymbol{\theta}, \mathbf{x}) \right] + \frac{1}{w} D_f(q : \pi) \right\}$$

$$= \underset{q \in \mathcal{Q}}{\arg\min} \left\{ \mathbb{E}_{q(\boldsymbol{\theta})} \left[ L(\mathbf{y}; \boldsymbol{\theta}, \mathbf{x}) \right] + \frac{1}{w} \mathbb{E}_{q(\boldsymbol{\theta})} \left[ f \left( \frac{q(\boldsymbol{\theta})}{\pi(\boldsymbol{\theta})} \right) \right] \right\}$$

$$= \underset{q \in \mathcal{Q}}{\arg\min} \left\{ \frac{1}{w} \mathbb{E}_{q(\boldsymbol{\theta})} \left[ w \cdot L(\mathbf{y}; \boldsymbol{\theta}, \mathbf{x}) + f \left( \frac{q(\boldsymbol{\theta})}{\pi(\boldsymbol{\theta})} \right) \right] \right\}$$

$$= \underset{q \in \mathcal{Q}}{\arg\min} \left\{ \mathbb{E}_{q(\boldsymbol{\theta})} \left[ w \cdot L(\mathbf{y}; \boldsymbol{\theta}, \mathbf{x}) + f \left( \frac{q(\boldsymbol{\theta})}{\pi(\boldsymbol{\theta})} \right) \right] \right\}$$

$$= \underset{q \in \mathcal{Q}}{\arg\min} \left\{ \mathbb{E}_{q(\boldsymbol{\theta})} \left[ w \cdot L(\mathbf{y}; \boldsymbol{\theta}, \mathbf{x}) \right] + D_f(q : \pi) \right\} := P(w \cdot L, D_f, \mathcal{Q})$$

■

Therefore, when referring to the loss in the following we mean it to be the weighted loss so that we can utilise the weighted KL divergence. This easily recovers the KL–divergence for $f : u \mapsto -\log u$.

## B.2. Proposition 4.3: A Logarithmic Opinion Pool through Damping

**Proof** Consider the server update at some iteration $t$, where we gather the client updates. Under the KL divergence, we then solve the server optimisation procedure as:

$$q_s^{(t)}(\boldsymbol{\theta}) = \underset{q \in \mathcal{Q}}{\arg\min} \left\{ \mathbb{E}_{q(\boldsymbol{\theta})} \left[ \ell_s^{(t)}(\boldsymbol{\theta}) \right] + KL(q : \pi) \right\} = \underset{q \in \mathcal{Q}}{\arg\min} \left\{ \mathbb{E}_{q(\boldsymbol{\theta})} \left[ \log \frac{q(\boldsymbol{\theta})}{\pi(\boldsymbol{\theta}) \exp\{-\ell_s^{(t)}(\boldsymbol{\theta})\}} \right] \right\}$$

we know that this is minimised at:

$$q_s^{(t)}(\boldsymbol{\theta}) \propto \pi(\boldsymbol{\theta}) \exp\{-\ell_s^{(t)}(\boldsymbol{\theta})\} = \pi(\boldsymbol{\theta}) \exp \left\{ -\ell_s^{(t-1)}(\boldsymbol{\theta}) - \sum_{m=1}^{M} \Delta_m^{(t)}(\boldsymbol{\theta}) \right\}$$

$$\propto \underbrace{\pi(\boldsymbol{\theta}) \exp\{-\ell_s^{(t-1)}(\boldsymbol{\theta})\}}_{\propto q_s^{(t-1)}(\boldsymbol{\theta})} \exp \left\{ -\sum_{m=1}^{M} -\tau_m \log \frac{q_m^{(t)}(\boldsymbol{\theta})}{q_s^{(t-1)}(\boldsymbol{\theta})} \right\} \propto q_s^{(t-1)}(\boldsymbol{\theta}) \prod_{m=1}^{M} \left( \frac{q_m^{(t)}(\boldsymbol{\theta})}{q_s^{(t-1)}(\boldsymbol{\theta})} \right)^{\tau_m}$$

$$= \frac{q_s^{(t-1)}(\boldsymbol{\theta}) \prod_{m=1}^{M} (q_m^{(t)}(\boldsymbol{\theta}))^{\tau_m}}{(q_s^{(t-1)}(\boldsymbol{\theta}))^{\sum_{m=1}^{M} \tau_m}}$$

By assumption we have that $\sum_{m=1}^{M} \tau_m = 1$, therefore $(q_s^{(t-1)}(\boldsymbol{\theta}))^{\sum_{m=1}^{M} \tau_m} = q_s^{(t-1)}(\boldsymbol{\theta})$ and:

$$q_s^{(t)}(\boldsymbol{\theta}) \propto \prod_{m=1}^{M} (q_m^{(t)}(\boldsymbol{\theta}))^{\tau_m}$$

$$q_s^{(t)}(\boldsymbol{\theta}) = \frac{\prod_{m=1}^{M}(q_m^{(t)}(\boldsymbol{\theta}))^{\tau_m}}{\int_{\Theta}\prod_{m=1}^{M}(q_m^{(t)}(\boldsymbol{\theta}))^{\tau_m}\,\mu(d\boldsymbol{\theta})}, \ \mu - a.e.$$

This forms an externally Bayesian logarithmic opinion pool (Genest, 1984; Genest et al., 1986). $\blacksquare$

### B.3. Proof of Proposition 4.4

The proof of Proposition 4.4 is adapted from that for Partitioned Variational Inference in Ashman et al. (2022). We show the proof of Proposition 4.4 by comparing the derivatives with respect to the variational parameters of $q(\boldsymbol{\theta}|\boldsymbol{\kappa})$ of the sum of local objectives with those of the global objective. This is motivated by the equivalence of a sum of local GVI objectives (from each client) with some added constants and the global GVI objective, demonstrated in Appendix B.3.1. The main proof is in Appendix B.3.2.

#### B.3.1. RECOVERING A GLOBAL GVI OBJECTIVE FROM LOCAL OBJECTIVES

First, we provide an analogue of Ashman et al. (2022, Property 2) which states that the sum of the local (client) FEDGVI objectives and some constant, which we find to be the negative log normalising constants of the cavity and the server distributions, equals the global GVI objective. We define the following:

$$q_s^{(t)}(\boldsymbol{\theta}) = \frac{1}{Z_{q_s^{(t)}}}\pi(\boldsymbol{\theta})\exp\{-\sum_{m=1}^{M}\ell_m^{(t)}(\boldsymbol{\theta})\}$$

$$q^{\backslash m}(\boldsymbol{\theta}) = \frac{1}{Z_{q^{\backslash m}}}\pi(\boldsymbol{\theta})\exp\{-\sum_{k\neq m}\ell_k^{(t)}(\boldsymbol{\theta})\} \propto \frac{q_s^{(t)}(\boldsymbol{\theta})}{\exp\{-\ell_m^{(t)}(\boldsymbol{\theta})\}}$$

$$\mathrm{Obj}(m, q_s^{(t)}) := \mathbb{E}_{q(\boldsymbol{\theta})}\left[L_m(\mathbf{y}_m;\boldsymbol{\theta},\mathbf{x}_m)\right] + \frac{1}{w}D_{KL}(q : q^{\backslash m})$$

$$\mathrm{Obj}(q_s^{(t)}) := \mathbb{E}_{q(\boldsymbol{\theta})}\left[\sum_{m=1}^{M}L_m(\mathbf{y}_m;\boldsymbol{\theta},\mathbf{x}_m)\right] + \frac{1}{w}D_{KL}(q : \pi)$$

Then we can recover the global objective by summing over the local objectives and subtracting the log normalising constants of the cavity distributions and the current server posterior.

$$\sum_{m=1}^{M}\mathrm{Obj}(m, q_s^{(t)}) - \frac{1}{w}(\log Z_{q_s^{(t)}} + \sum_{m=1}^{M}\log Z_{q^{\backslash m}})$$

$$= \sum_{m=1}^{M}\left(\mathbb{E}_{q(\boldsymbol{\theta})}\left[L_m(\mathbf{y}_m;\boldsymbol{\theta},\mathbf{x}_m)\right] + \frac{1}{w}D_{KL}(q : q^{\backslash m})\right) - \frac{1}{w}(\log Z_{q_s^{(t)}} + \sum_{m=1}^{M}\log Z_{q^{\backslash m}})$$

$$= \sum_{m=1}^{M}\mathbb{E}_{q(\boldsymbol{\theta})}\left[L_m(\mathbf{y}_m;\boldsymbol{\theta},\mathbf{x}_m)\right] + \sum_{m=1}^{M}\frac{1}{w}\mathbb{E}_{q(\boldsymbol{\theta})}\left[\log\frac{q(\boldsymbol{\theta})}{q^{\backslash m}(\boldsymbol{\theta})}\right] - \frac{1}{w}(\log Z_{q_s^{(t)}} + \sum_{m=1}^{M}\log Z_{q^{\backslash m}})$$

$$= \mathbb{E}_{q(\boldsymbol{\theta})}\left[\sum_{m=1}^{M}L_m(\mathbf{y}_m;\boldsymbol{\theta},\mathbf{x}_m)\right] + \frac{1}{w}\mathbb{E}_{q(\boldsymbol{\theta})}\left[\sum_{m=1}^{M}\log\frac{q(\boldsymbol{\theta})}{q^{\backslash m}(\boldsymbol{\theta})}\right] - \frac{1}{w}(\log Z_{q_s^{(t)}} + \sum_{m=1}^{M}\log Z_{q^{\backslash m}})$$

$$= \mathbb{E}_{q(\boldsymbol{\theta})}\left[\sum_{m=1}^{M}L_m(\mathbf{y}_m;\boldsymbol{\theta},\mathbf{x}_m)\right] + \frac{1}{w}\mathbb{E}_{q(\boldsymbol{\theta})}\left[\log\prod_{m=1}^{M}\frac{q(\boldsymbol{\theta})\exp\{-\ell_m^{(t)}(\boldsymbol{\theta})\}}{q_s^{(t)}(\boldsymbol{\theta})}\right] - \frac{1}{w}(\log Z_{q_s^{(t)}} + \sum_{m=1}^{M}\log Z_{q^{\backslash m}})$$

$$= \mathbb{E}_{q(\boldsymbol{\theta})}\left[\sum_{m=1}^{M}L_m(\mathbf{y}_m;\boldsymbol{\theta},\mathbf{x}_m)\right] + \frac{1}{w}\mathbb{E}_{q(\boldsymbol{\theta})}\left[\log\frac{q(\boldsymbol{\theta})\exp\{-\sum_{m=1}^{M}\ell_m^{(t)}(\boldsymbol{\theta})\}}{q_s^{(t)}(\boldsymbol{\theta})}\right] - \frac{1}{w}\log Z_{q_s^{(t)}}$$

$$= \mathbb{E}_{q(\boldsymbol{\theta})}\left[\sum_{m=1}^{M}L_m(\mathbf{y}_m;\boldsymbol{\theta},\mathbf{x}_m)\right] + \frac{1}{w}\mathbb{E}_{q(\boldsymbol{\theta})}\left[\log\frac{q(\boldsymbol{\theta})}{\pi(\boldsymbol{\theta})/Z_{q_s^{(t)}}}\right] - \frac{1}{w}\log Z_{q_s^{(t)}} = \mathrm{Obj}(q_s^{(t)})$$

Hence, by using the weighted KL divergence at the clients optimisation step, can we recover a global GVI objective by summing over the local objectives and adding some constants independent of the variational parameters of interest in the optimisation problem. We note that the added logarithms of the normalising constants are independent of $\boldsymbol{\kappa}$, since these are fixed through the current posterior and cavity distribution and do not depend on the variational parameters.

### B.3.2. PROPOSITION 4.4: FIXED POINTS RECOVERS A GLOBAL FIXED POINT

We denote a fixed point of the algorithm as $q_s^*(\boldsymbol{\theta}|\boldsymbol{\kappa}^*)$ such that for all $m \in [M]$ we have $q_s^*(\boldsymbol{\theta}|\boldsymbol{\kappa}^*) \in \arg\min_{q\in\mathcal{Q}} \mathrm{Obj}(m, q_s^*)$, then we have the property that no update will change the posterior found. Recall:

**Proposition 4.4** *Let $D = \frac{1}{w}D_{KL}$ at the clients, local loss $L_m$ and $\mathcal{Q} := \{q(\boldsymbol{\theta}|\boldsymbol{\kappa}) : \boldsymbol{\kappa} \in \boldsymbol{K}\} \subset \mathcal{P}(\Theta)$ as a variational family. Assume that FEDGVI finds a fixed point $q_s^*(\boldsymbol{\theta}|\boldsymbol{\kappa}^*)$, such that for all clients we have that $q_s^*(\boldsymbol{\theta}|\boldsymbol{\kappa}^*) \in \arg\min_{q\in\mathcal{Q}} \mathrm{Obj}(m, q_s^*)$. Then, it holds that $q_s^*(\boldsymbol{\theta}|\boldsymbol{\kappa}^*) \in \arg\min_{q\in\mathcal{Q}} \mathrm{Obj}(q_s^*)$.*

**Proof** First we note that we consider only the KL divergence in this proof, which is equivalent to saying we modify the loss $L$ to be multiplied by $w > 0$, which results in the equivalent formulation, as shown in Knoblauch et al. (2022) where $P(L, \frac{1}{w}D_{KL}, \mathcal{Q}) = P(w \cdot L, KL, \mathcal{Q})$, see also Lemma B.1.

Note that the condition $\forall m \in [M]$ we have that $q_s^*(\boldsymbol{\theta}|\boldsymbol{\kappa}^*) \in \arg\min_{q\in\mathcal{Q}} \mathrm{Obj}(m, q_s^*)$ is equivalent to requiring that $\Delta_m^*(\boldsymbol{\theta}) = 0$, since this means that the local loss approximations remain unchanged and hence $\ell_s^*(\boldsymbol{\theta})$ remains unchanged. This then implies that the posterior at the server will not change. This is the same as saying that the client optimisation step has found the global solution and hence $q_m^*(\boldsymbol{\theta})$ and $q_s^*(\boldsymbol{\theta})$ will be the same which implies that $\Delta_m^*(\boldsymbol{\theta}) = 0$.

In the following all integrals are assumed to be over the parameter space $\Theta$, even when we don't make it explicit.

We can furthermore show that we can express the derivative of the local objective as a single integral under the weighted KL divergence.

$$\nabla_{\boldsymbol{\kappa}}\mathrm{Obj}(m, q_s^*) = \nabla_{\boldsymbol{\kappa}} \left\{ \mathbb{E}_{q(\boldsymbol{\theta})}[L_m(\mathbf{y}_m; \boldsymbol{\theta}, \mathbf{x}_m)] + D_{KL}\left(q : \frac{q_s^*(\boldsymbol{\theta}|\boldsymbol{\kappa}^*)}{\exp\{-\ell_m^*(\boldsymbol{\theta}|\boldsymbol{\kappa}^*)\}Z_{q_s}^*}\right) \right\}$$

$$= \nabla_{\boldsymbol{\kappa}} \int q(\boldsymbol{\theta}|\boldsymbol{\kappa}) \log \frac{1}{\exp\{-L_m(\mathbf{y}_m; \boldsymbol{\theta}, \mathbf{x}_m)\}} + q(\boldsymbol{\theta}|\boldsymbol{\kappa})\left(\log \frac{q(\boldsymbol{\theta}|\boldsymbol{\kappa})\exp\{-\ell_m^*(\boldsymbol{\theta}|\boldsymbol{\kappa}^*)\}}{q_s^*(\boldsymbol{\theta}|\boldsymbol{\kappa}^*)} + \log Z_{q_s}^*\right) \mu(d\boldsymbol{\theta})$$

$$= \nabla_{\boldsymbol{\kappa}} \int q(\boldsymbol{\theta}|\boldsymbol{\kappa}) \log \frac{q(\boldsymbol{\theta}|\boldsymbol{\kappa})\exp\{-\ell_m^*(\boldsymbol{\theta}|\boldsymbol{\kappa}^*)\}}{q_s^*(\boldsymbol{\theta}|\boldsymbol{\kappa}^*)\exp\{-L_m(\mathbf{y}_m; \boldsymbol{\theta}, \mathbf{x}_m)\}} \mu(d\boldsymbol{\theta}) + \nabla_{\boldsymbol{\kappa}} \log Z_{q_s}^* \underset{\nearrow}{\overset{0}{\int q(\boldsymbol{\theta}|\boldsymbol{\kappa})\,\mu(d\boldsymbol{\theta})}}$$

Now we first show that the fixed point is an extremum of the global objective and then that it is a minimum. We do this by first differentiating the local objective with respect to the variational parameters $\boldsymbol{\kappa}$ and then that the sum of the local derivatives evaluated at $\boldsymbol{\kappa} = \boldsymbol{\kappa}^*$ equal the derivative of the global objective.

$$\nabla_{\boldsymbol{\kappa}}\mathrm{Obj}(m, q_s^*) = \nabla_{\boldsymbol{\kappa}} \int q(\boldsymbol{\theta}|\boldsymbol{\kappa}) \log \frac{q(\boldsymbol{\theta}|\boldsymbol{\kappa})\exp\{-\ell_m^*(\boldsymbol{\theta}|\boldsymbol{\kappa}^*)\}}{q_s^*(\boldsymbol{\theta}|\boldsymbol{\kappa}^*)\exp\{-L_m(\mathbf{y}_m; \boldsymbol{\theta}, \mathbf{x}_m)\}} \mu(d\boldsymbol{\theta})$$

$$= \nabla_{\boldsymbol{\kappa}} \int q(\boldsymbol{\theta}|\boldsymbol{\kappa})(L_m(\mathbf{y}_m; \boldsymbol{\theta}, \mathbf{x}_m) - \ell_m^*(\boldsymbol{\theta}|\boldsymbol{\kappa}^*))\, \mu(d\boldsymbol{\theta}) + \nabla_{\boldsymbol{\kappa}} \int q(\boldsymbol{\theta}|\boldsymbol{\kappa}) \log \frac{q(\boldsymbol{\theta}|\boldsymbol{\kappa})}{q_s^*(\boldsymbol{\theta}|\boldsymbol{\kappa}^*)}\, \mu(d\boldsymbol{\theta})$$

$$= \nabla_{\boldsymbol{\kappa}} \int q(\boldsymbol{\theta}|\boldsymbol{\kappa})(L_m(\mathbf{y}_m; \boldsymbol{\theta}, \mathbf{x}_m) - \ell_m^*(\boldsymbol{\theta}|\boldsymbol{\kappa}^*))\, \mu(d\boldsymbol{\theta}) + \int (\nabla_{\boldsymbol{\kappa}} q(\boldsymbol{\theta}|\boldsymbol{\kappa})) \log \frac{q(\boldsymbol{\theta}|\boldsymbol{\kappa})}{q_s^*(\boldsymbol{\theta}|\boldsymbol{\kappa}^*)}\, \mu(d\boldsymbol{\theta})$$

$$+ \int \underset{\nearrow}{\overset{0}{\nabla_{\boldsymbol{\kappa}} q(\boldsymbol{\theta}|\boldsymbol{\kappa})\,\mu(d\boldsymbol{\theta})}}$$

where first line follows since we can compose the expectation and (weighted) KL divergence and the normalising constant of the cavity distribution is constant with respect to $\boldsymbol{\kappa}$. The last line follows from the fact that $\frac{d}{dx}f(x)\log f(x) = f'(x)\log f(x) + f'(x)$ and that we can exchange the order of integration and differentiation. We further note that at convergence, where $\boldsymbol{\kappa} = \boldsymbol{\kappa}^*$, that $\log \frac{q(\boldsymbol{\theta}|\boldsymbol{\kappa})}{q_s^*(\boldsymbol{\theta}|\boldsymbol{\kappa}^*)}\big|_{\boldsymbol{\kappa}=\boldsymbol{\kappa}^*} = 0$. Evaluating the expression above at $\boldsymbol{\kappa} = \boldsymbol{\kappa}^*$ then yields:

$$\nabla_{\boldsymbol{\kappa}}\mathrm{Obj}(m, q_s^*)\Big|_{\boldsymbol{\kappa}=\boldsymbol{\kappa}^*} = \nabla_{\boldsymbol{\kappa}} \int q(\boldsymbol{\theta}|\boldsymbol{\kappa})(L_m(\mathbf{y}_m; \boldsymbol{\theta}, \mathbf{x}_m) - \ell_m^*(\boldsymbol{\theta}|\boldsymbol{\kappa}^*))\, \mu(d\boldsymbol{\theta})\Big|_{\boldsymbol{\kappa}=\boldsymbol{\kappa}^*}$$

Summing over all these client objectives then yields the following expression:

$$\sum_{m=1}^{M} \nabla_{\boldsymbol{\kappa}} \mathrm{Obj}(m, q_s^*)\Big|_{\boldsymbol{\kappa}=\boldsymbol{\kappa}^*} = \sum_{m=1}^{M} \nabla_{\boldsymbol{\kappa}} \int q(\boldsymbol{\theta}|\boldsymbol{\kappa})(L_m(\mathbf{y}_m; \boldsymbol{\theta}, \mathbf{x}_m) - \ell_m^*(\boldsymbol{\theta}|\boldsymbol{\kappa}^*))\, \mu(d\boldsymbol{\theta})\Big|_{\boldsymbol{\kappa}=\boldsymbol{\kappa}^*}$$

$$= \nabla_{\boldsymbol{\kappa}} \int q(\boldsymbol{\theta}|\boldsymbol{\kappa})(\sum_{m=1}^{M} L_m(\mathbf{y}_m; \boldsymbol{\theta}, \mathbf{x}_m) - \sum_{m=1}^{M} \ell_m^*(\boldsymbol{\theta}|\boldsymbol{\kappa}^*))\, \mu(d\boldsymbol{\theta})\Big|_{\boldsymbol{\kappa}=\boldsymbol{\kappa}^*}$$

$$= \int (\nabla_{\boldsymbol{\kappa}} q(\boldsymbol{\theta}|\boldsymbol{\kappa})) \log \frac{q_s^*(\boldsymbol{\theta}|\boldsymbol{\kappa}^*)}{\pi(\boldsymbol{\theta}) \exp\{\sum_{m=1}^{M} L_m(\mathbf{y}_m; \boldsymbol{\theta}, \mathbf{x}_m)\}}\, \mu(d\boldsymbol{\theta})\Big|_{\boldsymbol{\kappa}=\boldsymbol{\kappa}^*}$$

$$+ \nabla_{\boldsymbol{\kappa}} \int q(\boldsymbol{\theta}|\boldsymbol{\kappa}) \log Z_{q^*}\, \mu(d\boldsymbol{\theta})^{\nearrow 0}$$

To compare this with a global fixed point we differentiate the global objective at $q^*$, not yet assumed to be a minimiser of the global objective, with respect to the variational parameters.

$$\nabla_{\boldsymbol{\kappa}} \mathrm{Obj}(q_s^*) = \nabla_{\boldsymbol{\kappa}} \int q(\boldsymbol{\theta}|\boldsymbol{\kappa}) \log \frac{q(\boldsymbol{\theta}|\boldsymbol{\kappa})}{\pi(\boldsymbol{\theta}) \exp\{\sum_{m=1}^{M} L_m(\mathbf{y}_m; \boldsymbol{\theta}, \mathbf{x}_m)\}}\, \mu(d\boldsymbol{\theta})$$

$$= \int (\nabla_{\boldsymbol{\kappa}} q(\boldsymbol{\theta}|\boldsymbol{\kappa})) \log \frac{q(\boldsymbol{\theta}|\boldsymbol{\kappa})}{\pi(\boldsymbol{\theta}) \exp\{\sum_{m=1}^{M} L_m(\mathbf{y}_m; \boldsymbol{\theta}, \mathbf{x}_m)\}}\, \mu(d\boldsymbol{\theta}) + \int \nabla_{\boldsymbol{\kappa}} q(\boldsymbol{\theta}|\boldsymbol{\kappa})\, \mu(d\boldsymbol{\theta})^{\nearrow 0}$$

Then,

$$\nabla_{\boldsymbol{\kappa}} \mathrm{Obj}(q_s^*)\Big|_{\boldsymbol{\kappa}=\boldsymbol{\kappa}^*} = \int (\nabla_{\boldsymbol{\kappa}} q(\boldsymbol{\theta}|\boldsymbol{\kappa})) \log \frac{q(\boldsymbol{\theta}|\boldsymbol{\kappa})}{\pi(\boldsymbol{\theta}) \exp\{\sum_{m=1}^{M} L_m(\mathbf{y}_m; \boldsymbol{\theta}, \mathbf{x}_m)\}}\, \mu(d\boldsymbol{\theta})\Big|_{\boldsymbol{\kappa}=\boldsymbol{\kappa}^*} = \sum_{m=1}^{M} \nabla_{\boldsymbol{\kappa}} \mathrm{Obj}(m, q_s^*)\Big|_{\boldsymbol{\kappa}=\boldsymbol{\kappa}^*}$$

And since $q_s^*(\boldsymbol{\theta}|\boldsymbol{\kappa}^*)$ is a fixed point of each client, we have that $\nabla_{\boldsymbol{\kappa}} \mathrm{Obj}(m, q_s^*)\big|_{\boldsymbol{\kappa}=\boldsymbol{\kappa}^*} = 0$. Therefore,

$$\sum_{m=1}^{M} \nabla_{\boldsymbol{\kappa}} \mathrm{Obj}(m, q_s^*)\Big|_{\boldsymbol{\kappa}=\boldsymbol{\kappa}^*} = 0 \qquad \implies \qquad \nabla_{\boldsymbol{\kappa}} \mathrm{Obj}(q_s^*)\Big|_{\boldsymbol{\kappa}=\boldsymbol{\kappa}^*} = 0$$

This means that $q_s^*(\boldsymbol{\theta}|\boldsymbol{\kappa}^*)$ is an extremum of FEDGVI, and further that it is also an extremum of GVI with $D = \frac{1}{w} D_{KL}$. We now show that it is further a minimum of the global GVI objective. We consider the Hessian $\nabla \nabla_{\boldsymbol{\kappa}}$ and proceed like before.

$$\nabla \nabla_{\boldsymbol{\kappa}} \mathrm{Obj}(m, q_s^*) = \nabla \nabla_{\boldsymbol{\kappa}} \int q(\boldsymbol{\theta}|\boldsymbol{\kappa}) \log \frac{q(\boldsymbol{\theta}|\boldsymbol{\kappa}) \exp\{-\ell_m^*(\boldsymbol{\theta}|\boldsymbol{\kappa}^*)\}}{q_s^*(\boldsymbol{\theta}|\boldsymbol{\kappa}^*) \exp\{-L_m(\mathbf{y}_m; \boldsymbol{\theta}, \mathbf{x}_m)\}}\, \mu(d\boldsymbol{\theta})$$

$$= \nabla \nabla_{\boldsymbol{\kappa}} \int q(\boldsymbol{\theta}|\boldsymbol{\kappa})(L_m(\mathbf{y}_m; \boldsymbol{\theta}, \mathbf{x}_m) - \ell_m^*(\boldsymbol{\theta}|\boldsymbol{\kappa}^*))\, \mu(d\boldsymbol{\theta}) + \nabla \nabla_{\boldsymbol{\kappa}} \int q(\boldsymbol{\theta}|\boldsymbol{\kappa}) \log \frac{q(\boldsymbol{\theta}|\boldsymbol{\kappa})}{q_s^*(\boldsymbol{\theta}|\boldsymbol{\kappa}^*)}\, \mu(d\boldsymbol{\theta})$$

$$= \nabla \nabla_{\boldsymbol{\kappa}} \int q(\boldsymbol{\theta}|\boldsymbol{\kappa})(L_m(\mathbf{y}_m; \boldsymbol{\theta}, \mathbf{x}_m) - \ell_m^*(\boldsymbol{\theta}|\boldsymbol{\kappa}^*))\, \mu(d\boldsymbol{\theta})$$

$$+ \nabla_{\boldsymbol{\kappa}} \left( \int (\nabla_{\boldsymbol{\kappa}} q(\boldsymbol{\theta}|\boldsymbol{\kappa})) \log \frac{q(\boldsymbol{\theta}|\boldsymbol{\kappa})}{q_s^*(\boldsymbol{\theta}|\boldsymbol{\kappa}^*)}\, \mu(d\boldsymbol{\theta}) + \int \nabla_{\boldsymbol{\kappa}} \log q(\boldsymbol{\theta}|\boldsymbol{\kappa})\, \mu(d\boldsymbol{\theta})^{\nearrow 0} \right)$$

$$= \nabla \nabla_{\boldsymbol{\kappa}} \int q(\boldsymbol{\theta}|\boldsymbol{\kappa})(L_m(\mathbf{y}_m; \boldsymbol{\theta}, \mathbf{x}_m) - \ell_m^*(\boldsymbol{\theta}|\boldsymbol{\kappa}^*))\, \mu(d\boldsymbol{\theta})$$

$$+ \int (\nabla \nabla_{\boldsymbol{\kappa}} q(\boldsymbol{\theta}|\boldsymbol{\kappa})) \log \frac{q(\boldsymbol{\theta}|\boldsymbol{\kappa})}{q_s^*(\boldsymbol{\theta}|\boldsymbol{\kappa}^*)}\, \mu(d\boldsymbol{\theta}) + \int (\nabla_{\boldsymbol{\kappa}} q(\boldsymbol{\theta}|\boldsymbol{\kappa}))(\nabla_{\boldsymbol{\kappa}} \log q(\boldsymbol{\theta}|\boldsymbol{\kappa}))\, \mu(d\boldsymbol{\theta})$$

Ashman et al. (2022) point out that this last term can equivalently be expressed through it's transpose.

$$\left( \int (\nabla_{\boldsymbol{\kappa}} q(\boldsymbol{\theta}|\boldsymbol{\kappa}))(\nabla_{\boldsymbol{\kappa}} \log q(\boldsymbol{\theta}|\boldsymbol{\kappa}))\, \mu(d\boldsymbol{\theta}) \right)^{\top}$$

$$= \nabla_{\boldsymbol{\kappa}} \int q(\boldsymbol{\theta}|\boldsymbol{\kappa})(\nabla_{\boldsymbol{\kappa}} \log q(\boldsymbol{\theta}|\boldsymbol{\kappa}))\,\mu(d\boldsymbol{\theta}) + \overbrace{\int \nabla\nabla_{\boldsymbol{\kappa}}q(\boldsymbol{\theta}|\boldsymbol{\kappa})\,\mu(d\boldsymbol{\theta})}^{0}$$

$$= \nabla_{\boldsymbol{\kappa}} \int q(\boldsymbol{\theta}|\boldsymbol{\kappa})\frac{1}{q(\boldsymbol{\theta}|\boldsymbol{\kappa})}\,\mu(d\boldsymbol{\theta}) = \mathbf{0}$$

Evaluating this Hessian at $\boldsymbol{\kappa} = \boldsymbol{\kappa}^*$:

$$\nabla\nabla_{\boldsymbol{\kappa}}\mathrm{Obj}(m, q_s^*)\Big|_{\boldsymbol{\kappa}=\boldsymbol{\kappa}^*} =$$

$$\nabla\nabla_{\boldsymbol{\kappa}} \int q(\boldsymbol{\theta}|\boldsymbol{\kappa})(L_m(\mathbf{y}_m;\boldsymbol{\theta},\mathbf{x}_m) - \ell_m^*(\boldsymbol{\theta}|\boldsymbol{\kappa}^*))\,\mu(d\boldsymbol{\theta})\Big|_{\boldsymbol{\kappa}=\boldsymbol{\kappa}^*} + \overbrace{\int (\nabla\nabla_{\boldsymbol{\kappa}}q(\boldsymbol{\theta}|\boldsymbol{\kappa}))\log\frac{q(\boldsymbol{\theta}|\boldsymbol{\kappa})}{q_s^*(\boldsymbol{\theta}|\boldsymbol{\kappa}^*)}\,\mu(d\boldsymbol{\theta})\Big|_{\boldsymbol{\kappa}=\boldsymbol{\kappa}^*}}^{0}$$

Therefore, when summing over the individual Hessians of the clients, we get:

$$\sum_{m=1}^{M} \nabla\nabla_{\boldsymbol{\kappa}}\mathrm{Obj}(m, q_s^*)\Big|_{\boldsymbol{\kappa}=\boldsymbol{\kappa}^*} = \sum_{m=1}^{M} \nabla\nabla_{\boldsymbol{\kappa}} \int q(\boldsymbol{\theta}|\boldsymbol{\kappa})(L_m(\mathbf{y}_m;\boldsymbol{\theta},\mathbf{x}_m) - \ell_m^*(\boldsymbol{\theta}|\boldsymbol{\kappa}^*))\,\mu(d\boldsymbol{\theta})\Big|_{\boldsymbol{\kappa}=\boldsymbol{\kappa}^*}$$

$$= \nabla\nabla_{\boldsymbol{\kappa}} \int q(\boldsymbol{\theta}|\boldsymbol{\kappa})(\sum_{m=1}^{M} L_m(\mathbf{y}_m;\boldsymbol{\theta},\mathbf{x}_m) - \sum_{m=1}^{M} \ell_m^*(\boldsymbol{\theta}|\boldsymbol{\kappa}^*))\,\mu(d\boldsymbol{\theta})\Big|_{\boldsymbol{\kappa}=\boldsymbol{\kappa}^*}$$

$$= \nabla\nabla_{\boldsymbol{\kappa}} \int q(\boldsymbol{\theta}|\boldsymbol{\kappa}) \log \frac{q_s^*(\boldsymbol{\theta}|\boldsymbol{\kappa}^*)}{\pi(\boldsymbol{\theta})\exp\{\sum_{m=1}^{M} L_m(\mathbf{y}_m;\boldsymbol{\theta},\mathbf{x}_m)\}}\,\mu(d\boldsymbol{\theta})\Big|_{\boldsymbol{\kappa}=\boldsymbol{\kappa}^*}$$

$$+ \overbrace{\nabla\nabla_{\boldsymbol{\kappa}} \int q(\boldsymbol{\theta}|\boldsymbol{\kappa}) \log Z_{q^*}\,\mu(d\boldsymbol{\theta})}^{0}$$

$$= \int (\nabla\nabla_{\boldsymbol{\kappa}}q(\boldsymbol{\theta}|\boldsymbol{\kappa})) \log \frac{q_s^*(\boldsymbol{\theta}|\boldsymbol{\kappa}^*)}{\pi(\boldsymbol{\theta})\exp\{\sum_{m=1}^{M} L_m(\mathbf{y}_m;\boldsymbol{\theta},\mathbf{x}_m)\}}\,\mu(d\boldsymbol{\theta})\Big|_{\boldsymbol{\kappa}=\boldsymbol{\kappa}^*}$$

which is a sum of positive definite matrices, and therefore, the extremum at the fixed point is a minimum.

We now compare this with the Hessian of the global objective of GVI.

$$\nabla\nabla_{\boldsymbol{\kappa}}\mathrm{Obj}(q_s^*) = \nabla\nabla_{\boldsymbol{\kappa}} \int q(\boldsymbol{\theta}|\boldsymbol{\kappa}) \log \frac{q(\boldsymbol{\theta}|\boldsymbol{\kappa})}{\pi(\boldsymbol{\theta})\exp\{-\sum_{m=1}^{M} L_m(\mathbf{y}_m;\boldsymbol{\theta},\mathbf{x}_m)\}}\,\mu(d\boldsymbol{\theta})$$

$$= \nabla_{\boldsymbol{\kappa}} \left( \int (\nabla_{\boldsymbol{\kappa}}q(\boldsymbol{\theta}|\boldsymbol{\kappa})) \log \frac{q(\boldsymbol{\theta}|\boldsymbol{\kappa})}{\pi(\boldsymbol{\theta})\exp\{-\sum_{m=1}^{M} L_m(\mathbf{y}_m;\boldsymbol{\theta},\mathbf{x}_m)\}}\,\mu(d\boldsymbol{\theta}) \right.$$

$$\left. + \overbrace{\int (\nabla_{\boldsymbol{\kappa}} \log q(\boldsymbol{\theta}|\boldsymbol{\kappa}))q(\boldsymbol{\theta}|\boldsymbol{\kappa})\,\mu(d\boldsymbol{\theta})}^{0} \right)$$

$$= \int (\nabla\nabla_{\boldsymbol{\kappa}}q(\boldsymbol{\theta}|\boldsymbol{\kappa})) \log \frac{q(\boldsymbol{\theta}|\boldsymbol{\kappa})}{\pi(\boldsymbol{\theta})\exp\{-\sum_{m=1}^{M} L_m(\mathbf{y}_m;\boldsymbol{\theta},\mathbf{x}_m)\}}\,\mu(d\boldsymbol{\theta})$$

$$+ \overbrace{\int (\nabla_{\boldsymbol{\kappa}}q(\boldsymbol{\theta}|\boldsymbol{\kappa}))(\nabla_{\boldsymbol{\kappa}} \log q(\boldsymbol{\theta}|\boldsymbol{\kappa}))\,\mu(d\boldsymbol{\theta})}^{0}$$

Therefore, we can see that, evaluated at $\boldsymbol{\kappa} = \boldsymbol{\kappa}^*$,

$$\nabla\nabla_{\boldsymbol{\kappa}}\mathrm{Obj}(q_s^*)\Big|_{\boldsymbol{\kappa}=\boldsymbol{\kappa}^*} = \int (\nabla\nabla_{\boldsymbol{\kappa}}q(\boldsymbol{\theta}|\boldsymbol{\kappa})) \log \frac{q(\boldsymbol{\theta}|\boldsymbol{\kappa})}{\pi(\boldsymbol{\theta})\exp\{-\sum_{m=1}^{M} L_m(\mathbf{y}_m;\boldsymbol{\theta},\mathbf{x}_m)\}}\,\mu(d\boldsymbol{\theta})\Big|_{\boldsymbol{\kappa}=\boldsymbol{\kappa}^*}$$

$$= \int (\nabla\nabla_{\boldsymbol{\kappa}}q(\boldsymbol{\theta}|\boldsymbol{\kappa})) \log \frac{q_s^*(\boldsymbol{\theta}|\boldsymbol{\kappa}^*)}{\pi(\boldsymbol{\theta})\exp\{-\sum_{m=1}^{M} L_m(\mathbf{y}_m;\boldsymbol{\theta},\mathbf{x}_m)\}}\,\mu(d\boldsymbol{\theta})\Big|_{\boldsymbol{\kappa}=\boldsymbol{\kappa}^*}$$

$$= \sum_{m=1}^{M} \nabla\nabla_{\boldsymbol{\kappa}}\mathrm{Obj}(m, q_s^*)\Big|_{\boldsymbol{\kappa}=\boldsymbol{\kappa}^*}$$

Hence, the Hessian of the global GVI objective is positive definite and therefore we have found a local minimum at $q_s^*(\boldsymbol{\theta}|\boldsymbol{\kappa}^*)$ through FEDGVI. ∎

### B.4. Proof of Lemma 4.6

By combining Remark 4.1 and Proposition 4.4, we can show that, under infinite computational resources, specifically if we are able to optimise over the entire space of possible distribution parametrised by $\boldsymbol{\theta} \in \Theta$, then we are able to recover the Generalised Bayesian Posterior of Bissiri et al. (2016) in a distributed fashion by partitioning the input data and solving several smaller optimisation problems in parallel. This is achieved by using the weighted Kullback–Leibler divergence at the clients and the regular KL divergence at the server.

Under the assumption that the prior is not misspecified, we can perform distributed Bayesian updating with our framework, similar to the Bayesian Committee Machine (Tresp, 2000) where we combine local posterior distributions. We aim to recover the Generalised Bayesian Posterior (Bissiri et al., 2016):

$$q_{GBI}(\boldsymbol{\theta}|\boldsymbol{\kappa}) = \frac{\exp\{-\beta L(\mathbf{y};\boldsymbol{\theta},\mathbf{x})\}\,\pi(\boldsymbol{\theta})}{\int_\Theta \exp\{-\beta L(\mathbf{y};\boldsymbol{\theta},\mathbf{x})\}\,\pi(\boldsymbol{\theta})\mu(d\boldsymbol{\theta})}$$

where $\beta$ is some parameter that controls the learning rate from the data.

We will show that using $w = \beta$ at the clients will recover this GBI posterior after a single iteration of our algorithm, and further that the algorithm shows convergence for any subsequent iteration. We assume that $\mathcal{Q} = \mathcal{P}(\Theta)$ and that $q_{GBI}(\boldsymbol{\theta}|\mathbf{y},\mathbf{x}) \in \mathcal{Q}$. Furthermore, for simplicity we assume that the loss function $L(\cdot)$ is the additive across clients and that the data set is partitioned such that there are no intersections.

**Proof** The $M$ clients have data sets $\{\mathbf{x}_m, \mathbf{y}_m\}_{m=1}^M$ such that $\mathbf{x}_k \cap \mathbf{x}_j = \emptyset$ for all $k \neq j$ and we write $\cup_{m=1}^M \mathbf{x}_m = \mathbf{x}_1^M$ and $\cup_{m=1}^M \mathbf{y}_m = \mathbf{y}_1^M$ to symbolise the entire data set.

Then we can rewrite the GBI posterior as:

$$q_{GBI}(\boldsymbol{\theta}|\mathbf{y}_1^M, \mathbf{x}_1^M) = \frac{\exp\{-\beta \sum_{m=1}^M L(\mathbf{y}_m;\boldsymbol{\theta},\mathbf{x}_m)\}\,\pi(\boldsymbol{\theta})}{\int_\Theta \exp\{-\beta \sum_{m=1}^M L(\mathbf{y}_m;\boldsymbol{\theta},\mathbf{x}_m)\}\,\pi(\boldsymbol{\theta})\mu(d\boldsymbol{\theta})}$$

The FEDGVI approximation then takes the following form: $q_s^{(0)}(\boldsymbol{\theta}) = \prod_{m=1}^M \exp\{-\ell_m^{(0)}(\boldsymbol{\theta})\}\pi(\boldsymbol{\theta})/Z_{q_s}$ and as we initiate $\ell_m^{(0)}(\boldsymbol{\theta}) = 0$ we have that $q_s^{(0)}(\boldsymbol{\theta}) = \pi(\boldsymbol{\theta})$.

Then in parallel, the each client $m \in [M]$ carries out their optimisation step:

The cavity distribution can be found through division as:

$$q^{\backslash m}(\boldsymbol{\theta}) \propto \frac{q_s^{(0)}(\boldsymbol{\theta})}{\exp\{-\ell_m^{(0)}(\boldsymbol{\theta})\}} = \frac{\pi(\boldsymbol{\theta})}{1} = \pi(\boldsymbol{\theta})$$

And the Generalised Variational Inference step with the cavity distribution as a local prior solves the following optimisation problem:

$$
\begin{aligned}
q_m^{(1)}(\boldsymbol{\theta}) &= \arg\min_{q \in \mathcal{Q}} \left\{ \mathbb{E}_{q(\boldsymbol{\theta})}[L(\mathbf{y}_m;\boldsymbol{\theta},\mathbf{x}_m)] + \frac{1}{\beta}D_{KL}(q:\pi) \right\} \\
&\overset{(1)}{=} \arg\min_{q \in \mathcal{Q}} \mathbb{E}_{q(\boldsymbol{\theta})}\left[ \log\frac{q(\boldsymbol{\theta})}{\pi(\boldsymbol{\theta})\exp\{-\beta L(\mathbf{y}_m;\boldsymbol{\theta},\mathbf{x}_m)\}} \right] \\
&\overset{(2)}{=} \pi(\boldsymbol{\theta})\exp\{-\beta L(\mathbf{y}_m;\boldsymbol{\theta},\mathbf{x}_m)\}/Z_{q_m}
\end{aligned}
$$

Where (1) follows through the equivalence between the weighted KL divergence and the tempered loss as discussed in Appendix B.1, and (2) follows due to the properties of a statistical divergence which is minimised when the inside of the expectation is zero and since $\mathcal{Q} = \mathcal{P}(\Theta)$.

This then implies that the update we send to the server is of the form:

$$\Delta_m^{(1)}(\boldsymbol{\theta}) = -\log \frac{q_m^{(1)}(\boldsymbol{\theta})}{q_s^{(0)}(\boldsymbol{\theta})} = -\log \frac{\pi(\boldsymbol{\theta}) \exp\{-\beta L(\mathbf{y}_m; \boldsymbol{\theta}, \mathbf{x}_m)\}/Z_{q_m}}{\pi(\boldsymbol{\theta})}$$

$$= \beta L(\mathbf{y}_m; \boldsymbol{\theta}, \mathbf{x}_m) + \log Z_{q_m}$$

At the server, we can combine these such that we get:

$$\ell_s^{(1)}(\boldsymbol{\theta}) = \sum_{m=1}^{M} \beta L(\mathbf{y}_m; \boldsymbol{\theta}, \mathbf{x}_m) + \sum_{m=1}^{M} Z_{q_m} + \overbrace{\ell_s^{(0)}(\boldsymbol{\theta})}^{=0} = \beta L(\mathbf{y}_1^M; \boldsymbol{\theta}, \mathbf{x}_1^M) + \sum_{m=1}^{M} Z_{q_m}$$

As GBI depends on the prior and hence trusts it, we use the KL divergence at the server, which is optimal with respect to the GBI posterior (Zellner, 1988; Knoblauch et al., 2022). Thus, the GVI objective at the server becomes:

$$q_s^{(1)}(\boldsymbol{\theta}) = \underset{q \in \mathcal{Q}}{\arg\min} \left\{ \mathbb{E}_{q(\boldsymbol{\theta})} \left[ \ell_s^{(1)}(\boldsymbol{\theta}) \right] + D_{KL}(q : \pi) \right\}$$

$$= \underset{q \in \mathcal{Q}}{\arg\min} \left\{ \mathbb{E}_{q(\boldsymbol{\theta})} \left[ \beta L(\mathbf{y}_1^M; \boldsymbol{\theta}, \mathbf{x}_1^M) + \sum_{m=1}^{M} Z_{q_m} \right] + D_{KL}(q : \pi) \right\}$$

$$\overset{(3)}{=} \underset{q \in \mathcal{Q}}{\arg\min} \left\{ \mathbb{E}_{q(\boldsymbol{\theta})} \left[ \beta L(\mathbf{y}_1^M; \boldsymbol{\theta}, \mathbf{x}_1^M) \right] + \sum_{m=1}^{M} \cancel{Z_{q_m}}^{0} + D_{KL}(q : \pi) \right\}$$

$$= \underset{q \in \mathcal{Q}}{\arg\min} \, \mathbb{E}_{q(\boldsymbol{\theta})} \left[ \log \frac{q(\boldsymbol{\theta})}{\pi(\boldsymbol{\theta}) \exp\{-\beta L(\mathbf{y}_1^M; \boldsymbol{\theta}, \mathbf{x}_1^M)\}} \right]$$

$$\overset{(4)}{=} \pi(\boldsymbol{\theta}) \exp\{-\beta L(\mathbf{y}_1^M; \boldsymbol{\theta}, \mathbf{x}_1^M)\}/Z_{q_s^{(1)}}$$

(3) follows since $Z_{q_m}$ does not depend on $\boldsymbol{\theta}$, nor the variational parameters, and hence does not affect our optimisation problem. Line (4) is a result of $\mathcal{Q} = \mathcal{P}(\Theta)$ and the assumption that the GBI posterior is contained within this set.

This implies that the posterior that we find at the server is the Generalised Bayesian Inference posterior.

$$q_s^{(1)}(\boldsymbol{\theta}) = \pi(\boldsymbol{\theta}) \exp\{-\beta L(\mathbf{y}_1^M; \boldsymbol{\theta}, \mathbf{x}_1^M)\}/Z_{q_s^{(1)}}$$

Thereby, we have shown that FEDGVI recovers the GBI posterior under the assumptions and that this occurs after the first iteration. It remains to be shown that any further iteration steps will not change the posterior, and hence that we have recovered a fixed point as defined in Proposition 4.4.

We repeat the client optimisation steps in parallel. We first find the cavity distribution:

$$q^{\backslash m}(\boldsymbol{\theta}) \propto \frac{q_s^{(1)}(\boldsymbol{\theta})}{\exp\{-\beta L(\mathbf{y}_m; \boldsymbol{\theta}, \mathbf{x}_m)\}} \propto \frac{\pi(\boldsymbol{\theta}) \exp\{-\beta \sum_{k=1}^{M} L(\mathbf{y}_k; \boldsymbol{\theta}, \mathbf{x}_k)\}}{\exp\{-\beta L(\mathbf{y}_m; \boldsymbol{\theta}, \mathbf{x}_m)\}} = \pi(\boldsymbol{\theta}) \exp\{-\beta \sum_{k \neq m} L(\mathbf{y}_k; \boldsymbol{\theta}, \mathbf{x}_k)\}$$

Note that we ignore the normalising constant, since, similar to the server side optimisation step before, it does not depend on the variational parameters nor $\boldsymbol{\theta}$.

The optimisation step is then given through:

$$q_m^{(2)}(\boldsymbol{\theta}) = \underset{q \in \mathcal{Q}}{\arg\min} \left\{ \mathbb{E}_{q(\boldsymbol{\theta})} \left[ L(\mathbf{y}_m; \boldsymbol{\theta}, \mathbf{x}_m) \right] + \frac{1}{\beta} D_{KL}(q : q^{\backslash m}) \right\}$$

$$= \underset{q \in \mathcal{Q}}{\arg\min} \, \mathbb{E}_{q(\boldsymbol{\theta})} \left[ \log \frac{q(\boldsymbol{\theta})}{q^{\backslash m}(\boldsymbol{\theta}) \exp\{-\beta L(\mathbf{y}_m; \boldsymbol{\theta}, \mathbf{x}_m)\}} \right]$$

This statistical divergence is minimised at:

$$q_m^{(2)}(\boldsymbol{\theta}) = q^{\backslash m}(\boldsymbol{\theta}) \exp\{-\beta L(\mathbf{y}_m; \boldsymbol{\theta}, \mathbf{x}_m)\}/\tilde{Z}$$

$$= \pi(\boldsymbol{\theta}) \exp\{-\beta \sum_{k \neq m} L(\mathbf{y}_k; \boldsymbol{\theta}, \mathbf{x}_k)\} \exp\{-\beta L(\mathbf{y}_m; \boldsymbol{\theta}, \mathbf{x}_m)\}/Z_{q_m^{(2)}}$$

$$= \pi(\boldsymbol{\theta}) \exp\{-\beta \sum_{m=1}^{M} L(\mathbf{y}_m; \boldsymbol{\theta}, \mathbf{x}_m)\}/Z_{q_m^{(2)}}$$

where we note that $Z_{q_m^{(2)}} = Z_{q_s^{(1)}}$ and we have recovered the GBI posterior we currently have as our server distribution. As a result, $\Delta_m^{(2)}(\boldsymbol{\theta}) = -(\log q_m^{(2)}(\boldsymbol{\theta}) - \log q_s^{(1)}(\boldsymbol{\theta})) = -\log 1 = 0$ for all $m \in [M]$.

This satisfies the conditions for Proposition 4.4 and hence we have achieved a fixed point, which will not change the server distribution, since:

$$\ell_s^{(2)}(\boldsymbol{\theta}) = \sum_{m=1}^{M} \overbrace{\Delta_m^{(2)}(\boldsymbol{\theta})}^{=0} + \ell_s^{(1)}(\boldsymbol{\theta}) = \ell_s^{(1)}(\boldsymbol{\theta}) = \beta L(\mathbf{y}_1^M; \boldsymbol{\theta}, \mathbf{x}_1^M)$$

which means that the server optimisation routine would not be different from the one during the previous iteration.

$$q_s^{(2)}(\boldsymbol{\theta}) = \underset{q \in \mathcal{Q}}{\arg\min} \left\{ \mathbb{E}_{q(\boldsymbol{\theta})} \left[ \ell_s^{(2)}(\boldsymbol{\theta}) \right] + D_{KL}(q : \pi) \right\} = \underset{q \in \mathcal{Q}}{\arg\min} \left\{ \mathbb{E}_{q(\boldsymbol{\theta})} \left[ \ell_s^{(1)}(\boldsymbol{\theta}) \right] + D_{KL}(q : \pi) \right\} = q_s^{(1)}(\boldsymbol{\theta})$$

And thus $q_s^{(2)}(\boldsymbol{\theta}) = q_s^{(1)}(\boldsymbol{\theta}) = q_{GBI}^*(\boldsymbol{\theta}|\mathbf{y}_1^M, \mathbf{x}_1^M)$.

For the moreover part, we define the damping parameter $\delta = \frac{1}{M}$, and show that $q_s^{(t)}(\boldsymbol{\theta}) \to q_{GBI}(\boldsymbol{\theta}|\mathbf{y}_1^M, \mathbf{x}_1^M)$ as $t \to \infty$. As the data here is implicit, we simplify notation by denoting the losses of a client as $L_m(\boldsymbol{\theta})$ and the GBI posterior as $q_{GBI}(\boldsymbol{\theta})$. Furthermore, we assume that the GBI learning rate parameter $\beta$ is implicitly included in each client's loss. Then by the usual modes of convergence, we show that:

$$\left| q_s^{(t)}(\boldsymbol{\theta}) - q_{GBI}(\boldsymbol{\theta}) \right| \to 0$$

Note that under KL divergences at the server and client, we will have that $\ell_s^{(t)}(\boldsymbol{\theta}) = \sum_{m=1}^{M} \ell_m^{(t)}(\boldsymbol{\theta})$ (see proof of Remark 4.1).

$$\left| \pi(\boldsymbol{\theta}) \exp \left\{ -\sum_{m}^{M} \ell_m^{(t)}(\boldsymbol{\theta}) \right\} - \pi(\boldsymbol{\theta}) \exp \left\{ -\sum_{m}^{M} L_m(\boldsymbol{\theta}) \right\} \right| = \pi(\boldsymbol{\theta}) \left| \exp \left\{ -\sum_{m}^{M} \ell_m^{(t)}(\boldsymbol{\theta}) \right\} - \exp \left\{ -\sum_{m}^{M} L_m(\boldsymbol{\theta}) \right\} \right|$$

This converges when the exponents are equal, hence it is sufficient to prove that $\forall m \in [M]$ we have $\ell_m^{(t)}(\boldsymbol{\theta}) \to L_m(\boldsymbol{\theta})$.

Since for all $m \in [M]$, at each iteration $t$ we have that under the KL divergences:

$$q^{\backslash m}(\boldsymbol{\theta}) \propto \frac{q_s^{(t-1)}(\boldsymbol{\theta})}{\exp\{-\ell_m^{(t-1)}(\boldsymbol{\theta})\}} = \frac{\pi(\boldsymbol{\theta}) \exp\{-\sum_{m=1}^{M} \ell_m^{(t-1)}(\boldsymbol{\theta})\}}{\exp\{-\ell_m^{(t-1)}(\boldsymbol{\theta})\}} = \pi(\boldsymbol{\theta}) \exp \left\{ -\sum_{k \neq m} \ell_k^{(t-1)}(\boldsymbol{\theta}) \right\}$$

$$q_m^{(t)}(\boldsymbol{\theta}) \propto \exp\{L_m(\boldsymbol{\theta})\} q^{\backslash m}(\boldsymbol{\theta})$$

$$\Delta_m^{(t)}(\boldsymbol{\theta}) = -\frac{1}{M} \log \frac{\exp\{L_m(\boldsymbol{\theta})\} \pi(\boldsymbol{\theta}) \exp \left\{ -\sum_{k \neq m} \ell_k^{(t-1)}(\boldsymbol{\theta}) \right\}}{\pi(\boldsymbol{\theta}) \exp\{-\sum_{m=1}^{M} \ell_m^{(t-1)}(\boldsymbol{\theta})\}} = \frac{1}{M} L_m(\boldsymbol{\theta}) - \frac{1}{M} \ell_m^{(t-1)}(\boldsymbol{\theta})$$

$$\ell_m^{(t)}(\boldsymbol{\theta}) = \ell_m^{(t-1)}(\boldsymbol{\theta}) + \Delta_m^{(t)}(\boldsymbol{\theta}) = \frac{1}{M} L_m(\boldsymbol{\theta}) + \frac{M-1}{M} \ell_m^{(t-1)}(\boldsymbol{\theta})$$

By expansion of $\ell_m^{(t-1)}(\boldsymbol{\theta})$, by recursively applying the definition above, we get the following closed form expression:

$$\ell_m^{(t-1)}(\boldsymbol{\theta}) = \left( \left( \frac{M-1}{M} \right) \frac{1}{M} + \left( \frac{M-1}{M} \right) \left( \frac{M-1}{M} \right) \frac{1}{M} + ... + \left( \frac{M-1}{M} \right)^t \frac{1}{M} \ell_m^{(0)}(\boldsymbol{\theta}) \right) L_m(\boldsymbol{\theta})$$

written as a summation and recalling that $\ell_m^{(0)}(\boldsymbol{\theta}) = 0$ by definition, we can interpret this as the series:

$$\ell_m^{(t)}(\boldsymbol{\theta}) = L_m(\boldsymbol{\theta}) \sum_{i=0}^{t-1} \frac{1}{M} \left( \frac{M-1}{M} \right)^t$$

which is a geometric series. And since $\frac{M-1}{M} \in (0, 1)$ by elementary analysis this converges, as $t \to \infty$, to the limit

$$\lim_{t \to \infty} \ell_m^{(t)}(\boldsymbol{\theta}) = L_m(\boldsymbol{\theta}) \frac{1}{M} M = L_m(\boldsymbol{\theta}).$$

Therefore, as $t \to \infty$ $q_s^{(t)}(\boldsymbol{\theta}) \to q_{GBI}(\boldsymbol{\theta})$ $\boldsymbol{\theta}$ almost everywhere. We can only guarantee almost everywhere pointwise convergence, since integral operators such as the KL divergence only guarantee equivalence up to null sets. ∎

Notably, the reason for using the cavity distribution instead of some other effective prior for the client optimisation step is that we want to recover the (generalised) Bayesian posterior eventually with our framework assuming that we can optimise over the entire space of probability measures that characterise their respective probability distributions. We further assume that we can find a global minimiser of any optimisation problem. Then, under these assumptions, we would like to not change the current posterior any further after recovering the GBI posterior.

We have previously shown that our algorithm achieves just this, and we can furthermore show that the cavity distribution is indeed the only choice in the client update that causes this.

## B.5. Proof of Theorem 4.9

We are interested in verifying whether the cavity distribution is necessary in Equation (4). It acts to regularise the optimisation problem at the client, which we restate here, using some arbitrary probability density $\rho \in \mathcal{P}(\Theta)$:

$$q_m^{(t)}(\boldsymbol{\theta}) = \underset{q \in \mathcal{Q}}{\arg\min} \left\{ \mathbb{E}_{q(\boldsymbol{\theta})} \left[ L_m^{(t)}(\mathbf{y}_m; \boldsymbol{\theta}, \mathbf{x}_m) \right] + D(q : \rho) \right\}$$

where it is regularised by $D(\cdot : \rho)$. It is clear that this should not be the prior distribution after the server has additional information about client data available since we would not be doing anything different for subsequent updates and this would result in a Bayesian Committee Machine where each client does not learn from the others. Therefore it is imperative to ask what this 'effective prior' $\rho$ should be? And in fact it turns out that it needs to be the cavity distribution.

We will approach this problem by considering the case where we know what we would want to target in the optimization problem and hence the sequence $\{q_s^{(t)}(\boldsymbol{\theta})\}_{t \in \mathbb{N}_0}$ should converge to. We, however, have to restrict ourselves to the Federated Learning scenario and therefore any distribution that we come up with needs to satisfy the Assumptions 4.7 and 4.8. For this we require the following assumption so that we are able to target the GBI posterior.

**Assumption B.2.** We are able to find global minimisers over the entire space of probability distributions parametrised by $\boldsymbol{\theta}$, $\mathcal{P}(\Theta)$.

Then it turns out that this regularising distribution is uniquely described by Theorem 4.9, which we restate here.

**Theorem 4.9** *Let the assumptions be as in Lemma 4.6, i.e. $\mathcal{Q} = \mathcal{P}(\Theta)$, $D = \frac{1}{\beta} D_{KL}$ for $\beta > 0$, $D_s = D_{KL}$, $L_m^{(t)} = L$, and $\tau_m = 1$, and further assume that Assumptions 4.7 and 4.8 are satisfied, then the following are equivalent:*

1. *$\exists t \in [T]$ for which $q_s^{(t)}(\boldsymbol{\theta}) = q_{\mathrm{GBI}}(\boldsymbol{\theta})$ (a.e.) is invariant under further FEDGVI updates.*

2. *The cavity distribution regularises the client optimisation problem.*

**Proof** $(2 \implies 1)$ This is a direct consequence of Lemma 4.6 and can easily be seen by iterating through the algorithm with the cavity distribution.

$(1 \implies 2)$ Without loss of generality we consider the GBI posterior to be found after the first iteration. We show that the unique way that satisfies the axioms and does not change the GBI posterior at the second iteration (or any further iterations) is uniquely achieved by the cavity distribution. By the statement we have

$$q_s^{(2)}(\boldsymbol{\theta}) = \exp\{-\ell_s^{(2)}(\boldsymbol{\theta})\} \pi(\boldsymbol{\theta})/Z_s^{(2)} = \exp\{-\ell_s^{(1)}(\boldsymbol{\theta})\} \pi(\boldsymbol{\theta})/Z_s^{(1)} = q_s^{(1)}(\boldsymbol{\theta}).$$

We now need to relate this to the client updates and hence the solutions of the client optimization problem.

$$
\begin{aligned}
q_s^{(2)}(\boldsymbol{\theta}) = q_s^{(1)}(\boldsymbol{\theta}) &\iff \exp\{-\ell_s^{(2)}(\boldsymbol{\theta})\}/Z_s^{(2)} = \exp\{-\ell_s^{(1)}(\boldsymbol{\theta})\}/Z_s^{(1)} \\
&\iff \ell_s^{(2)}(\boldsymbol{\theta}) + \log Z_s^{(2)} = \ell_s^{(1)}(\boldsymbol{\theta}) + \log Z_s^{(1)} \\
&\iff \ell_s^{(2)}(\boldsymbol{\theta}) = \ell_s^{(1)}(\boldsymbol{\theta}) + C, \quad C \in \mathbb{R} \\
&\stackrel{\text{def}}{\iff} \sum_{m=1}^{M} \Delta_m^{(2)}(\boldsymbol{\theta}) + \ell_s^{(1)}(\boldsymbol{\theta}) = \ell_s^{(1)}(\boldsymbol{\theta}) + C \\
&\iff \sum_{m=1}^{M} \Delta_m^{(2)}(\boldsymbol{\theta}) = C \\
&\iff \sum_{m=1}^{M} \log \frac{q_m^{(2)}(\boldsymbol{\theta})}{q_s^{(1)}(\boldsymbol{\theta})} = C \\
&\iff \prod_{m=1}^{M} q_m^{(2)}(\boldsymbol{\theta}) = K \left( q_s^{(1)}(\boldsymbol{\theta}) \right)^M, \quad K = e^C
\end{aligned}
\tag{10}
$$

Now, for some transformation operator $\xi_m : \mathcal{P}(\Theta) \to \mathcal{P}(\Theta)$ acting on the information available at the client from the server in the form of the current approximate posterior, which we denote as $\xi_m[q_s^{(1)}](\boldsymbol{\theta})$, that satisfies the Assumptions 4.7 and 4.8, we get the client optimisation problem $\forall m$:

$$
\begin{aligned}
q_m^{(2)}(\boldsymbol{\theta}) &= \arg\min_{q \in \mathcal{P}(\Theta)} \left\{ \mathbb{E}_{q(\boldsymbol{\theta})} \left[ L(\mathbf{y}_m; \boldsymbol{\theta}, \mathbf{x}_m) \right] + \frac{1}{\beta} D_{KL}(q : \xi_m[q_s^{(1)}]) \right\} \\
&= \arg\min_{q \in \mathcal{P}(\Theta)} \left\{ \frac{1}{\beta} \mathbb{E}_{q(\boldsymbol{\theta})} \left[ -\beta \log \exp\{-L(\mathbf{y}_m; \boldsymbol{\theta}, \mathbf{x}_m)\} \right] + \frac{1}{\beta} \mathbb{E}_{q(\boldsymbol{\theta})} \left[ \log \frac{q(\boldsymbol{\theta})}{\xi_m[q_s^{(1)}](\boldsymbol{\theta})} \right] \right\} \\
&= \arg\min_{q \in \mathcal{P}(\Theta)} \left\{ \frac{1}{\beta} \mathbb{E}_{q(\boldsymbol{\theta})} \left[ \log \frac{q(\boldsymbol{\theta})}{\exp\{-\beta L(\mathbf{y}_m; \boldsymbol{\theta}, \mathbf{x}_m)\} \xi_m[q_s^{(1)}](\boldsymbol{\theta})} \right] \right\} \\
\implies q_m^{(2)}(\boldsymbol{\theta}) &= \exp\{-\beta L(\mathbf{y}_m; \boldsymbol{\theta}, \mathbf{x}_m)\} \xi_m[q_s^{(1)}](\boldsymbol{\theta}) / Z_m^{(2)}
\end{aligned}
$$

Substituting this into Equation (10) and using the definition of $q_s^{(1)}(\boldsymbol{\theta})$ we can derive a relation between the individual client approximations.

$$
\prod_{m=1}^{M} q_m^{(2)}(\boldsymbol{\theta}) = K \left( q_s^{(1)}(\boldsymbol{\theta}) \right)^M
$$

$$
\prod_{m=1}^{M} \frac{\xi_m[q_s^{(1)}](\boldsymbol{\theta}) \exp\{-\beta L(\mathbf{y}_m; \boldsymbol{\theta}, \mathbf{x}_m)\}}{Z_m^{(2)}} = K(\pi(\boldsymbol{\theta}))^M \exp \left\{ -M\beta \sum_{m=1}^{M} L(\mathbf{y}_m; \boldsymbol{\theta}, \mathbf{x}_m) \right\} / \left( Z_s^{(1)} \right)^M
$$

$$
\prod_{m=1}^{M} \xi_m[q_s^{(1)}](\boldsymbol{\theta}) / Z_m^{(2)} = K(\pi(\boldsymbol{\theta}))^M \exp \left\{ -(M-1)\beta \sum_{m=1}^{M} L(\mathbf{y}_m; \boldsymbol{\theta}, \mathbf{x}_m) \right\} / \left( Z_s^{(1)} \right)^M
$$

$$
\implies \prod_{m=1}^{M} \xi_m[q_s^{(1)}](\boldsymbol{\theta}) \propto \prod_{m=1}^{M} \pi(\boldsymbol{\theta}) \exp \left\{ -\beta \sum_{k \neq m} L(\mathbf{y}_k; \boldsymbol{\theta}, \mathbf{x}_k) \right\} \propto \prod_{m=1}^{M} \frac{q_s^{(1)}(\boldsymbol{\theta})}{\exp\{-\beta L(\mathbf{y}_m; \boldsymbol{\theta}, \mathbf{x}_m)\}}
$$

Here, proportional '$\propto$' means equivalent up to some constant independent of $\boldsymbol{\theta}$. To see that the cavity distribution is in fact the only choice that satisfies the above equation, we need to recall the two axioms: (Assumption 4.8) $\xi_m[q_s^{(1)}](\boldsymbol{\theta})$ needs to be generated in the same way across clients, and (Assumption 4.7) since we are in federated learning, each client will only be able to access it's own data. This implies that we can write $\xi_m[q_s^{(1)}](\boldsymbol{\theta})$ as a function of the current approximation and

the client data, $\xi_m[q_s^{(1)}](\boldsymbol{\theta}) = \xi[q_s^{(1)}, \mathbf{y}_m, \mathbf{x}_m](\boldsymbol{\theta})$.

$$\prod_{m=1}^{M} \xi[q_s^{(1)}, \mathbf{y}_m, \mathbf{x}_m](\boldsymbol{\theta}) \propto \prod_{m=1}^{M} \frac{q_s^{(1)}(\boldsymbol{\theta})}{\exp\{-\beta L(\mathbf{y}_m; \boldsymbol{\theta}, \mathbf{x}_m)\}}$$

The only client that would have access to an explicit expression for the denominator would be client $m$, to which the data $\{\mathbf{x}_m, \mathbf{y}_m\}$ belongs, and hence it must be entirely contained within that client's regularisation term $\xi_m$. Therefore, we can conclude that $q_m^{(2)}(\boldsymbol{\theta}) = q_s^{(1)}(\boldsymbol{\theta})$ and find a closed form for $\xi_m[q_s^{(1)}](\boldsymbol{\theta})$. Note that this implies $C = 0$ and hence $K = 1$.

$$\exp\{-\beta L(\mathbf{y}_m; \boldsymbol{\theta}, \mathbf{x}_m)\}\xi_m[q_s^{(1)}](\boldsymbol{\theta})/Z_{q_m^{(2)}} = \exp\{-\sum_{m=1}^{M} \beta L(\mathbf{y}_m; \boldsymbol{\theta}, \mathbf{x}_m)\}\pi(\boldsymbol{\theta})/Z_{q_s^{(1)}}$$

$$\xi_m[q_s^{(1)}](\boldsymbol{\theta}) \propto \frac{\exp\{-\sum_{m=1}^{M} \beta L(\mathbf{y}_m; \boldsymbol{\theta}, \mathbf{x}_m)\}\pi(\boldsymbol{\theta})/Z_{q_s^{(1)}}}{\exp\{-\beta L(\mathbf{y}_m; \boldsymbol{\theta}, \mathbf{x}_m)\}/Z_{q_m^{(2)}}}$$

This is exactly the cavity distribution as described in Equation (3). ∎

This gives a justification for using the cavity distribution in our algorithm, since under the assumption that the prior is well specified, we would like to converge to the generalised Bayesian posterior distribution. Furthermore, we can note that this single step of FEDGVI recovers the principle of the Bayesian Committee Machine (BCM) of Tresp (2000) where we use generalised loss functions instead of the negative log likelihood in our formulation. Furthermore, a single pass through FEDGVI—with the divergences as described above— will recover a generalised version of the BCM irregardless of the space we optimise over.

*Remark* B.3. For the last two proofs we have assumed that we can find the global minimisers of the equations. This isn't strictly necessary to have since the use of the (weighted) Kullback–Leibler divergence allows us to formulate a closed form expression for what these will look like.

### B.6. Proof of Proposition 4.10

This proposition is a direct result of Proposition 3.1 in Altamirano et al. (2023) and the proof is analogous, we merely include it here for completeness. And while the stated result is in a regression setting, it can be be extended to the classification setting similar to Altamirano et al. (2024) where Gaussian Processes are considered.

We assume that each client has a data set $\{\mathbf{x}_i\}_{i=1}^{n_m}$ of size $n_m$. The divergence operator $\nabla \cdot f(\mathbf{x})$ is defined in the usual way as the inner product between the vector of partial derivative operators and the vector of some vector valued function $f(\mathbf{x})$ as $\nabla \cdot f(\mathbf{x}) = \langle (\partial/\partial x_1, ..., \partial/\partial x_d)^\top, (f_1(\mathbf{x}), ..., f_d(\mathbf{x}))^\top \rangle$, and $\nabla_{\mathbf{x}} g(\mathbf{x})$ is the Jacobian, the vector of partial derivatives of $g(\mathbf{x})$. We further assume that $\Xi \subseteq \mathbb{R}^d$, and that $p_{\boldsymbol{\theta}}(\mathbf{x}) \in \mathcal{P}(\Theta)$.

**Proof** The loss of some client $m \in [M]$ at some arbitrary iteration $t \in [T]$ is given by

$$\hat{D}(\boldsymbol{\theta}, \mathbb{P}_{n_m}) := \frac{1}{n_m} \sum_{i=1}^{n_m} \underbrace{||w_m^{(t)^\top} \nabla_{\mathbf{x}} \log p_{\boldsymbol{\theta}}(\mathbf{x}_i)||_2^2}_{(1)} + 2 \underbrace{\nabla \cdot (w_m^{(t)} w_m^{(t)^\top} \nabla_{\mathbf{x}} \log p_{\boldsymbol{\theta}}(\mathbf{x}_i))}_{(2)}$$

where $\nabla_{\mathbf{x}} \log p_{\boldsymbol{\theta}}(\mathbf{x}_i) = \nabla_{\mathbf{x}} \boldsymbol{\eta}(\boldsymbol{\theta})^\top \phi(\mathbf{x}_i) + \nabla_{\mathbf{x}} h(\mathbf{x}_i)$. We can then expand the terms in the above terms which we then give equal up to an additive constant independent of $\boldsymbol{\theta}$.

$$(1) = (w_m^{(t)^\top} (\nabla_{\mathbf{x}} \phi(\mathbf{x}_i)^\top \boldsymbol{\eta}(\boldsymbol{\theta}) + \nabla_{\mathbf{x}} h(\mathbf{x}_i)))^\top (w_m^{(t)^\top} (\nabla_{\mathbf{x}} \phi(\mathbf{x}_i)^\top \boldsymbol{\eta}(\boldsymbol{\theta}) + \nabla_{\mathbf{x}} h(\mathbf{x}_i)))$$

$$= (w_m^{(t)^\top} \nabla_{\mathbf{x}} \phi(\mathbf{x}_i)^\top \boldsymbol{\eta}(\boldsymbol{\theta}))^\top (w_m^{(t)^\top} \nabla_{\mathbf{x}} \phi(\mathbf{x}_i)^\top \boldsymbol{\eta}(\boldsymbol{\theta})) + (w_m^{(t)^\top} \nabla_{\mathbf{x}} h(\mathbf{x}_i))^\top (w_m^{(t)^\top} \nabla_{\mathbf{x}} h(\mathbf{x}_i))$$

$$+ 2(w_m^{(t)^\top} \nabla_{\mathbf{x}} \phi(\mathbf{x}_i)^\top \boldsymbol{\eta}(\boldsymbol{\theta}))^\top ((w_m^{(t)^\top} \nabla_{\mathbf{x}} h(\mathbf{x}_i)))$$

$$\overset{\pm c}{=} \boldsymbol{\eta}(\boldsymbol{\theta})^\top \nabla_{\mathbf{x}} \phi(\mathbf{x}_i) w_m^{(t)} w_m^{(t)^\top} \nabla_{\mathbf{x}} \phi(\mathbf{x}_i)^\top \boldsymbol{\eta}(\boldsymbol{\theta}) + \boldsymbol{\eta}(\boldsymbol{\theta})^\top \nabla_{\mathbf{x}} \phi(\mathbf{x}_i) w_m^{(t)} w_m^{(t)^\top} \nabla_{\mathbf{x}} h(\mathbf{x}_i)$$

where the last line follows since the middle terms are independent of $\boldsymbol{\theta}$ as long as the weight function is independent of $\boldsymbol{\theta}$.

$$(2) = \nabla \cdot (w_m^{(t)} w_m^{(t)^\top} \nabla_{\mathbf{x}} \boldsymbol{\eta}(\boldsymbol{\theta})^\top \phi(\mathbf{x}_i)) + \nabla \cdot (w_m^{(t)} w_m^{(t)^\top} \nabla_{\mathbf{x}} h(\mathbf{x}_i))$$

$$\overset{\pm c}{=} \boldsymbol{\eta}(\boldsymbol{\theta})^\top (\nabla \cdot (w_m^{(t)} w_m^{(t)^\top} \nabla_{\mathbf{x}} \boldsymbol{\phi}(\mathbf{x}_i)))$$

Then, this has the form $\hat{D}(\boldsymbol{\theta}, \mathbb{P}) \overset{\pm c}{=} \boldsymbol{\eta}(\boldsymbol{\theta})^\top \Lambda_m^{(t)} \boldsymbol{\eta}(\boldsymbol{\theta}) + \boldsymbol{\eta}(\boldsymbol{\theta})^\top \boldsymbol{\nu}_m^{(t)}$, where

$$\Lambda_m^{(t)} := \frac{1}{n_m} \sum_{i=1}^{n_m} \nabla_{\mathbf{x}} \boldsymbol{\phi}(\mathbf{x}_i) w_m^{(t)} w_m^{(t)^\top} \nabla_{\mathbf{x}} \boldsymbol{\phi}(\mathbf{x}_i)^\top \qquad \text{and} \qquad \boldsymbol{\nu}_m^{(t)} := \frac{2}{n_m} \sum_{i=1}^{n_m} \nabla \cdot (w_m^{(t)} w_m^{(t)^\top} \nabla_{\mathbf{x}} \boldsymbol{\phi}(\mathbf{x}_i)).$$

The first art follows by setting $q_m^{(t)}(\boldsymbol{\theta}) \propto q_{(t)}^{\backslash m}(\boldsymbol{\theta}) \exp\{-\beta n_m (\boldsymbol{\eta}(\boldsymbol{\theta})^\top \Lambda_m^{(t)} \boldsymbol{\eta}(\boldsymbol{\theta}) + \boldsymbol{\eta}(\boldsymbol{\theta})^\top \boldsymbol{\nu}_m^{(t)})\}$. Then, if $\boldsymbol{\eta}(\boldsymbol{\theta}) = \boldsymbol{\theta}$, and the local cavity distribution has the form $q_{(t)}^{\backslash m}(\boldsymbol{\theta}) \propto \exp\{-\frac{1}{2}(\boldsymbol{\theta} - \boldsymbol{\mu}_{\backslash m}^{(t)})^\top \Sigma_{\backslash m}^{(t)^{-1}} (\boldsymbol{\theta} - \boldsymbol{\mu}_{\backslash m}^{(t)})\}$, then the local posterior is conjugate and is given by $q_m^{(t)}(\boldsymbol{\theta}) \propto \exp\{-\frac{1}{2}(\boldsymbol{\theta} - \boldsymbol{\mu}_m^{(t)})^\top \Sigma_m^{(t)^{-1}} (\boldsymbol{\theta} - \boldsymbol{\mu}_m^{(t)})\}$, where

$$\begin{aligned}
q_m^{(t)}(\boldsymbol{\theta}) &\propto q_{(t)}^{\backslash m}(\boldsymbol{\theta}) \exp\{-\beta n_m (\boldsymbol{\eta}(\boldsymbol{\theta})^\top \Lambda_m^{(t)} \boldsymbol{\eta}(\boldsymbol{\theta}) + \boldsymbol{\eta}(\boldsymbol{\theta})^\top \boldsymbol{\nu}_m^{(t)})\} \\
&\propto \exp\{-\frac{1}{2}(\boldsymbol{\theta} - \boldsymbol{\mu}_{\backslash m}^{(t)})^\top \Sigma_{\backslash m}^{(t)^{-1}} (\boldsymbol{\theta} - \boldsymbol{\mu}_{\backslash m}^{(t)})\} \exp\{-\beta n_m (\boldsymbol{\theta}^\top \Lambda_m^{(t)} \boldsymbol{\theta} + \boldsymbol{\theta}^\top \boldsymbol{\nu}_m^{(t)})\} \\
&\propto \exp\{-\frac{1}{2}[\boldsymbol{\theta}^\top \Sigma_{\backslash m}^{(t)^{-1}} \boldsymbol{\theta} - 2\boldsymbol{\theta}^\top \Sigma_{\backslash m}^{(t)^{-1}} \boldsymbol{\mu}_{\backslash m}^{(t)} + 2\beta n_m \boldsymbol{\theta}^\top \Lambda_m^{(t)} \boldsymbol{\theta} + 2\beta n_m \boldsymbol{\theta}^\top \boldsymbol{\nu}_m^{(t)}]\} \\
&\propto \exp\{-\frac{1}{2}(\boldsymbol{\theta} - \boldsymbol{\mu}_m^{(t)})^\top \Sigma_m^{(t)^{-1}} (\boldsymbol{\theta} - \boldsymbol{\mu}_m^{(t)})\}
\end{aligned}$$

where in the last line, we complete the square and get parameters

$$\Sigma_m^{(t)^{-1}} := \Sigma_{\backslash m}^{(t)^{-1}} + \beta n_m \Lambda_m^{(t)} \qquad \text{and} \qquad \boldsymbol{\mu}_m^{(t)} := \Sigma_m^{(t)} (\Sigma_{\backslash m}^{(t)^{-1}} \boldsymbol{\mu}_{\backslash m}^{(t)} - \Lambda_m^{(t)} \boldsymbol{\nu}_m^{(t)}).$$

The moreover part can now easily be seen. The update will be quadratic in $\boldsymbol{\theta}$ and hence summing these results in a quadratic function, and since the posterior will have Gaussian distribution, the expectation with respect to the posterior of this quadratic function will have closed form. Therefore, if the divergence at the server allows for closed form solutions between Multivariate Gaussians, then the entire Equation (7) will have a closed form optimisation procedure that does not require sampling to approximate integrals. ∎

Note we have implicitly used the weighted KL divergence with parameter $\beta n_m$. Note also that this does not immediately follow from Lemma 4.6 since the weighting function is allowed to change depending on the iteration and the client. In our experiments, we for instance use the weighting function as measuring some deviation of a data point to the cavity mean. Furthermore, the weighting function does depend on the data point, but we suppress this dependence here to lighten notation.

## B.7. Proof of Theorem 4.12

This result is more involved to prove where we show by induction that at each iteration, the posterior generated at the server is robust to outliers through the robustness of each client's loss function to outliers. To prove this result, we first consider what we mean by robustness and introduce some terminology. We consider the empirical data distribution of all clients $\mathbb{P}_n = \frac{1}{n} \sum_i^n \delta_{\mathbf{x}_i}$ which is perturbed by some Huber contamination with parameter $\varepsilon$ at some adversarially chosen data point $z \in \Xi$ as $\mathbb{P}_{n,\varepsilon,z} := (1 - \varepsilon)\mathbb{P}_n + \varepsilon \delta_z$, where the subscript $n$ indicates how many data points are drawn from the distribution. Note that $\mathbb{P}_n = \frac{1}{n} \sum_{m=1}^M \sum_{i=1}^{n_m} \delta_{\mathbf{x}_{mi}} = \frac{1}{n} \sum_{m=1}^M n_m \mathbb{P}_{n_m}$. We then write $q_s^{(t)}(\boldsymbol{\theta}; \mathbb{P}_{n,\varepsilon,z})$ to indicate a distribution with respect to data generated from the specified DGP. We first recall the notion of robustness introduced by Ghosh & Basu (2016a). Note that we suppress the measure $\mu(d\boldsymbol{\theta})$ in the following and simply write $d\boldsymbol{\theta}$. The posterior influence is given by:

$$\text{PIF}(z, \boldsymbol{\theta}, \mathbb{P}_n) := \lim_{\varepsilon \downarrow 0} \frac{q_s^{(t)}(\boldsymbol{\theta}; \mathbb{P}_{n,\varepsilon,z}) - q_s^{(t)}(\boldsymbol{\theta}; \mathbb{P}_n)}{\varepsilon} = \frac{d}{d\varepsilon} q_s^{(t)}(\boldsymbol{\theta}; \mathbb{P}_{n,\varepsilon,z})|_{\varepsilon=0}$$

where the last line follows by L'Hopital's rule. Ghosh & Basu (2016a) further show, and one can easily check, that for $q_s^{(t)}(\boldsymbol{\theta}; \mathbb{P}_{n,\varepsilon,z}) = \pi(\boldsymbol{\theta}) \exp\{-\ell_s^{(t)}(\boldsymbol{\theta}; \mathbb{P}_{n,\varepsilon,z})\} / \int \pi(\boldsymbol{\theta}) \exp\{-\ell_s^{(t)}(\boldsymbol{\theta}; \mathbb{P}_{n,\varepsilon,z})\} d\boldsymbol{\theta}$ this is equal to:

$$\text{PIF}(z, \boldsymbol{\theta}, \mathbb{P}_n) = q_s^{(t)}(\boldsymbol{\theta}; \mathbb{P}_n) \left( -\frac{d}{d\varepsilon} \ell_s^{(t)}(\boldsymbol{\theta}; \mathbb{P}_{n,\varepsilon,z})|_{\varepsilon=0} + \int_\Theta \frac{d}{d\varepsilon} \ell_s^{(t)}(\boldsymbol{\theta}; \mathbb{P}_{n,\varepsilon,z})|_{\varepsilon=0} \Pi(d\boldsymbol{\theta}) \right)$$

We call loss robust if it has finite posterior influence, i.e. $\sup_{\boldsymbol{\theta} \in \Theta} \sup_{z \in \Xi} |\text{PIF}(z, \boldsymbol{\theta}, \mathbb{P}_n)| < \infty$. To this end, we now state a Lemma due to Matsubara et al. (2022), adapted to our notation for FEDGVI which we have rephrased into Definition 4.11.

**Lemma B.4** ([Matsubara et al. (2022)](#)). *Let $q_{(\cdot)}^{(t)}(\boldsymbol{\theta}; \mathbb{P}_n)$ be a posterior computed at the server or the client with fixed $n \in \mathbb{N}$ with loss $\ell_{(\cdot)}^{(t)}(\boldsymbol{\theta}; \mathbb{P}_n)$ and a prior $\pi(\boldsymbol{\theta})$. Suppose that $\ell_{(\cdot)}^{(t)}(\boldsymbol{\theta}; \mathbb{P}_n)$ is lower bounded and that $\pi(\boldsymbol{\theta})$ is upper bounded over $\boldsymbol{\theta} \in \Theta$, for any $\mathbb{P}_n$. Then if there exists some function $\gamma_{(\cdot)}^{(t)} : \Theta \to \mathbb{R}$ such that*

$$1. \quad \sup_{z \in \Xi} \left| \frac{d}{d\varepsilon} \ell_{(\cdot)}^{(t)}(\boldsymbol{\theta}; \mathbb{P}_{n,\varepsilon,z}) \big|_{\varepsilon=0} \right| \le \gamma_{(\cdot)}^{(t)}(\boldsymbol{\theta}),$$

$$2. \quad \sup_{\boldsymbol{\theta} \in \Theta} \pi(\boldsymbol{\theta}) \gamma_{(\cdot)}^{(t)}(\boldsymbol{\theta}) < \infty, \text{ and}$$

$$3. \quad \int_{\Theta} \pi(\boldsymbol{\theta}) \gamma_{(\cdot)}^{(t)}(\boldsymbol{\theta}) d\boldsymbol{\theta} < \infty$$

*hold, then $q_{(\cdot)}^{(t)}(\boldsymbol{\theta}; \mathbb{P}_n)$ is globally bias–robust.*

We provide further clarification on these conditions in Appendix C.2, and we are now able to give the proof of Theorem 4.12.

**Proof** By the Lemma B.4, we need to show that

$$\sup_{z \in \Xi} \left| \frac{d}{d\varepsilon} \ell_s^{(t)}(\boldsymbol{\theta}; \mathbb{P}_{n,\varepsilon,z}) \big|_{\varepsilon=0} \right| \le \gamma_s^{(t)}(\boldsymbol{\theta})$$

and that this $\gamma_s^{(t)}(\boldsymbol{\theta})$ satisfies conditions (2.) and (3.) of the Lemma. Per assumption we know that the clients are robust to likelihood misspecification, so we need to relate the server loss to the client posterior influence functions. To this end, we consider the loss at the server.

$$\ell_s^{(t)}(\boldsymbol{\theta}; \mathbb{P}_{n,\varepsilon,z}) = \sum_{m=1}^{M} \Delta_m^{(t)}(\boldsymbol{\theta}; \mathbb{P}_{n,\varepsilon,z}) + \ell_s^{(t-1)}(\boldsymbol{\theta}; \mathbb{P}_{n,\varepsilon,z})$$

where for each client, the update is given through Equation (5)

$$\Delta_m^{(t)}(\boldsymbol{\theta}; \mathbb{P}_{n,\varepsilon,z}) = -\log \frac{q_m^{(t)}(\boldsymbol{\theta}; \mathbb{P}_{n,\varepsilon,z})}{q_s^{(t-1)}(\boldsymbol{\theta}; \mathbb{P}_{n,\varepsilon,z})} = -\log \frac{\frac{q_{(t)}^{\backslash m}(\boldsymbol{\theta}; \mathbb{P}_{n,\varepsilon,z}) \exp\{-\beta n_m L_m^{(t)}(\boldsymbol{\theta}; \mathbb{P}_{n_m,\varepsilon,z})\}}{\int q_{(t)}^{\backslash m}(\boldsymbol{\theta}; \mathbb{P}_{n,\varepsilon,z}) \exp\{-\beta n_m L_m^{(t)}(\boldsymbol{\theta}; \mathbb{P}_{n,\varepsilon,z})\} d\boldsymbol{\theta}}}{q_s^{(t-1)}(\boldsymbol{\theta}; \mathbb{P}_{n,\varepsilon,z})}$$

$$= -\log \frac{\frac{\frac{q_s^{(t-1)}(\boldsymbol{\theta}; \mathbb{P}_{n,\varepsilon,z})}{\exp\{-\beta n_m L_m^{(t-1)}(\boldsymbol{\theta}; \mathbb{P}_{n,\varepsilon,z})\}}}{\int \frac{q_s^{(t-1)}(\boldsymbol{\theta}; \mathbb{P}_{n,\varepsilon,z})}{\exp\{-\beta n_m L_m^{(t-1)}(\boldsymbol{\theta}; \mathbb{P}_{n,\varepsilon,z})\}} d\boldsymbol{\theta}} \exp\{-\beta n_m L_m^{(t)}(\boldsymbol{\theta}; \mathbb{P}_{n_m,\varepsilon,z})\}}{\int \frac{\frac{q_s^{(t-1)}(\boldsymbol{\theta}; \mathbb{P}_{n,\varepsilon,z})}{\exp\{-\beta n_m L_m^{(t-1)}(\boldsymbol{\theta}; \mathbb{P}_{n,\varepsilon,z})\}}}{\int \frac{q_s^{(t-1)}(\boldsymbol{\theta}; \mathbb{P}_{n,\varepsilon,z})}{\exp\{-\beta n_m L_m^{(t-1)}(\boldsymbol{\theta}; \mathbb{P}_{n,\varepsilon,z})\}} d\boldsymbol{\theta}} \exp\{-\beta n_m L_m^{(t)}(\boldsymbol{\theta}; \mathbb{P}_{n_m,\varepsilon,z})\} d\boldsymbol{\theta}}}{q_s^{(t-1)}(\boldsymbol{\theta}; \mathbb{P}_{n,\varepsilon,z})}$$

$$= -\log \frac{\frac{\exp\{-\beta n_m L_m^{(t)}(\boldsymbol{\theta}; \mathbb{P}_{n_m,\varepsilon,z})\}}{\exp\{-\beta n_m L_m^{(t-1)}(\boldsymbol{\theta}; \mathbb{P}_{n,\varepsilon,z})\}}}{\int \frac{q_s^{(t-1)}(\boldsymbol{\theta}; \mathbb{P}_{n,\varepsilon,z})}{\exp\{-\beta n_m L_m^{(t-1)}(\boldsymbol{\theta}; \mathbb{P}_{n,\varepsilon,z})\}} \exp\{-\beta n_m L_m^{(t)}(\boldsymbol{\theta}; \mathbb{P}_{n_m,\varepsilon,z})\} d\boldsymbol{\theta}}$$

$$= -\log \frac{\exp\{-\beta n_m L_m^{(t)}(\boldsymbol{\theta}; \mathbb{P}_{n_m,\varepsilon,z})\}}{\exp\{-\beta n_m L_m^{(t-1)}(\boldsymbol{\theta}; \mathbb{P}_{n,\varepsilon,z})\}} + \log Z_m^{(t)}(\mathbb{P}_{n,\varepsilon,z})$$

Therefore,

$$\ell_s^{(t)}(\boldsymbol{\theta}; \mathbb{P}_{n,\varepsilon,z}) = \sum_{m=1}^{M} -\log \frac{\exp\{-\beta n_m L_m^{(t)}(\boldsymbol{\theta}; \mathbb{P}_{n_m,\varepsilon,z})\}}{\exp\{-\beta n_m L_m^{(t-1)}(\boldsymbol{\theta}; \mathbb{P}_{n,\varepsilon,z})\}} + \log Z_m^{(t)}(\mathbb{P}_{n,\varepsilon,z}) + \ell_s^{(t-1)}(\boldsymbol{\theta}; \mathbb{P}_{n,\varepsilon,z})$$

$$= \sum_{m=1}^{M} -\log \exp\{-\beta n_m L_m^{(t)}(\boldsymbol{\theta}; \mathbb{P}_{n_m,\varepsilon,z})\} + \sum_{m=1}^{M} \sum_{i=1}^{t} \log Z_m^{(i)}(\mathbb{P}_{n,\varepsilon,z})$$

$$= \sum_{m=1}^{M} \beta n_m L_m^{(t)}(\boldsymbol{\theta}; \mathbb{P}_{n_m,\varepsilon,z}) + \sum_{m=1}^{M} \sum_{i=1}^{t} \log Z_m^{(i)}(\mathbb{P}_{n,\varepsilon,z})$$

We will now show by induction on $t$ that the posterior at the server is robust.

Concretely we will show that $\forall t \in [T]$, $T \in \mathbb{N}$, and $M \in \mathbb{N}$ finite, then

$$\sup_{z \in \Xi} \left| \frac{d}{d\varepsilon} \ell_s^{(t)}(\boldsymbol{\theta}; \mathbb{P}_{n,\varepsilon,z})\big|_{\varepsilon=0} \right| \leq \beta \sum_{m=1}^{M} n_m \sup_{z \in \Xi} \left| \frac{d}{d\varepsilon} L_m^{(t)}(\boldsymbol{\theta}; \mathbb{P}_{n_m,\varepsilon,z})\big|_{\varepsilon=0} \right| + \sum_{m=1}^{M} \sum_{i=1}^{t} \sup_{z \in \Xi} \left| \frac{d}{d\varepsilon} \log Z_m^{(i)}(\mathbb{P}_{n,\varepsilon,z})\big|_{\varepsilon=0} \right| \leq \gamma_s^{(t)}(\boldsymbol{\theta})$$

such that this function $\gamma_s^{(t)}(\boldsymbol{\theta})$ satisfies the conditions of Lemma B.4. Note that the first inequality follows by Minkowski's inequality.

We begin by considering the case where $t = 1$, then we have $q_s^{(1-1)}(\boldsymbol{\theta}; \mathbb{P}_{n,\varepsilon,z}) = \pi(\boldsymbol{\theta})$ and $L_m^{(1-1)}(\boldsymbol{\theta}; \mathbb{P}_{n,\varepsilon,z}) = 0$ as initialised in the algorithm.

Consider the term $\frac{d}{d\varepsilon} \log Z_m^{(i)}(\mathbb{P}_{n,\varepsilon,z})\big|_{\varepsilon=0}$, then we have

$$\frac{d}{d\varepsilon} \log Z_m^{(1)}(\mathbb{P}_{n,\varepsilon,z})\big|_{\varepsilon=0} = \frac{\frac{d}{d\varepsilon} Z_m^{(1)}(\mathbb{P}_{n,\varepsilon,z})|_{\varepsilon=0}}{Z_m^{(1)}(\mathbb{P}_{n,\varepsilon,z})|_{\varepsilon=0}}$$

$$= \frac{\int \frac{d}{d\varepsilon} \frac{q_s^{(1-1)}(\boldsymbol{\theta}; \mathbb{P}_{n,\varepsilon,z}) \exp\{-\beta n_m L_m^{(1)}(\boldsymbol{\theta}; \mathbb{P}_{n_m,\varepsilon,z})\}}{\exp\{-\beta n_m L_m^{(1-1)}(\boldsymbol{\theta}; \mathbb{P}_{-n_m,\varepsilon,z})\}}\Big|_{\varepsilon=0} d\boldsymbol{\theta}}{Z_m^{(1)}(\mathbb{P}_{n_m})}$$

$$= \frac{\int \frac{d}{d\varepsilon} \pi(\boldsymbol{\theta}) \exp\{-\beta n_m L_m^{(1)}(\boldsymbol{\theta}; \mathbb{P}_{n_m,\varepsilon,z})\}\big|_{\varepsilon=0} d\boldsymbol{\theta}}{Z_m^{(1)}(\mathbb{P}_{n_m})}$$

$$= -\int \left( \frac{d}{d\varepsilon} \beta n_m L_m^{(1)}(\boldsymbol{\theta}; \mathbb{P}_{n_m,\varepsilon,z})\big|_{\varepsilon=0} \right) q_m^{(1)}(\boldsymbol{\theta}; \mathbb{P}_{n_m}) d\boldsymbol{\theta}$$

where the last equation follows since $\frac{d}{dx} \exp\{f(x)\} = \exp\{f(x)\} \frac{d}{dx} f(x)$.

Consequently, using Jensen's inequality

$$\sup_{z \in \Xi} \left| \frac{d}{d\varepsilon} \ell_s^{(1)}(\boldsymbol{\theta}; \mathbb{P}_{n,\varepsilon,z})\big|_{\varepsilon=0} \right|$$

$$\leq \beta \sum_{m=1}^{M} n_m \sup_{z \in \Xi} \left| \frac{d}{d\varepsilon} L_m^{(1)}(\boldsymbol{\theta}; \mathbb{P}_{n_m,\varepsilon,z})\big|_{\varepsilon=0} \right| + \sum_{m=1}^{M} \sup_{z \in \Xi} \left| \int \left( \frac{d}{d\varepsilon} \beta n_m L_m^{(1)}(\boldsymbol{\theta}; \mathbb{P}_{n_m,\varepsilon,z})\big|_{\varepsilon=0} \right) q_m^{(1)}(\boldsymbol{\theta}; \mathbb{P}_{n_m}) d\boldsymbol{\theta} \right|$$

$$\leq \beta \sum_{m=1}^{M} n_m \left( \sup_{z \in \Xi} \left| \frac{d}{d\varepsilon} L_m^{(1)}(\boldsymbol{\theta}; \mathbb{P}_{n_m,\varepsilon,z})\big|_{\varepsilon=0} \right| + \int \sup_{z \in \Xi} \left| \frac{d}{d\varepsilon} L_m^{(1)}(\boldsymbol{\theta}; \mathbb{P}_{n_m,\varepsilon,z})\big|_{\varepsilon=0} \right| q_m^{(1)}(\boldsymbol{\theta}; \mathbb{P}_{n_m}) d\boldsymbol{\theta} \right)$$

Then, if $L_m^{(1)}(\boldsymbol{\theta}; \mathbb{P}_{n_m,\varepsilon,z})$ is robust, then there exists some function $\gamma_m^{(1)}(\boldsymbol{\theta})$ : $\Theta \rightarrow \mathbb{R}$ such that $\sup_{z \in \Xi} \left| \frac{d}{d\varepsilon} L_m^{(1)}(\boldsymbol{\theta}; \mathbb{P}_{n_m,\varepsilon,z})\big|_{\varepsilon=0} \right| \leq \gamma_m^{(1)}(\boldsymbol{\theta})$ and which satisfies:

$$\sup_{\boldsymbol{\theta} \in \Theta} \pi(\boldsymbol{\theta}) \gamma_m^{(1)}(\boldsymbol{\theta}) < \infty, \quad \text{and} \quad \int \pi(\boldsymbol{\theta}) \gamma_m^{(1)}(\boldsymbol{\theta}) d\boldsymbol{\theta} < \infty.$$

Substituting this into the above, we have that

$$\sup_{z \in \Xi} \left| \frac{d}{d\varepsilon} \ell_s^{(1)}(\boldsymbol{\theta}; \mathbb{P}_{n,\varepsilon,z}) \big|_{\varepsilon=0} \right| \leq \beta \sum_{m=1}^{M} n_m \left( \gamma_m^{(1)}(\boldsymbol{\theta}) + \int \gamma_m^{(1)}(\boldsymbol{\theta}) q_m^{(1)}(\boldsymbol{\theta}; \mathbb{P}_{n_m}) d\boldsymbol{\theta} \right)$$

Now recall that $q_m^{(1)}(\boldsymbol{\theta}; \mathbb{P}_{n_m}) = \pi(\boldsymbol{\theta}) \exp\{-\beta n_m L_m^{(1)}(\boldsymbol{\theta}; \mathbb{P}_{n_m})\}/Z_{(1)}^m(\mathbb{P}_{n_m})$, and per the assumption we have that the loss is lower bounded and that $0 < Z_{(1)}^m(\mathbb{P}_{n_m}) < \infty$, therefore $q_m^{(1)}(\boldsymbol{\theta}; \mathbb{P}_{n_m}) \leq \pi(\boldsymbol{\theta}) \exp\{-\beta n_m \inf_{\boldsymbol{\theta} \in \Theta} L_m^{(1)}(\boldsymbol{\theta}; \mathbb{P}_{n_m})\}/Z_{(1)}^m(\mathbb{P}_{n_m}) \leq C_m^{(1)} \pi(\boldsymbol{\theta})$ so that,

$$\sup_{z \in \Xi} \left| \frac{d}{d\varepsilon} \ell_s^{(1)}(\boldsymbol{\theta}; \mathbb{P}_{n,\varepsilon,z}) \big|_{\varepsilon=0} \right| \leq \beta \sum_{m=1}^{M} n_m \left( \gamma_m^{(1)}(\boldsymbol{\theta}) + C_m^{(1)} \int \gamma_m^{(1)}(\boldsymbol{\theta}) \pi(\boldsymbol{\theta}) d\boldsymbol{\theta} \right) =: \gamma_s^{(1)}(\boldsymbol{\theta})$$

We now verify that the conditions hold. For condition 2, we have

$$\sup_{\boldsymbol{\theta} \in \Theta} \pi(\boldsymbol{\theta}) \gamma_s^{(1)}(\boldsymbol{\theta}) \leq \beta \sum_{m=1}^{M} n_m \left( \left( \sup_{\boldsymbol{\theta} \in \Theta} \pi(\boldsymbol{\theta}) \gamma_m^{(1)}(\boldsymbol{\theta}) \right) + \left( \sup_{\boldsymbol{\theta} \in \Theta} \pi(\boldsymbol{\theta}) \right) C_m^{(1)} \int \gamma_m^{(1)}(\boldsymbol{\theta}) \pi(\boldsymbol{\theta}) d\boldsymbol{\theta} \right) < \infty$$

which follows by the assumptions on the robustness of the loss and that the prior is upper bounded, as well as the finiteness of $\beta$, $n_m$, and $C_m^{(1)}$.

Condition 3 follows similar reasoning.

$$\int \gamma_s^{(1)}(\boldsymbol{\theta}) \pi(\boldsymbol{\theta}) d\boldsymbol{\theta} = \int \beta \sum_{m=1}^{M} n_m \left( \gamma_m^{(1)}(\boldsymbol{\theta}) + C_m^{(1)} \int \gamma_m^{(1)}(\boldsymbol{\theta}) \pi(\boldsymbol{\theta}) d\boldsymbol{\theta} \right) \pi(\boldsymbol{\theta}) d\boldsymbol{\theta}$$

$$= \beta \sum_{m=1}^{M} n_m \left( \int \pi(\boldsymbol{\theta}) \gamma_m^{(1)}(\boldsymbol{\theta}) d\boldsymbol{\theta} + \int \pi(\boldsymbol{\theta}) C_m^{(1)} \left( \int \gamma_m^{(1)}(\boldsymbol{\theta}) \pi(\boldsymbol{\theta}) d\boldsymbol{\theta} \right) d\boldsymbol{\theta} \right)$$

$$= \beta \sum_{m=1}^{M} n_m \left( \int \pi(\boldsymbol{\theta}) \gamma_m^{(1)}(\boldsymbol{\theta}) d\boldsymbol{\theta} + C_m^{(1)} \int \gamma_m^{(1)}(\boldsymbol{\theta}) \pi(\boldsymbol{\theta}) d\boldsymbol{\theta} \right) < \infty$$

Since the loss is robust, the integrals are finite, and since all other terms are finite, we conclude that condition 3 is also satisfied. Therefore, for $t = 1$ the posterior computed at the server satisfies the conditions of Lemma B.4 and is therefore globally bias–robust. It remains to be shown that this holds for all $t \in \mathbb{N}$ such that $t \leq T$, i.e. is finite.

We now show by induction that if the posterior at the server is robust for $t = k$, then it will also be robust for $t = k + 1$.

$$\frac{d}{d\varepsilon} \ell_s^{(k+1)}(\boldsymbol{\theta}; \mathbb{P}_{n,\varepsilon,z}) = \beta \sum_{m=1}^{M} n_m \frac{d}{d\varepsilon} L_m^{(k+1)}(\boldsymbol{\theta}; \mathbb{P}_{n_m,\varepsilon,z}) \big|_{\varepsilon=0} + \sum_{m=1}^{M} \sum_{t=1}^{k+1} \frac{d}{d\varepsilon} \log Z_m^{(t)}(\mathbb{P}_{n,\varepsilon,z}) \big|_{\varepsilon=0}$$

$$= \beta \sum_{m=1}^{M} n_m \frac{d}{d\varepsilon} L_m^{(k+1)}(\boldsymbol{\theta}; \mathbb{P}_{n_m,\varepsilon,z}) \big|_{\varepsilon=0} + \sum_{m=1}^{M} \sum_{t=1}^{k+1} \underbrace{\frac{\frac{d}{d\varepsilon} Z_m^{(t)}(\mathbb{P}_{n,\varepsilon,z})|_{\varepsilon=0}}{Z_m^{(t)}(\mathbb{P}_{n,\varepsilon,z})|_{\varepsilon=0}}}_{(1)}$$

To show the boundedness of this, we need to consider the expansion of (1) above.

$$\frac{d}{d\varepsilon} \log Z_m^{(t)}(\mathbb{P}_{n,\varepsilon,z}) \big|_{\varepsilon=0} = \frac{\frac{d}{d\varepsilon} Z_m^{(t)}(\mathbb{P}_{n,\varepsilon,z})|_{\varepsilon=0}}{Z_m^{(t)}(\mathbb{P}_{n,\varepsilon,z})|_{\varepsilon=0}} = \frac{\int \frac{d}{d\varepsilon} \frac{q_s^{(t-1)}(\boldsymbol{\theta}; \mathbb{P}_{n,\varepsilon,z}) \exp\{-\beta n_m L_m^{(t)}(\boldsymbol{\theta}; \mathbb{P}_{n_m,\varepsilon,z})\}}{\exp\{-\beta n_m L_m^{(t-1)}(\boldsymbol{\theta}; \mathbb{P}_{n_m,\varepsilon,z})\}} \big|_{\varepsilon=0} d\boldsymbol{\theta}}{Z_m^{(t)}(\mathbb{P}_n)}$$

Now we consider the integral in the numerator. Using the chain rule when differentiating under the integral sign:

$$\int \frac{d}{d\varepsilon} \frac{q_s^{(t-1)}(\boldsymbol{\theta}; \mathbb{P}_{n,\varepsilon,z}) \exp\{-\beta n_m L_m^{(t)}(\boldsymbol{\theta}; \mathbb{P}_{n_m,\varepsilon,z})\}}{\exp\{-\beta n_m L_m^{(t-1)}(\boldsymbol{\theta}; \mathbb{P}_{n_m,\varepsilon,z})\}} \big|_{\varepsilon=0} d\boldsymbol{\theta}$$

$$= \int \left[ \frac{\exp\{-\beta n_m L_m^{(t)}(\boldsymbol{\theta}; \mathbb{P}_{n_m})\}}{\exp\{-\beta n_m L_m^{(t-1)}(\boldsymbol{\theta}; \mathbb{P}_{n_m})\}} \frac{d}{d\varepsilon} q_s^{(t-1)}(\boldsymbol{\theta}; \mathbb{P}_{n,\varepsilon,z}) \Big|_{\varepsilon=0} \right.$$

$$+ \frac{q_s^{(t-1)}(\boldsymbol{\theta}; \mathbb{P}_n) \exp\{-\beta n_m L_m^{(t)}(\boldsymbol{\theta}; \mathbb{P}_{n_m})\}}{\exp\{-\beta n_m L_m^{(t-1)}(\boldsymbol{\theta}; \mathbb{P}_{n_m})\}} \frac{d}{d\varepsilon}(-\beta n_m L_m^{(t)}(\boldsymbol{\theta}; \mathbb{P}_{n_m,\varepsilon,z})) \Big|_{\varepsilon=0}$$

$$\left. - \frac{q_s^{(t-1)}(\boldsymbol{\theta}; \mathbb{P}_n) \exp\{-\beta n_m L_m^{(t)}(\boldsymbol{\theta}; \mathbb{P}_{n_m})\}}{\exp\{-\beta n_m L_m^{(t-1)}(\boldsymbol{\theta}; \mathbb{P}_{n_m})\}} \frac{d}{d\varepsilon}(-\beta n_m L_m^{(t-1)}(\boldsymbol{\theta}; \mathbb{P}_{n_m,\varepsilon,z})) \Big|_{\varepsilon=0} \right] d\boldsymbol{\theta}$$

Bringing the denominator back, and recalling the definition of $q_m^{(t)}(\boldsymbol{\theta}; \mathbb{P}_{n_m})$, then we can simplify.

$$\frac{d}{d\varepsilon} \log Z_m^{(t)}(\mathbb{P}_{n,\varepsilon,z}) \Big|_{\varepsilon=0} = \int \left[ \frac{\left( \dfrac{\exp\{-\beta n_m L_m^{(t)}(\boldsymbol{\theta}; \mathbb{P}_{n_m})\}}{\exp\{-\beta n_m L_m^{(t-1)}(\boldsymbol{\theta}; \mathbb{P}_{n_m})\}} \right)}{Z_m^{(t)}(\mathbb{P}_n)} \frac{d}{d\varepsilon} q_s^{(t-1)}(\boldsymbol{\theta}; \mathbb{P}_{n,\varepsilon,z}) \Big|_{\varepsilon=0} \right.$$

$$- \underbrace{\frac{\dfrac{q_s^{(t-1)}(\boldsymbol{\theta}; \mathbb{P}_n) \exp\{-\beta n_m L_m^{(t)}(\boldsymbol{\theta}; \mathbb{P}_{n_m})\}}{\exp\{-\beta n_m L_m^{(t-1)}(\boldsymbol{\theta}; \mathbb{P}_{n_m})\}}}{Z_m^{(t)}(\mathbb{P}_n)}}_{=q_m^{(t)}(\boldsymbol{\theta}; \mathbb{P}_{n_m})} \frac{d}{d\varepsilon}(\beta n_m L_m^{(t)}(\boldsymbol{\theta}; \mathbb{P}_{n_m,\varepsilon,z})) \Big|_{\varepsilon=0}$$

$$\left. + \underbrace{\frac{\dfrac{q_s^{(t-1)}(\boldsymbol{\theta}; \mathbb{P}_n) \exp\{-\beta n_m L_m^{(t)}(\boldsymbol{\theta}; \mathbb{P}_{n_m})\}}{\exp\{-\beta n_m L_m^{(t-1)}(\boldsymbol{\theta}; \mathbb{P}_{n_m})\}}}{Z_m^{(t)}(\mathbb{P}_n)}}_{=q_m^{(t)}(\boldsymbol{\theta}; \mathbb{P}_{n_m})} \frac{d}{d\varepsilon}(\beta n_m L_m^{(t-1)}(\boldsymbol{\theta}; \mathbb{P}_{n_m,\varepsilon,z})) \Big|_{\varepsilon=0} \right] d\boldsymbol{\theta}$$

$$= \int \left[ \left( \frac{\exp\{-\beta n_m L_m^{(t)}(\boldsymbol{\theta}; \mathbb{P}_{n_m})\}}{Z_m^{(t)}(\mathbb{P}_n) \exp\{-\beta n_m L_m^{(t-1)}(\boldsymbol{\theta}; \mathbb{P}_{n_m})\}} \frac{d}{d\varepsilon} q_s^{(t-1)}(\boldsymbol{\theta}; \mathbb{P}_{n,\varepsilon,z}) \Big|_{\varepsilon=0} \right. \right.$$

$$- q_m^{(t)}(\boldsymbol{\theta}; \mathbb{P}_{n_m}) \frac{d}{d\varepsilon}(\beta n_m L_m^{(t)}(\boldsymbol{\theta}; \mathbb{P}_{n_m,\varepsilon,z})) \Big|_{\varepsilon=0}$$

$$\left. + q_m^{(t)}(\boldsymbol{\theta}; \mathbb{P}_{n_m}) \frac{d}{d\varepsilon}(\beta n_m L_m^{(t-1)}(\boldsymbol{\theta}; \mathbb{P}_{n_m,\varepsilon,z})) \Big|_{\varepsilon=0} \right] d\boldsymbol{\theta}$$

Consider now the derivative of the previous server posterior with respect to $\varepsilon$ evaluated at 0, which we can write as:

$$\frac{d}{d\varepsilon} q_s^{(t-1)}(\boldsymbol{\theta}; \mathbb{P}_{n,\varepsilon,z}) \Big|_{\varepsilon=0} = \frac{d}{d\varepsilon} \frac{\pi(\boldsymbol{\theta}) \exp\{-\ell_s^{(t-1)}(\boldsymbol{\theta}; \mathbb{P}_{n,\varepsilon,z})\}}{Z_s^{(t-1)}(\mathbb{P}_{n,\varepsilon,z})} \Big|_{\varepsilon=0}$$

$$= \pi(\boldsymbol{\theta}) \left( \frac{\exp\{-\ell_s^{(t-1)}(\boldsymbol{\theta}; \mathbb{P}_n)\}}{Z_s^{(t-1)}(\mathbb{P}_n)} \frac{d}{d\varepsilon}(-\ell_s^{(t-1)}(\boldsymbol{\theta}; \mathbb{P}_{n,\varepsilon,z})) \Big|_{\varepsilon=0} \right.$$

$$\left. - \frac{\exp\{-\ell_s^{(t-1)}(\boldsymbol{\theta}; \mathbb{P}_n)\}}{(Z_s^{(t-1)}(\mathbb{P}_n))^2} \int \pi(\boldsymbol{\theta}) \frac{d}{d\varepsilon} \exp\{-\ell_s^{(t-1)}(\boldsymbol{\theta}; \mathbb{P}_{n,\varepsilon,z})\} \Big|_{\varepsilon=0} d\boldsymbol{\theta} \right)$$

$$= -q_s^{(t-1)}(\boldsymbol{\theta}; \mathbb{P}_n) \left( \frac{d}{d\varepsilon} \ell_s^{(t-1)}(\boldsymbol{\theta}; \mathbb{P}_{n,\varepsilon,z}) \Big|_{\varepsilon=0} - \int q_s^{(t-1)}(\boldsymbol{\theta}; \mathbb{P}_n) \frac{d}{d\varepsilon} \ell_s^{(t-1)}(\boldsymbol{\theta}; \mathbb{P}_{n,\varepsilon,z}) \Big|_{\varepsilon=0} d\boldsymbol{\theta} \right)$$

where we have used the definition of $q_s^{(t-1)}(\boldsymbol{\theta}; \mathbb{P}_n)$ by distributing the common terms outside the brackets and for the second term, since the normalising constant does not depend on $\boldsymbol{\theta}$, we can take one of them inside the integral. Furthermore, using the fact that

$$\frac{\dfrac{q_s^{(t-1)}(\boldsymbol{\theta}; \mathbb{P}_n) \exp\{-\beta n_m L_m^{(t)}(\boldsymbol{\theta}; \mathbb{P}_{n_m})\}}{\exp\{-\beta n_m L_m^{(t-1)}(\boldsymbol{\theta}; \mathbb{P}_{n_m})\}}}{Z_m^{(t)}(\mathbb{P}_n)} = q_m^{(t)}(\boldsymbol{\theta}; \mathbb{P}_{n_m})$$

then substituting the result for $\frac{d}{d\varepsilon}q_s^{(t-1)}(\boldsymbol{\theta};\mathbb{P}_{n,\varepsilon,z})\big|_{\varepsilon=0}$ into $\frac{d}{d\varepsilon}\log Z_m^{(t)}(\mathbb{P}_{n,\varepsilon,z})\big|_{\varepsilon=0}$, we get that:

$$
\frac{d}{d\varepsilon}\log Z_m^{(t)}(\mathbb{P}_{n,\varepsilon,z})\big|_{\varepsilon=0} = \int q_m^{(t)}(\boldsymbol{\theta};\mathbb{P}_{n_m})\left(-\frac{d}{d\varepsilon}\ell_s^{(t-1)}(\boldsymbol{\theta};\mathbb{P}_{n,\varepsilon,z})\Big|_{\varepsilon=0} + \int q_s^{(t-1)}(\boldsymbol{\theta};\mathbb{P}_n)\frac{d}{d\varepsilon}\ell_s^{(t-1)}(\boldsymbol{\theta};\mathbb{P}_{n,\varepsilon,z})\big|_{\varepsilon=0}d\boldsymbol{\theta}\right.
$$
$$
\left. -\beta n_m\frac{d}{d\varepsilon}L_m^{(t)}(\boldsymbol{\theta};\mathbb{P}_{n_m,\varepsilon,z})\big|_{\varepsilon=0} + \beta n_m\frac{d}{d\varepsilon}L_m^{(t-1)}(\boldsymbol{\theta};\mathbb{P}_{n_m,\varepsilon,z})\big|_{\varepsilon=0}\right)d\boldsymbol{\theta}
$$

Substituting this expression back into the original equation for $\frac{d}{d\varepsilon}\ell_s^{(t)}(\boldsymbol{\theta};\mathbb{P}_{n,\varepsilon,z})|_{\varepsilon=0}$, taking the supremum over $z\in\Xi$ of the absolute value of this, and applying Minkowski's inequality, results in the following upper bound.

$$
\sup_{z\in\Xi}\left|\frac{d}{d\varepsilon}\ell_s^{(k+1)}(\boldsymbol{\theta};\mathbb{P}_{n,\varepsilon,z})\right|
$$
$$
\leq \sum_{m=1}^{M}\beta n_m\sup_{z\in\Xi}\left|\frac{d}{d\varepsilon}L_m^{(k+1)}(\boldsymbol{\theta};\mathbb{P}_{n_m,\varepsilon,z})\big|_{\varepsilon=0}\right| + \sum_{m=1}^{M}\sum_{t=1}^{k+1}\left\{\int q_m^{(t)}(\boldsymbol{\theta};\mathbb{P}_{n_m})\left[\sup_{z\in\Xi}\left|\frac{d}{d\varepsilon}\ell_s^{(t-1)}(\boldsymbol{\theta};\mathbb{P}_{n,\varepsilon,z})\big|_{\varepsilon=0}\right|\right.\right.
$$
$$
+ \left(\int_{\Theta}\sup_{z\in\Xi}\left|\frac{d}{d\varepsilon}\ell_s^{(t-1)}(\boldsymbol{\theta};\mathbb{P}_{n,\varepsilon,z})\big|_{\varepsilon=0}\right|q_s^{(t-1)}(\boldsymbol{\theta};\mathbb{P}_n)d\boldsymbol{\theta}\right) + \beta n_m\sup_{z\in\Xi}\left|\frac{d}{d\varepsilon}L_m^{(t)}(\boldsymbol{\theta};\mathbb{P}_{n_m,\varepsilon,z})\big|_{\varepsilon=0}\right|
$$
$$
\left.\left. + \beta n_m\sup_{z\in\Xi}\left|\frac{d}{d\varepsilon}L_m^{(t-1)}(\boldsymbol{\theta};\mathbb{P}_{n_m,\varepsilon,z})\big|_{\varepsilon=0}\right|\right]d\boldsymbol{\theta}\right\}
$$

By the inductive assumption $\forall t\in[k+1],\exists\gamma_s^{(t-1)}(\boldsymbol{\theta})$ such that $\sup_{z\in\Xi}\left|\frac{d}{d\varepsilon}\ell_s^{(t-1)}(\boldsymbol{\theta};\mathbb{P}_{n,\varepsilon,z})|_{\varepsilon=0}\right| \leq \gamma_s^{(t-1)}(\boldsymbol{\theta})$. Additionally, as, by assumption, the loss is lower bounded and robust $\exists\gamma_m^{(t)}(\boldsymbol{\theta})\,\forall t\in[k+1]$ such that $\sup_{z\in\Xi}\left|L_m^{(t)}(\boldsymbol{\theta};\mathbb{P}_{n_m,\varepsilon,z})|_{\varepsilon=0}\right| \leq \gamma_m^{(t)}(\boldsymbol{\theta})$. Furthermore, these functions satisfy the conditions of Lemma B.4. Note also that $q_s^{(t-1)}(\boldsymbol{\theta};\mathbb{P}_n) \leq C_s^{(t-1)}\pi(\boldsymbol{\theta})$, since the normalising constant of this distribution is finite and the loss is lower bounded per the inductive assumption, so we get $q_s^{(t-1)}(\boldsymbol{\theta};\mathbb{P}_n) \leq \pi(\boldsymbol{\theta})\exp\{-\inf_{\boldsymbol{\theta}\in\Theta}\ell_s^{(t-1)}(\boldsymbol{\theta};\mathbb{P}_n)\}/Z_s^{(t-1)}(\mathbb{P}_n) \leq C_s^{(t-1)}\pi(\boldsymbol{\theta})$, as seen in similar arguments before. Utilising this, we conclude:

$$
\leq \sum_{m=1}^{M}\beta n_m\gamma_m^{(k+1)}(\boldsymbol{\theta}) + \sum_{m=1}^{M}\sum_{t=1}^{k+1}\int q_m^{(t)}(\boldsymbol{\theta};\mathbb{P}_{n_m})\left\{\gamma_s^{(t-1)}(\boldsymbol{\theta}) + \left(\int\gamma_s^{(t-1)}(\boldsymbol{\theta})C_s^{(t-1)}\pi(\boldsymbol{\theta})d\boldsymbol{\theta}\right)\right.
$$
$$
\left. + \beta n_m\gamma_m^{(t)}(\boldsymbol{\theta}) + \beta n_m\gamma_m^{(t-1)}(\boldsymbol{\theta})\right\}d\boldsymbol{\theta} := \gamma_s^{(k+1)}(\boldsymbol{\theta})
$$

We now need to show that this satisfies conditions (2) and (3) of Lemma B.4. Let's recall what these conditions state:

$$
(2) = \sup_{\boldsymbol{\theta}\in\Theta}\pi(\boldsymbol{\theta})\gamma_s^{(k+1)}(\boldsymbol{\theta}) < \infty \tag{11}
$$
$$
(3) = \int\pi(\boldsymbol{\theta})\gamma_s^{(k+1)}(\boldsymbol{\theta})d\boldsymbol{\theta} < \infty \tag{12}
$$

We first verify that condition (2) holds.

$$
\sup_{\boldsymbol{\theta}\in\Theta}\pi(\boldsymbol{\theta})\gamma_s^{(k+1)}(\boldsymbol{\theta}) = \sup_{\boldsymbol{\theta}\in\Theta}\pi(\boldsymbol{\theta})\left\{\sum_{m=1}^{M}\beta n_m\gamma_m^{(k+1)}(\boldsymbol{\theta})\right.
$$
$$
\left. + \sum_{m=1}^{M}\sum_{t=1}^{k+1}\int q_m^{(t)}(\boldsymbol{\theta};\mathbb{P}_{n_m})\left\{\gamma_s^{(t-1)}(\boldsymbol{\theta}) + C_s^{(t-1)}\int\gamma_s^{(t-1)}(\boldsymbol{\theta})\pi(\boldsymbol{\theta})d\boldsymbol{\theta} + \beta n_m\gamma_m^{(t)}(\boldsymbol{\theta}) + \beta n_m\gamma_m^{(t-1)}(\boldsymbol{\theta})\right\}d\boldsymbol{\theta}\right\}
$$
$$
\leq \beta n_m\sum_{m=1}^{M}\sup_{\boldsymbol{\theta}\in\Theta}\pi(\boldsymbol{\theta})\gamma_m^{(k+1)}(\boldsymbol{\theta}) + \sup_{\boldsymbol{\theta}\in\Theta}\pi(\boldsymbol{\theta})\left\{\sum_{m=1}^{M}\sum_{t=1}^{k+1}\int q_m^{(t)}(\boldsymbol{\theta};\mathbb{P}_{n_m})\left[\gamma_s^{(t)}(\boldsymbol{\theta})\right.\right.
$$
$$
\left.\left. + C_s^{(t-1)}\int\gamma_s^{(t-1)}(\boldsymbol{\theta})\pi(\boldsymbol{\theta})d\boldsymbol{\theta} + \beta n_m\gamma_m^{(t)}(\boldsymbol{\theta}) + \beta n_m\gamma_m^{(t-1)}(\boldsymbol{\theta})\right]d\boldsymbol{\theta}\right\} < \infty
$$

Since $\beta, n_m$, and $M$ are finite, and any finite linear combination of finite terms is finite, we can easily see that the first part is finite. This follows since $\gamma_m^{(k+1)}(\boldsymbol{\theta})$ satisfies condition (2) of Lemma B.4. Furthermore, since $\pi(\boldsymbol{\theta})$ is upper bounded, we

now need to verify whether the inside of the curly brackets is finite. Since this is a finite sum, we need to verify if $\forall m \in [M]$ and $\forall t \in [k+1]$, the following holds:

$$\int q_m^{(t)}(\boldsymbol{\theta}; \mathbb{P}_{n_m}) \left[ \gamma_s^{(t)}(\boldsymbol{\theta}) + C_s^{(t-1)} \int \gamma_s^{(t-1)}(\boldsymbol{\theta})\pi(\boldsymbol{\theta})d\boldsymbol{\theta} + \beta n_m \gamma_m^{(t)}(\boldsymbol{\theta}) + \beta n_m \gamma_m^{(t-1)}(\boldsymbol{\theta}) \right] d\boldsymbol{\theta} < \infty$$

By the inductive step, this is true $\forall t \in [k]$, so we need to show that it also holds for $t = k+1$. So,

$$\int q_m^{(k+1)}(\boldsymbol{\theta}; \mathbb{P}_{n_m}) \left[ \gamma_s^{(k+1)}(\boldsymbol{\theta}) + C_s^{(k)} \int \gamma_s^{(k)}(\boldsymbol{\theta})\pi(\boldsymbol{\theta})d\boldsymbol{\theta} + \beta n_m \gamma_m^{(k+1)}(\boldsymbol{\theta}) + \beta n_m \gamma_m^{(k)}(\boldsymbol{\theta}) \right] d\boldsymbol{\theta} < \infty$$

Note that $q_m^{(k+1)}(\boldsymbol{\theta}; \mathbb{P}_{n_m})$ is equal to $\pi(\boldsymbol{\theta})\exp\{-\beta n_m L_m^{(k+1)}(\boldsymbol{\theta}; \mathbb{P}_{n_m})\}\exp\{-\beta \sum_{i \neq m} n_i L_i^{(k)}(\boldsymbol{\theta}; \mathbb{P}_{n_i})\}/Z_m^{(k+1)}Z_s^{(k)}$, and since the normalising constants are finite and positive, and the losses are lower bounded, then we can write

$$q_m^{(k+1)}(\boldsymbol{\theta}; \mathbb{P}_{n_m}) \leq \pi(\boldsymbol{\theta})\exp\{-\beta n_m \inf_{\boldsymbol{\theta} \in \Theta} L_m^{(k+1)}(\boldsymbol{\theta}; \mathbb{P}_{n_m})\}\exp\{-\beta \sum_{i \neq m} n_i \inf_{\boldsymbol{\theta} \in \Theta} L_i^{(k)}(\boldsymbol{\theta}; \mathbb{P}_{n_i})\}/Z_m^{(k+1)}Z_s^{(k)}$$

$$\leq C_m^{(k+1)}\pi(\boldsymbol{\theta})$$

where $0 < C_m^{(k+1)} < \infty$. Thereby, we have

$$\int q_m^{(k+1)}(\boldsymbol{\theta}; \mathbb{P}_{n_m}) \left[ \gamma_s^{(k+1)}(\boldsymbol{\theta}) + C_s^{(k)} \int \gamma_s^{(k)}(\boldsymbol{\theta})\pi(\boldsymbol{\theta})d\boldsymbol{\theta} + \beta n_m \gamma_m^{(k+1)}(\boldsymbol{\theta}) + \beta n_m \gamma_m^{(k)}(\boldsymbol{\theta}) \right] d\boldsymbol{\theta}$$

$$\leq C_m^{(k+1)} \int \pi(\boldsymbol{\theta})\gamma_s^{(k+1)}(\boldsymbol{\theta})d\boldsymbol{\theta} + C_m^{(k+1)}C_s^{(k)} \left( \int \gamma_s^{(k)}(\boldsymbol{\theta})\pi(\boldsymbol{\theta})d\boldsymbol{\theta} \right) \left( \int \pi(\boldsymbol{\theta})d\boldsymbol{\theta} \right)$$

$$+ \beta n_m C_m^{(k+1)} \int \pi(\boldsymbol{\theta})\gamma_m^{(k+1)}(\boldsymbol{\theta})d\boldsymbol{\theta} + \beta n_m C_m^{(k+1)} \int \pi(\boldsymbol{\theta})\gamma_m^{(k)}(\boldsymbol{\theta})d\boldsymbol{\theta} < \infty$$

This expression is finite since the individual integrals must be finite by the definition of the bounding functions $\gamma$, as these need to satisfy condition (3) of Lemma B.4 with the prior $\pi(\boldsymbol{\theta})$. Hence, we have shown that condition (2) holds for $\gamma_s^{(k+1)}(\boldsymbol{\theta})$ and Equation (11) is indeed finite.

It remains to be shown that condition (3), Equation (12), also holds. Using the same expression for $\gamma_s^{(k+1)}(\boldsymbol{\theta})$ as before, we have:

$$\int \pi(\boldsymbol{\theta})\gamma_s^{(k+1)}(\boldsymbol{\theta})d\boldsymbol{\theta} = \int \pi(\boldsymbol{\theta})\left\{ \sum_{m=1}^{M} \beta n_m \gamma_m^{(k+1)}(\boldsymbol{\theta}) \right.$$

$$\left. + \sum_{m=1}^{M}\sum_{t=1}^{k+1} \int q_m^{(t)}(\boldsymbol{\theta}; \mathbb{P}_{n_m})\left\{ \gamma_s^{(t-1)}(\boldsymbol{\theta}) + C_s^{(t-1)} \int \gamma_s^{(t-1)}(\boldsymbol{\theta})\pi(\boldsymbol{\theta})d\boldsymbol{\theta} + \beta n_m \gamma_m^{(t)}(\boldsymbol{\theta}) + \beta n_m \gamma_m^{(t-1)}(\boldsymbol{\theta}) \right\}d\boldsymbol{\theta} \right\} d\boldsymbol{\theta}$$

Since, the summations are finite, we can exchange the integrals and sums to get

$$= \left( \sum_{m=1}^{M} \beta n_m \underbrace{\int \pi(\boldsymbol{\theta})\gamma_m^{(k+1)}(\boldsymbol{\theta})d\boldsymbol{\theta}}_{<\infty \; \forall m \in [M]} \right) + \underbrace{\int \pi(\boldsymbol{\theta})d\boldsymbol{\theta}}_{=1} \left( \sum_{m=1}^{M}\sum_{t=1}^{k+1} \int q_m^{(t)}(\boldsymbol{\theta}; \mathbb{P}_{n_m})\left\{ \gamma_s^{(t-1)}(\boldsymbol{\theta}) + C_s^{(t-1)} \int \gamma_s^{(t-1)}(\boldsymbol{\theta})\pi(\boldsymbol{\theta})d\boldsymbol{\theta} \right.\right.$$

$$\left.\left. + \beta n_m \gamma_m^{(t)}(\boldsymbol{\theta}) + \beta n_m \gamma_m^{(t-1)}(\boldsymbol{\theta}) \right\}d\boldsymbol{\theta} \right)$$

where the first part is finite since for each $\gamma_m^{(k+1)}(\boldsymbol{\theta})$, we have by definition that this expression is finite as it needs to satisfy condition (3). Therefore, we need to show that the summation is finite. By the inductive step, this is true $\forall t \in [k]$, and we will now show that $\forall m \in [M]$ it also is finite for $t = k+1$.

$$\int q_m^{(k+1)}(\boldsymbol{\theta}; \mathbb{P}_{n_m})\left\{ \gamma_s^{(k)}(\boldsymbol{\theta}) + C_s^{(k)} \int \gamma_s^{(k)}(\boldsymbol{\theta})\pi(\boldsymbol{\theta})d\boldsymbol{\theta} + \beta n_m \gamma_m^{(k+1)}(\boldsymbol{\theta}) + \beta n_m \gamma_m^{(k)}(\boldsymbol{\theta}) \right\}d\boldsymbol{\theta}$$

Recall from before that $q_m^{(k+1)}(\boldsymbol{\theta}; \mathbb{P}_{n_m}) \leq C_m^{(k+1)} \pi(\boldsymbol{\theta})$ and hence, it is now immediate to see that by the same argument as in the proof of condition (2), this integral is finite. Therefore, condition (3) of Lemma B.4 also holds and Equation (12) is true.

We conclude that all conditions of Lemma B.4 are satisfied.

Therefore, by induction, as long as we have a robust loss function (in the sense of Ghosh & Basu, 2016a; Matsubara et al., 2022) at the clients, then irregardless of the current iteration by using the weighted KL divergence at the clients and the KL divergence at the server, FEDGVI achieves global bias robustness to outliers. ∎

Note that when assuming that $q_m^{(k+1)}(\boldsymbol{\theta}; \mathbb{P}_{n_m}) \leq C_m^{(k+1)} \pi(\boldsymbol{\theta})$, or similarly at the server in the uncontaminated case, we have used that the normalising constants in the well specified case are finite. This is necessary to hold, since otherwise we will not have valid distributions, and furthermore we can always choose a prior distribution that is bounded above so this will always be finite. However, this finiteness is not assumed for the normalising constants that are contaminated by the outliers, so the proof is needed to show boundedness of the posterior influence under contamination.

## C. Additional Details on FEDGVI

We present some additional details on FEDGVI in order to aid clarity and contextualise it in the broader literature.

### C.1. A Note on the Learning Rate Parameter in GBI

The $\beta$ parameter comes from the power/cold/tempered posteriors of e.g. Grünwald (2012), where the likelihood in Bayesian posteriors is raised to some power of $\beta > 0$. This was originally done to add some robustness to the posterior, down–weighting observations if $\beta < 1$ and up weighting these for $\beta > 1$. Through a known result (Knoblauch et al., 2022) which we highlight in Lemma B.1 in the Appendix, this is equivalent to having a weighted Kullback–Leibler divergence, $\frac{1}{\beta}\text{KL}$. This also allows us to define if we want to trust the prior more $\beta < 1$ or less $\beta > 1$, since up weighting the data means down weighting the prior and vice versa.

### C.2. A Note on Definition 4.11

The three conditions combined allow us to say whether the client posterior (or simply the posterior in a global, 1 Client, GBI setting) derived from such a robust loss is provably robust to Huber contamination.

From Condition 1 we are able to bound an infinitesimal change in the loss with the contaminating data point $z$ by some auxiliary function $\gamma$, possibly infinite for some values of $\boldsymbol{\theta}$.

Condition 2 states that the product function, $\gamma(\boldsymbol{\theta})\pi(\boldsymbol{\theta})$ has finite uniform norm. This ensures that this product under the worst case contamination and the worst parameter $\boldsymbol{\theta}$, is finite and hence it cannot be made arbitrarily bad, which does not hold for the negative log likelihood in general. Alternatively, the prior decays to zero faster than the auxiliary function can diverge to infinity in $\boldsymbol{\theta}$.

Condition 3 further says that $\gamma(\boldsymbol{\theta})\pi(\boldsymbol{\theta})$ is finitely integrable, i.e. that this is in $L^1(\Theta, \mu)$. This, in effect bounds the normalising constant of the contaminated posterior and will ensure that this is finite.

Taking all these conditions together tells us that the product function $\pi(\boldsymbol{\theta})\gamma(\boldsymbol{\theta})$ is in $L^1(\Theta, \mu) \cap L^\infty(\Theta, \mu)$, and that it is in fact finite everywhere. These two conditions that are mutually independent so both Condition 2 and 3 need to hold. Equivalently, we require that $\gamma(\boldsymbol{\theta})$ is bounded and integrable with respect to the prior probability measure $\pi(\boldsymbol{\theta})\mu(d\boldsymbol{\theta}) =: \Pi(d\boldsymbol{\theta})$.

These conditions characterise the notion of robustness we use for Theorem 4.12, with derivation in Appendix B.7, by considering the worst choice for the contamination $z$ and the parameter $\boldsymbol{\theta}$ with respect to small perturbations of the resulting posterior through $\varepsilon$. The influence of the contamination $z$ and parameter $\boldsymbol{\theta}$ on the posterior is defined as $\frac{d}{d\varepsilon} q_m^{(t)}(\boldsymbol{\theta}; \mathbb{P}_{n_m, \varepsilon, z})|_{\varepsilon=0}$, which is bounded through the conditions. Our result then implies that the posterior is '*globally bias robust*', i.e. robust to Huber contamination.

### C.3. FEDGVI in the Context of GVI and FL

When viewing GVI/GBI as an optimisation problem on the space of probability distributions $\mathcal{P}(\Theta)$, Bayesian inference, VI, hierarchical Bayes/VI, all target a single element of this space. These methods either target the standard Bayesian posterior

explicitly, or the posterior within some variational family with closest Kullback–Leibler distance to the Bayesian one (Blei et al., 2017; Walker, 2013). Through GBI and GVI we are able to target different elements of a subspace of $\mathcal{P}(\Theta)$, then simply a single point; in that regard, these approaches 'generalise' Bayes. In this paper, 'generalised' is inherited from GVI and GBI. We should note that in the FEDGVI setting, GBI and GVI allow us to generalise PVI or FEDAVG to a broader subspace of possible posteriors. Figure 8 displays FEDGVI in regards to the related GVI and GBI literature as well as the FL literature as in Figure 1.

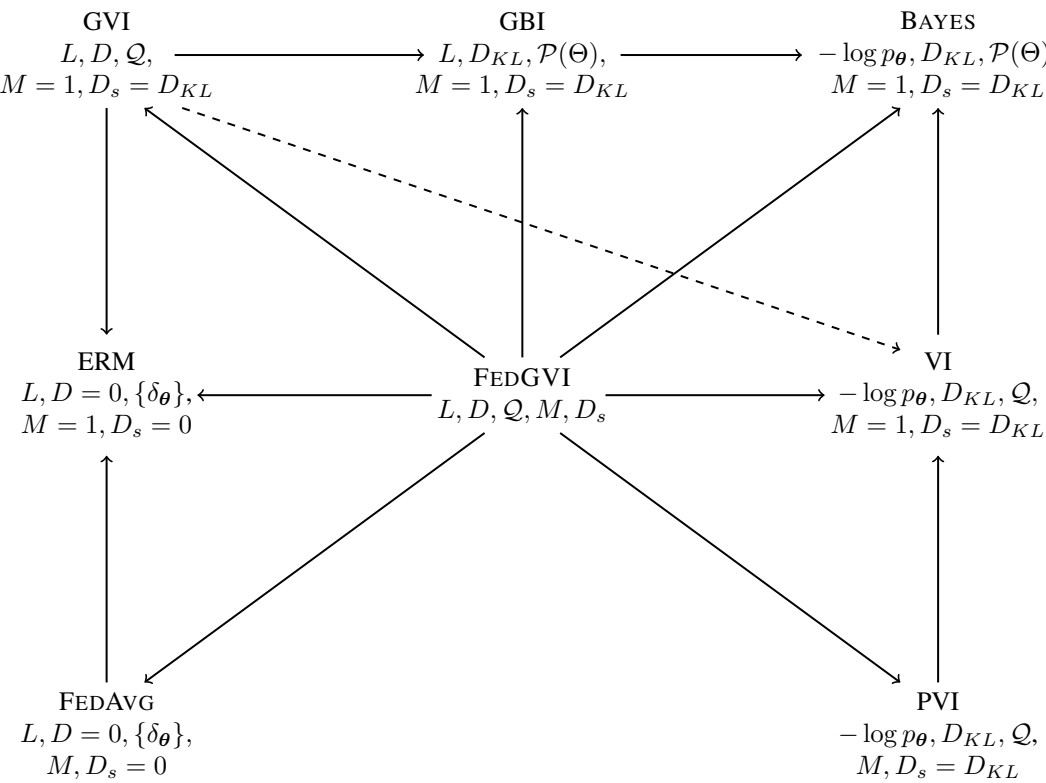

Figure 8: We illustrate the relationship of FEDGVI—characterised by the loss $L$, the client divergence $D$, the variational family $\mathcal{Q}$, the number of clients $M$, and the divergence at the server $D_s$—to Generalised Variational/Bayesian Inference (GVI/GBI), Partitioned Variational Inference (PVI), Variational Inference (VI), Federated Averaging (FEDAVG), Empirical Risk Minimisation (ERM), and Bayes.

## D. Additional Details on Experiments

For reproducibility we give additional details on the experiments that we have carried out to empirically support our contributions. Code to reproduce these can be found at:

https://github.com/Terje-M/FedGVI.

### D.1. Normal–Location Model

We assume the following well known model for the Data Generating Process and prior, with some unspecified prior mean $\mu_\pi$, in order to allow for prior misspecification:

$$\boldsymbol{\theta} \sim \mathcal{N}(\mu_\pi, 1^2) := \pi(\boldsymbol{\theta})$$

$$x_{1:N}|\boldsymbol{\theta} \overset{\text{iid}}{\sim} \mathcal{N}(\boldsymbol{\theta}, 1^2) := p(x_i|\boldsymbol{\theta}).$$

The true Data Generating Process under model misspecification through Huber contamination is given by:

$$\boldsymbol{\theta} \sim \mathcal{N}(0, 0.5^2)$$

$$x_{1:N} | \boldsymbol{\theta} \stackrel{\text{iid}}{\sim} (1 - \varepsilon) \mathcal{N}((\boldsymbol{\theta} - 2), 1^2) + \varepsilon \mathcal{N}((\boldsymbol{\theta} + 3), 0.5^2)$$

where the second term represents some $\varepsilon$ noise fraction that is added to the data. Our aim is to find the location of the first term in the above model, while modelling out the noise from the second term.

We consider PVI where the client optimisation is given by $P_m(-\log p(\cdot | \boldsymbol{\theta}), KL, \mathcal{N})$, and the server optimisation step by $P_s(\ell_s^{(\cdot)}(\boldsymbol{\theta}), KL, \mathcal{N})$. Under the assumption of likelihood misspecification, we consider the following divergences and losses at the clients, while leaving the server optimisation step unchanged: The weighted Kullback–Leibler divergence $\frac{1}{w} D_{KL}$, the Alpha–Rényi divergence $D_{AR}^{(\alpha)}$, the Fisher–Rao divergence $D_{FR}$, the score matching losses $\mathcal{L}_{SM}^{(w)}$, the beta–divergence based loss $\mathcal{L}_B^{(\beta)}$, and the gamma–divergence based loss $\mathcal{L}_G^{(\gamma)}$. Expect for $P_m(\mathcal{L}_{SM}^{(w)}, \frac{1}{w} D_{KL}, \mathcal{N})$, which allows for conjugate updates by Proposition 4.10, we have to resort to optimisation. This however does not require Monte–Carlo sampling since the divergence terms and the losses have closed forms under Gaussian distributions, see Knoblauch et al. (2022) for the remaining losses, Pardo Llorente (2006) for the KL and Alpha–Rényi divergences and Nielsen (2023) for the Fisher–Rao divergence. For the optimisation, we use the Adam optimiser with a learning rate of $0.001$, leaving all other parameters at their default values.

**Explicit Losses and Divergences used**   As mentioned in Section 3.1, we employ a range of different loss functions and divergences throughout the experiments. The main one being the robust generalised cross entropy used in the real world experiments. For the synthetics, for instance in Figure 3 we compare four different losses with two different implementations for the Score–Matching loss.

For this example, where we only have one sequence of data points $x_{1:N}$ which are assumed to be independent, the losses are:

1. The Negative Log Likelihood:
$$\mathcal{L}_{NLL}(x_i, p_{\boldsymbol{\theta}}) = -\log p_{\boldsymbol{\theta}}(x_i)$$

2. The Density–Power Divergence based loss (Ghosh & Basu, 2016a;b):
$$\mathcal{L}_B^{(\beta)}(x_i, p_{\boldsymbol{\theta}}) = -\frac{1}{\beta} p_{\boldsymbol{\theta}}(x_i)^\beta + \frac{1}{1 + \beta} \int_{\Xi} p_{\boldsymbol{\theta}}(x)^{\beta+1} \mu(dx)$$

3. The Gamma divergence based loss (Hung et al., 2018):
$$\mathcal{L}_G^{(\gamma)}(x_i, p_{\boldsymbol{\theta}}) = -\frac{1}{(\gamma - 1)} p_{\boldsymbol{\theta}}(x_i)^{\gamma-1} \cdot \frac{\gamma}{\left( \int_{\Xi} p_{\boldsymbol{\theta}}(x)^\gamma \mu(dx) \right)^{\frac{\gamma-1}{\gamma}}}$$

4. The weighted Score Matching Loss (Altamirano et al., 2023):
$$\mathcal{L}_{SM}^{(w_m^{(t)})}(x_i, p_{\boldsymbol{\theta}}) = ||w_m^{(t)}(x_i)^\top \nabla_x \log p_{\boldsymbol{\theta}}(x_i)||_2^2 + 2\nabla \cdot (w_m^{(t)}(x_i) w_m^{(t)}(x_i)^\top \nabla_x \log p_{\boldsymbol{\theta}}(x_i))$$

We use two different weight functions $w_m^{(t)}$, where $\mu_{\setminus m}^{(t)}$ is the mean of the cavity distribution:

(a) The Squared Exponential Kernel (SE):
$$w_m^{(t)}(x_i) = \beta \exp \left\{ -\frac{(x_i - \mu_{\setminus m}^{(t)})^2}{2c^2} \right\}$$

(b) The Inverse Multi-Quadratic Kernel (IMQ):
$$w_m^{(t)}(x_i) = \beta \left( 1 + \frac{(x_i - \mu_{\setminus m}^{(t)})^2}{2ac^2} \right)^{-a}$$

See Appendix B.6 for details on what this posterior looks like using this particular loss.

All the above losses have closed form objectives under the expectation with respect to the approximating distribution $q(\boldsymbol{\theta})$ and the assumed Gaussian likelihood. Furthermore, the negative log likelihood and the score matching loss admit conjugate updates under the KL divergence.

We also use different divergences, mainly:

1. The Kullback–Leibler divergence (Kullback & Leibler, 1951):

$$D_{KL}(q : \pi) = \int_{\Theta} q(\boldsymbol{\theta}) \log \frac{q(\boldsymbol{\theta})}{\pi(\boldsymbol{\theta})} \mu(d\boldsymbol{\theta})$$

2. The Reverse KL divergence:

$$D_{RKL}(q : \pi) = D_{KL}(\pi : q)$$

3. The Alpha–Rényi divergence:

$$D_{AR}^{(\alpha)}(q : \pi) = \frac{1}{\alpha(1 - \alpha)} \log \int_{\Theta} q(\boldsymbol{\theta})^{\alpha} \pi(\boldsymbol{\theta})^{1-\alpha} \mu(d\boldsymbol{\theta})$$

4. Weighted Divergences of the form:

$$\frac{1}{\beta} D(q : \pi)$$

For Gaussian distributions, these have closed form solutions.

### D.1.1. INFLUENCE FUNCTIONS

For the influence functions experiment in Figure 3, we still assume the same likelihood function, but we have a different data generating process. We generate 99 data points from the following student-t distribution with 4 degrees of freedom, mean 0 and scale 1,:

$$x_{1:99} \sim Student\, T(0, 1, 4)$$
$$x_{100} \sim \delta_y(x),\, y \in \mathbb{R}$$

We place Huber contamination on the hypothesis, where we add an additional observation to one of seven clients that is increasingly farther from the true mean, and calculate the posteriors with this outlier, $y$. We have used the losses described previously for the posteriors and the Kullback–Leibler divergence, running all experiments to convergence. We compare the resulting distributions using the Fisher–Rao divergence (Nielsen, 2023), which has closed form between two univariate Gaussians $q(\boldsymbol{\theta}) \sim \mathcal{N}(\mu_q, \sigma_q^2)$ and $\pi(\boldsymbol{\theta}) \sim \mathcal{N}(\mu_\pi, \sigma_\pi^2)$

$$D_{FR}(q : \pi) = \sqrt{2} \log \left( \frac{1 + \Delta(\mu_q, \sigma_q : \mu_\pi, \sigma_\pi)}{1 - \Delta(\mu_q, \sigma_q : \mu_\pi, \sigma_\pi)} \right)$$

$$\Delta(a, b : c, d) := \sqrt{\frac{(c - a)^2 + (d - b)^2}{(c - a)^2 - (d + b)^2}}, \quad (a, b, c, d) \in \mathbb{R}^4 \backslash \{0\}$$

### D.2. Logistic Regression with Gaussian Design

We place a mean field Gaussian distribution over the parameters of linear model $\boldsymbol{\theta}^{\top} \mathbf{x} + b$ by augmenting the data to $\tilde{\mathbf{x}} = [1, \mathbf{x}^{\top}]$ in order to allow for non–normalised data sets. We assume that the labels, $\mathbf{y}_i \in \{0, 1\}$, follow a Bernoulli distribution with sigmoid probabilities:

$$\mathbf{y}_i \sim \text{Ber}(\sigma(\boldsymbol{\theta}^{\top} \tilde{\mathbf{x}}_i))$$

where $\sigma(a) = (1 + e^{-a})^{-1}$ is the sigmoid function. This allows us to define the likelihood as follows:

$$p(\mathbf{y}_i | \boldsymbol{\theta}, \mathbf{x}_i) = \exp\{\mathbf{y}_i \tilde{\mathbf{x}}_i^{\top} \boldsymbol{\theta} - \psi(\tilde{\mathbf{x}}_i^{\top} \boldsymbol{\theta})\}$$

where $\psi(a) := \log(1 + e^a)$, which gives rise to the sigmoid through $\sigma(a) = \psi'(a)$ (Katsevich & Rigollet, 2024). We use this exponential family form above since taking the logarithm for the negative log–likelihood is easily achieved by removing the exponential and allows for slightly faster calculations during the optimisation. Further, we assume that the prior $\pi(\boldsymbol{\theta}) = \mathcal{N}(\mathbf{0}, \Sigma)$, where $\Sigma$ is fixed but generated through sampling from a Gamma distribution and averaging over the samples. More specifically we sampled 100 samples from a Gamma distribution with $\xi_{1:100} \overset{\text{iid}}{\sim} \text{Gamma}(1, 1/0.01)$, and use their mean, $\bar{\xi}$, to define $\Sigma := \bar{\xi}^{-1}\mathbf{I}_d$. This was done to ensure fairness with the Distributed Stein Variational Gradient Descent approach of Kassab & Simeone (2022), who use an Gaussian inverse Gamma prior, which we for ease of implementation forgo (the results of the experiments show that we easily match their performance, if not surpass it slightly). For the prediction, we use an approximation to the expectation with respect to the final distribution found, $q_s^{(T)}(\boldsymbol{\theta}) \sim \mathcal{N}(\mu_s, \Sigma_s)$ where $\Sigma_s$ is a diagonal matrix, as in Ashman et al. (2022).

$$p(\mathbf{y}_{\text{new}} = 1|\tilde{\mathbf{x}}_{\text{new}}) = \mathbb{E}_{q_s^{(T)}(\boldsymbol{\theta})}\left[p(\mathbf{y}_{\text{new}} = 1|\boldsymbol{\theta}, \tilde{\mathbf{x}}_{\text{new}})\right] \approx \sigma\left(\frac{\mu_s^\top \tilde{\mathbf{x}}_{\text{new}}}{\sqrt{1 + \pi \tilde{\mathbf{x}}_{\text{new}}^\top \Sigma_s \tilde{\mathbf{x}}_{\text{new}}}}\right)$$

This allows us to forgo Monte Carlo sampling to evaluate this expectation.

*Remark* D.1. Since neither GVI, nor FEDGVI targets the Bayesian posterior under different divergences or loss functions in comparison to vanilla VI, we cannot truly speak of this expectation approximating the Bayesian posterior predictive distribution, however since our aim is to find a distribution that is more valuable to a decision maker, using a FEDGVI posterior should allow us to make more informed predictions depending on what the DM wants to model. This can be better uncertainty quantification through changing the divergence, and/or better prediction accuracy through changing the loss.

### D.2.1. FURTHER EXPERIMENTS

In Figure 9 we compare the predictive performance of FEDGVI with two clients against that of GVI with only one client.

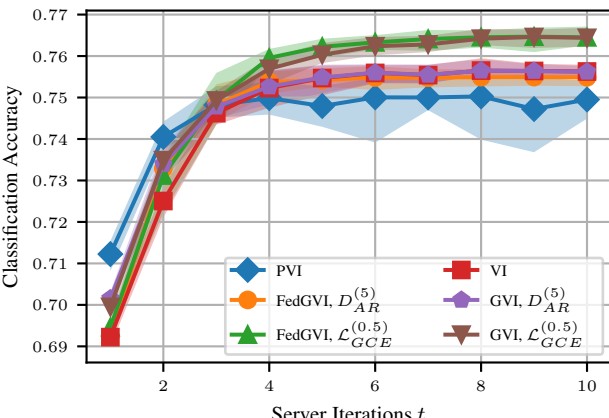

Figure 9: Comparing Logistic Regression with FEDGVI where the data set is split across clients, to GVI where the entire data set is available.

### D.3. Bayesian Neural Networks

The model architecture is a fully connected multi–layer perceptron with RELU activation.

### D.3.1. MNIST (LECUN ET AL., 1998) DETAILS AND ADDITIONAL EXPERIMENTS

For the hyperparameters of the competing methods in the BNNs, we follow Hasan et al. (2024) in using SGD with momentum with a learning rate of 0.1 for FEDAVG, and $\beta$–PREDBAYES, and 0.01 for FEDPA. The architecture for these is a 2 hidden layer fully connected neural network, where each hidden layer has 100 neurons.

For FEDGVI and PVI, we follow the set up of (Ashman et al., 2022) in using the ADAM optimiser (Kingma & Ba, 2015) with a learning rate of 0.0005, leaving all other parameters the default values in `PyTorch`. Here we use a fully connected NN with 1 hidden layer of 200 neurons.

The contamination maps all contaminated data points of one class to a single other class. In both cases, we carried out mini–batch optimisation.

Since we use different architectures for the BNN experiments for the MNIST data set in Table 1, we additionally report results for BNNs when we use the same Neural Network architecture and still retain superior performance of FEDGVI, see Table 3. We notice that the choosing the implementation with the two hidden layer NN for the competing methods performs better or on an equivalent level (within one standard deviation) of each other, while FEDGVI performs better on the single layer NN.

When examining their convergence behaviour under the different architectures, we further notice that the competing methods perform worse than FEDGVI, and FEDAVG and FEDPA exhibit no stability in their accuracy in the contaminated setting, see Figures 10 and 11. This phenomenon occurs even in the uncontaminated case as reported in Al-Shedivat et al. (2021), where we have chosen the optimiser and learning rates as suggested in their paper, and hence we conjecture that contamination further exacerbates this.

Table 3: Classification accuracy (highest in bold) on uncontaminated test data after training on 10% contaminated MNIST data. Here, we compare the results with a fully connected Neural Network with 1 hidden layer of 200 Neurons to the results of a fully connected Neural Network with 2 hidden layers of 100 Neurons each. We report the best performance across all server iterations.

| MODEL | 1 HIDDEN LAYER | | 2 HIDDEN LAYERS | |
| --- | --- | --- | --- | --- |
| | 10 CLIENTS | 3 CLIENTS | 10 CLIENTS | 3 CLIENTS |
| FEDAVG | 94.79±0.43 | 91.76±0.08 | 96.64± 0.07 | 96.34 ± 0.20 |
| FEDPA | 94.53±0.15 | 95.74±0.08 | 94.25± 0.39 | 95.31± 0.35 |
| $\beta$–PREDBAYES | 94.96±0.06 | 96.67±0.07 | 94.90± 0.08 | 96.73± 0.08 |
| PVI | 95.56± 0.18 | 96.68± 0.07 | 95.68±0.10 | 97.31±0.08 |
| FEDGVI $D_{AR}$ | 96.36± 0.09 | 97.13 ± 0.13 | 95.78±0.17 | 97.24±0.05 |
| FEDGVI $L_{GCE}$ | 97.06± 0.03 | 98.04 ± 0.07 | 96.57±0.04 | **97.74±0.11** |
| FEDGVI $D_{AR}$+$L_{GCE}$ | **97.50± 0.07** | **98.13± 0.08** | **96.77±0.10** | **97.79±0.10** |
| VI (1 CLIENT) | (96.96± 0.17) | | (90.87±0.50) | |
| GVI (1 CLIENT) | (**98.13± 0.07**) | | (**97.56±0.05**) | |

Table 4: Classification accuracy (highest in bold for each learning rate) on uncontaminated test data after training on 10% contaminated MNIST data split across 3 clients. Here, we compare the results of different initialisations of FEDGVI with the Alpha–Rényi divergence and generalised cross entropy loss achieved when optimising the posteriors with different learning rates of ADAM. We report the best performance after all server iterations.

| MODEL | LEARNING RATE $\eta$ | | | | | |
| --- | --- | --- | --- | --- | --- | --- |
| | $1e-2$ | $5e-3$ | $1e-3$ | $5e-4$ | $1e-4$ | $5e-3$ |
| PVI | 96.34±0.16 | 96.50±0.18 | 96.72±0.06 | 96.76±0.07 | 96.01±0.05 | 95.39±0.06 |
| FEDGVI $D_{AR}^{(2.5)}$ | 96.84±0.12 | 96.91±0.02 | 97.16±0.04 | 97.18±0.03 | 96.51±0.19 | 95.65±0.03 |
| FEDGVI $\mathcal{L}_{GCE}^{(0.8)}$ | 98.22±0.07 | **98.30±0.03** | 98.15±0.01 | **98.08±0.08** | 97.07±0.06 | 95.84±0.04 |
| FEDGVI $D_{AR}^{(2.5)} + \mathcal{L}_{GCE}^{(0.8)}$ | **98.31±0.10** | 98.24±0.07 | **98.23±0.06** | 98.06±0.09 | **97.50±0.01** | **96.35±0.08** |

In Table 4 we compare different learning rates of ADAM with different initialisations of FEDGVI showing that we in fact underperform with the learning rate selected for the experiments in Figures 6 and 7, and Table 1 for the outperforming methods of FEDGVI.

In Table 5 we further investigate the stability of FEDGVI posteriors when slightly varying the robustness parameter. This shows no significant variations in the accuracy achieved by FEDGVI when slightly perturbing $\delta = 0.8$.

We also want to highlight that by not carefully selecting the hyperparameters of FEDGVI, as well as the learning rate, and keeping these constant across the BNN experiments, we have shown that you do not require extensive knowledge to adapt existing PVI approaches to FEDGVI and outperform. For instance, FEDGVI performs even better for the robust losses at a

Table 5: We fix $\alpha = 2.5$ in the Alpha–Rényi divergence, and vary $\delta$, of the generalised cross entropy loss of Zhang & Sabuncu (2018), around $0.8$. We report accuracies on uncontaminated test data after training on 10% contaminated MNIST data split across 5 clients. These accuracies vary very little demonstrating stability in the FEDGVI posterior at slight perturbations in the loss parameter.

| MODEL | $\delta$ OF $\mathcal{L}_{GCE}^{(\delta)}$ | | | | | | |
|---|---|---|---|---|---|---|---|
| | 0.75 | 0.775 | 0.79 | 0.8 | 0.81 | 0.825 | 8.85 |
| FEDGVI $D_{AR}^{(2.5)} + \mathcal{L}_{GCE}^{(\delta)}$ | 98.05±0.02 | 98.14±0.09 | 98.04±0.09 | 98.06±0.06 | 98.06±0.05 | 97.99±0.07 | 97.98±0.03 |

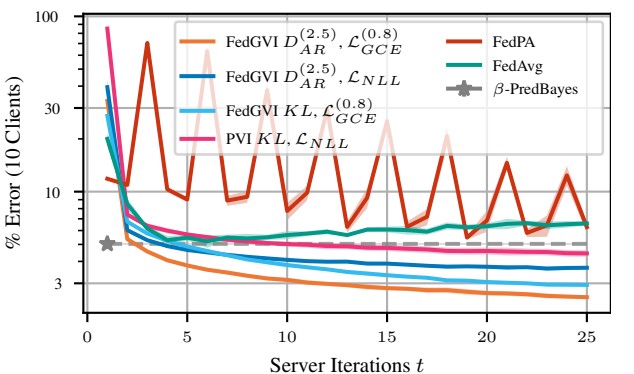
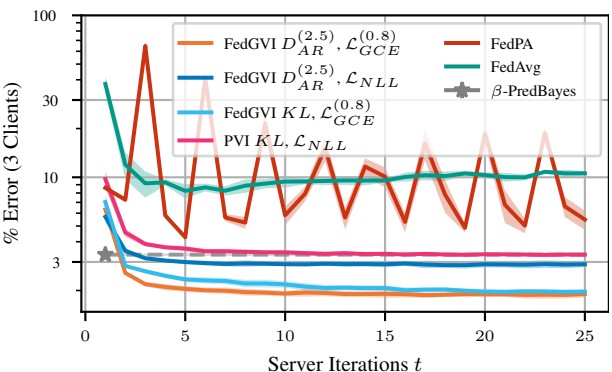

Figure 10: Accuracy on Fully Connected Neural Networks with **1 Hidden Layer**. We demonstrate convergence of the different approaches examined in the first multicolumn of Table 3. The models are trained on 10% label–contaminated data, and prediction accuracy is assessed on uncontaminated test data.

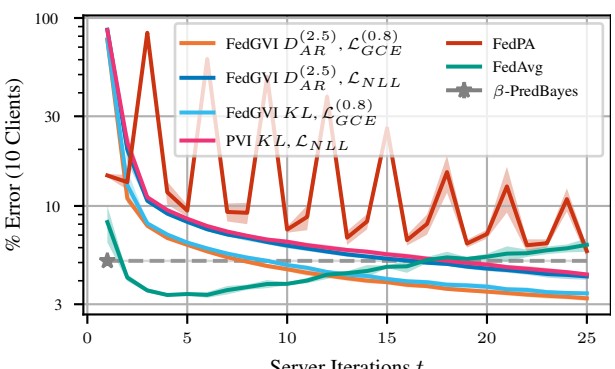
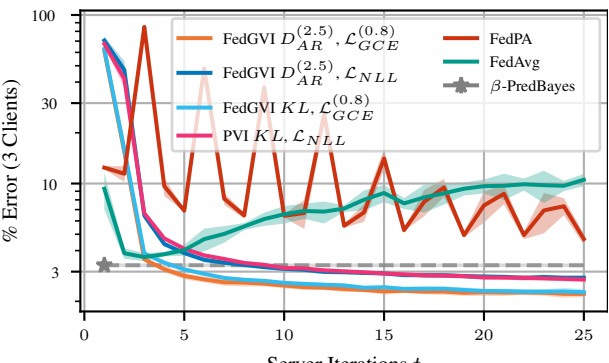

Figure 11: Accuracy on Fully Connected Neural Networks with **2 Hidden Layers**. We demonstrate convergence of the different approaches examined in the second multicolumn of Table 3. The models are trained on 10% label–contaminated data, and prediction accuracy is assessed on uncontaminated test data.

higher learning rate, but we have shown in Table 1 that it still outperforms even when not carefully selecting a learning rate. Furthermore, choosing $\delta = 0.6$ and $\alpha = 2.5$ would have performed better when varying only the robustness parameters of FedGVI, as seen in Figure 6.

Lastly, Figure 12 shows that even when the loss is not available in a conjugate, closed form way, that FEDGVI still incurs no significant computational overhead through choosing the Alpha–Rényi divergence or the generalised cross entropy loss.

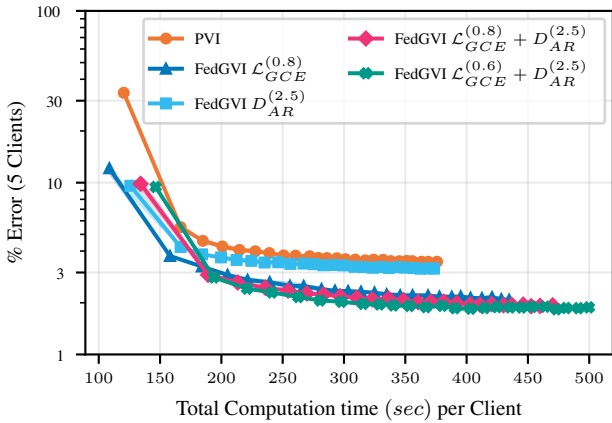

Figure 12: Wall–clock times for FEDGVI iterations per client. We plot the classification error against the computation time taken per client during each server iteration, where we train 5 Clients on 10% contaminated MNIST data. If we do not state the loss or divergence in the legend, it is $\mathcal{L}_{NLL}$ and $D_{KL}$ respectively. Here, FEDGVI outperforms PVI in terms of accuracy while having similar runtimes.

### D.3.2. FASHIONMNIST (XIAO ET AL., 2017) DETAILS

We vary the amount of contamination from 0.0, 0.1, 0.2, 0.4, where the contamination is random and assigns each contaminated data point a different class uniformly at random. The model architecture, prior, learning rate, and optimiser remain unchanged and are as before.

FEDGVI uses the Alpha–Rényi divergence with an alpha value of 2.5 for all, and the robust generalised cross entropy loss, where $\delta = 0.0$ indicates the negative log likelihood. Table 6 specifies Table 2 to a higher precision but the results are identical.

Table 6: Classification accuracy (highest in bold) on uncontaminated test data after training on different amounts of contaminated FASHIONMNIST data. Each Method has data split homogeneously across 3 Clients. We report the best performance during all server iterations for each method.

| MODEL | CONTAMINATION | | | |
|---|---|---|---|---|
| | 0% | 10% | 20% | 40% |
| FEDAVG | 85.72±0.52 | 78.99±1.90 | 71.16±1.53 | 48.97±6.51 |
| FEDPA | 88.08±0.30 | 87.36±0.15 | 86.54±0.16 | 85.36±0.53 |
| $\beta$–PREDBAYES | 87.58±0.13 | 87.20±0.12 | 86.82±0.07 | 85.77±0.10 |
| PVI | 86.21±0.21 | 85.14±0.13 | 84.36±0.12 | 82.81±0.05 |
| FEDGVI $\delta = 0.0$ | 87.12±0.12 | 86.23±0.15 | 85.56±0.11 | 83.78±0.09 |
| FEDGVI $\delta = 0.4$ | 88.73±0.21 | **88.60±0.09** | 87.01±0.37 | 78.14±0.39 |
| FEDGVI $\delta = 0.5$ | **89.02±0.18** | 88.57±0.16 | **88.39±0.21** | 85.06±0.67 |
| FEDGVI $\delta = 0.8$ | 88.59±0.03 | 88.44±0.07 | 87.95±0.04 | **87.21±0.10** |
| FEDGVI $\delta = 1.0$ | 88.09±0.08 | 87.83±0.14 | 87.54±0.15 | 85.97±0.27 |

