# OpenReview forum: "Federated Generalised Variational Inference: A Robust Probabilistic Federated Learning Framework"
_ICML.cc/2025/Conference — ICML 2025 spotlightposter_

### Official Review · Reviewer_4k22 · 2025-03-06

**Overall Recommendation:** 3

**Summary:**

The paper introduces FEDGVI, a novel probabilistic framework for federated learning that is designed for both prior and likelihood misspecification. ## update after rebuttal I will retain my original score.

**Claims And Evidence:**

Good.

**Essential References Not Discussed:**

N/A.

**Experimental Designs Or Analyses:**

Fair.

**Methods And Evaluation Criteria:**

Fair.

**Other Comments Or Suggestions:**

N/A

**Other Strengths And Weaknesses:**

Strengths:
1. The writing of the paper is clear.
2. The author propose solid theoretical analysis and empirical validation.

Weaknesses:
1. There is no ablation study regarding the hyperparameter selection.

**Questions For Authors:**

N/A

**Relation To Broader Scientific Literature:**

Yes.

**Theoretical Claims:**

Good.

---

> ### Author Rebuttal · Authors · 2025-03-31
>
> We thank the reviewer for their appreciation of the clarity of our paper and the theoretical results, as well as their important suggestion for an ablation study that we have now conducted and which will significantly improve the empirical study of this work. We will include these in the revised version of the main paper.
>
> **Hyperparameters of FedGVI**
>
> In particular, we have carried out an ablation study on the selection of the hyperparameters of FedGVI in the training of a Bayesian Neural Network on 10% contaminated MNIST data, split across 5 clients, extending the results discussed in Section 5.5. We vary the $\delta$ parameter of the Generalised Cross Entropy, and the $\alpha$ parameter of the Alpha-Rényi divergence in the client optimisation step (Eq. 4), and record the predictive accuracies on the uncontaminated test data for the FedGVI posteriors found with the respective hyperparameters:
>
> | | $\delta=0.0$ | $\delta=0.2$ | $\delta=0.4$ | $\delta=0.6$ | $\delta=0.8$ | $\delta=1.0$ |
> | :--: | :--: | :--: | :--: | :--: | :--: | :--: |
> | $\alpha=0.0$ | 93.84 | 96.86 | 98.08 | 97.74 | 97.45 | 97.17 |
> | $\alpha=0.5$ | 96.32 | 96.91 | 97.97 | 97.99 | 97.76 | 97.52 |
> | $\alpha=1.0$ | 96.54 | 96.98 | 97.92 | 98.05 | 97.92 | 97.72 |
> | $\alpha=1.5$ | 96.68 | 96.07 | 97.67 | 98.04 | 97.96 | 97.88 |
> | $\alpha=2.5$ | 96.84 | 94.79 | 97.19 | 98.17 | 98.09 | 97.87 |
> | $\alpha=5.0$ | 97.37 | 92.96 | 95.81 | 97.95 | 98.11 | 98.05 |
>
> where we note that $\delta=0$ implies the negative log likelihood since the loss converges to it as $\delta$ tends down to zero (Zhang & Sabuncu, 2018). For the Alpha-Rényi divergence, $\alpha=1$ implies the Kullback-Leibler divergence, and $\alpha=0$ implies the reverse Kullback-Leibler divergence, i.e. $D_{AR}^{(0)}(q:q^{\backslash m})=D_{RKL}(q:q^{\backslash m}):=D_{KL}(q^{\backslash m}:q)$ (Amari, 2016). This means that for $\alpha=1.0$ and $\delta=0.0$, we recover PVI.
>
> We propose to add these results on hyperparameter selection in FedGVI as an annotated heat map to Section 5.5 in the revised version of the paper; the figure can be viewed here: https://anonymous.4open.science/r/Resources-3CEF/ablation_study.png.
>
> In the figure, we present the maximum result achieved across all server iterations and plot the percentage errors on uncontaminated test data for the different hyperparameter combinations. As evident from the table most of the combinations of $\alpha$ and $\delta$ explored show stability of the posterior regarding classification accuracy with different hyperparameters, however some care should be placed in selecting these  since the wrong choice could significantly degrade model performance, e.g. $\alpha=5.0$ and $\delta=0.2$, where we do not sufficiently filter out outliers but place increased weight on the cavity distribution through the high alpha. Nevertheless, the majority of settings outperform PVI, especially for $\delta=0.6$.
>
> **Learning Rate Selection**
>
> Furthermore, we also present additional results on learning rate selection for the ADAM optimiser (Kingma & Ba, 2015) in the 3 Client FedGVI and PVI regimes presented in Tab. 1 of the paper. So far, we have fixed the learning rate to be 5e-4 in the BNN experiments of the paper, which we now vary while keeping the divergence and loss parameters fixed for the respective method. We highlight the best result for each learning rate in bold.
>
> | Learning Rate $\eta$ | 1e-2 | 5e-3 | 1e-3 | 5e-4 | 1e-4 | 5e-5 |
> | :--: | :--: | :--: | :--: | :--: | :--: | :--: |
> | PVI | 96.34$\pm$0.16 | 96.50$\pm$0.18 | 96.72$\pm$0.06 | 96.76$\pm$0.07 | 96.01$\pm$0.05 | 95.39$\pm$0.06  |
> | FedGVI $D_{AR}^{(2.5)}$ | 96.84$\pm$0.12 | 96.91$\pm$0.02 | 97.16$\pm$0.04 | 97.18$\pm$0.03 | 96.51$\pm$0.19 | 95.65$\pm$0.03  |
> | FedGVI $\mathcal{L}_{GCE}^{(0.8)}$ | 98.22$\pm$0.07 | **98.30$\pm$0.03** | 98.15$\pm$0.01 | **98.08$\pm$0.08** | 97.07$\pm$0.06 | 95.84$\pm$0.04 |
> | FedGVI $D_{AR}^{(2.5)}$ + $\mathcal{L}_{GCE}^{(0.8)}$ | **98.31$\pm$0.10** | 98.24$\pm$0.07 | **98.23$\pm$0.06** | 98.06$\pm$0.09 | **97.50$\pm$0.01** | **96.35$\pm$0.08** |
>
> Here, FedGVI with a robust loss outperforms PVI in every scenario.
>
> We also want to point out that by not carefully selecting the hyperparameters of FedGVI, as well as the learning rate, and keeping these constant across the BNN experiments, we have shown that you don't require extensive knowledge to adapt existing PVI approaches to FedGVI and outperform. For instance, FedGVI performs even better for the robust losses with a higher learning rate, but we have shown in Tab. 1 in the paper that it outperforms even when not carefully selecting a learning rate.
> Furthermore, choosing $\delta=0.6$ and $\alpha=2.5$ would have performed better when varying only the robustness parameters of FedGVI.
>
> **References not in paper**
>
> Kingma, D. P. and Ba, J. Adam: A method for stochastic optimization. In 3rd International Conference on Learning Representations, 2015.

---

### Official Review · Reviewer_RwrC · 2025-03-09

**Overall Recommendation:** 5

**Summary:**

The paper introduces *FEDGVI* (Federated Generalized Variational Inference), a probabilistic federated learning framework designed to be robust against both prior and likelihood misspecification. FEDGVI generalizes Partitioned Variational Inference by integrating robust and conjugate updates, thus reducing computational complexity at client devices. It provides theoretical results demonstrating fixed-point convergence, optimality of the cavity distribution, and robustness to model specification. Empirical evaluations on synthetic and real-world datasets highlight improved robustness and predictive accuracy compared to existing FL methods.

**Claims And Evidence:**

The claims made in the paper regarding robustness and improved predictive performance are generally supported by the theoretical analysis and experimental results. However, the examples provided are standard in statistical and ML literature and are specifically chosen to highlight FEDGVI's strengths over PVI, which clearly struggles in some of these settings. The claims related to applicability in more general settings may require further validation.

Indeed, federated hierarchical Bayesian models (which I provide references to below) would likely outperform FEDGVI in many of these settings. While hierarchical models may be out of scope for this work, methods designed for "personalised" federated settings are addressing similar issues. Such methods would be more convincing baselines, and they should at least be mentioned and referenced in the related work. Future work combining generalized Bayesian methods and hierarchical models in federated learning would be very interesting.

**Essential References Not Discussed:**

As mentioned previously I think the following references should at least be mentioned as hierarchical/personalized approaches to Bayesian federated inference and offer an alternative or additional approach to dealing with some of the issues that might arise in the numerical examples provided in the paper:
- Kotelevskii, Nikita, et al. "Fedpop: A bayesian approach for personalised federated learning." Advances in Neural Information Processing Systems 35 (2022): 8687-8701 (was cited, but just in a citation dump of various FL methods)
- Kim, Minyoung, and Timothy Hospedales. "Fedhb: Hierarchical bayesian federated learning." arXiv preprint arXiv:2305.04979 (2023).
- Hassan, Conor, Robert Salomone, and Kerrie Mengersen. "Federated variational inference methods for structured latent variable models." arXiv preprint arXiv:2302.03314 (2023).
- Zhang, Xu, et al. "Personalized federated learning via variational bayesian inference." International Conference on Machine Learning. PMLR, 2022.

**Experimental Designs Or Analyses:**

Experimental designs involving the clutter problem, logistic regression, MNIST, and FASHIONMNIST datasets are sound overall, though somewhat limited in terms of diversity and complexity. Experiments were explicitly crafted to demonstrate FEDGVI’s advantages, which may slightly exaggerate the practical benefits.

**Methods And Evaluation Criteria:**

The proposed methods and evaluation criteria, including theoretical robustness under various divergence-based losses and experimental evaluation on common benchmark datasets, are reasonable.

**Other Comments Or Suggestions:**

The paper would benefit from additional complex real-world experiments beyond simple label contamination scenarios, such as realistic non-IID client data distributions, large-scale deployments, or federated learning tasks with genuine privacy or computational constraints.

**Other Strengths And Weaknesses:**

**Strengths** of the paper include the theoretical analysis, clear positioning within existing literature, and the presentation of explicit theoretical gurantees of robustness and convergence. Additionally, the paper offers a systematic integration of GBI with federated learning, which can be valuable for further theoretical explorations. **Weaknesses** of the paper are that the experimental setups are simplistic and somewhat contrived, limiting the empirical evidence of generalization to more challenging, realistic scenarios. The paper's methodology is straightforward (applying generalized Bayesian inference in FL context), potentially limiting originality and novelty in methodological contribution.

**Questions For Authors:**

1. Do the authors see any unique methodological contributions that the use of generalized Bayesian inference could specifically enable in federated learning settings?

2. What future directions do the authors envision for further integrating generalized Bayesian methods with hierarchical or personalized federated learning approaches?

**Relation To Broader Scientific Literature:**

FEDGVI effectively integrates robust Bayesian inference methods (GBI and GVI) with federated learning. It builds upon prior work in Bayesian FL and robust inference, clearly identifying its position and contributions within the broader context. However, the common claim in generalized Bayesian methods literature that each new approach inherently "generalizes" existing Bayesian approaches is somewhat overstated. Generalization in this context is an expected consequence of employing generalized Bayesian inference, rather than a novel contribution specific to each individual application of Generalized Bayes.

**Theoretical Claims:**

The theoretical results, such as the proofs of robustness and convergence of FEDGVI (e.g., Theorems 4.8 and 4.11, Lemma 4.5, Proposition 4.9), are clearly stated, and are **the biggest strength of the paper**. I reviewed proofs provided in the supplementary material, and they appear sound and rigorous. No issues were identified.

---

> ### Author Rebuttal · Authors · 2025-04-01
>
> Thank you for appreciating our theoretical results and especially for taking the time to review our proofs.
>
> **Claims & Evidence**
>
> > the examples provided are [...] specifically chosen to highlight FEDGVI's strengths over PVI
>
> FedGVI with the negative log likelihood and KL divergence is equivalent to PVI, i.e. FedGVI(NLL,KLD)=PVI. In the idealised setting where the DGP $p_0=p_\theta$ is known, PVI with a correctly specified likelihood is preferable to FedGVI(RobustLoss, KLD), i.e. using a robust loss will degrade performance; see Knoblauch et al. (2022), Jewson et al. (2018), Bissiri et al. (2016). Arguably, we rarely have accurate likelihood functions, since these are either too complex to work with or practitioners opt for standard likelihoods without integrating domain expertise.
>
> We will amend Sec. 3.2 and add:
>
> *It is important to highlight that GVI and FedGVI may underperform when using robust losses in the case of correct likelihood specification.*
>
> **Relation to Literature**
>
> When viewing GVI/GBI as an optimisation problem on the space of probability distributions $\mathcal{P}(\Theta)$, Bayesian inference, VI, hierarchical Bayes/VI, all target a single element of this space. These methods either target the standard Bayesian posterior explicitly, or the posterior within some variational family with closest Kullback-Leibler distance to the Bayesian one.
> Through GBI and GVI we are able to target different elements of a  subspace of $\mathcal{P}(\Theta)$, then simply a single point; in that regard, these approaches do 'generalise' Bayes. In the paper, 'generalised' is inherited from GVI and GBI. We should note that in the FedGVI setting, GBI and GVI allow us to generalise PVI or FedAvg to a broader subspace of possible posteriors. We will clarify this naming distinction in the main paper.
>
> We have made a figure to highlight this, see https://anonymous.4open.science/r/Resources-3CEF/FedGVI.png, which we will include in the paper.
>
> **Question 2, Missed References, Hierarchical Bayes**
>
> Following your suggestion we will add and discuss the proposed references in relation to FedGVI and GVI, giving an additional discussion on personalised and hierarchical Bayesian FL in Sec 1.1. Furthermore, we propose to add the following to Sec. 6:
>
> *An interesting future direction is to extend FedGVI within personalised FL settings (Kotelevskii et al., 2020) and hierarchical Bayesian FL through latent variables (Kim & Hospedales, 2023), as well as the use of a structured posterior approximation (Hassan et al., 2023; 2024), in order to incorporate client level variations. Incorporating the hierarchical model structures and additional inductive biases from such settings, while maintaining conjugacy and favourable computational complexity, remain as open challenges.*
>
> **Weakness 1, Comments, Experimental Design**
>
> We have now carried out further empirical and ablation studies, https://anonymous.4open.science/r/Resources-3CEF, see also our responses to reviewers *4k22* and *gKSi*, which we will include in the revised paper. Nevertheless, as we have shown even the 'simplistic' MNIST setting already provides challenges to traditional Bayesian FL making it worth studying.
>
> We also agree with the reviewer that extending FedGVI to more complex data sets and scenarios, such as investigating different aspects of FL for instance privacy constraints, the cross-device setting, and massively distributed scenarios, present intriguing future directions for FedGVI. We thank the reviewer for pointing this out and will be included in Sec. 6.
>
> For 'computational constraints' see our next answer.
>
> **Weakness 2, Question 1**
>
> > unique methodological contributions that the use of [GBI] could specifically enable in [FL]
>
> To the best of our knowledge, we have proposed the first theoretically grounded probabilistic FL framework that deals with model misspecification. Our framework brings the following methodological advances:
>
> FedGVI can be more computationally efficient than PVI through the use of GBI, as we have shown in Prop. 4.9. The conjugacy with exponential family likelihoods through the robust score matching loss enables faster computation than PVI since it does not require sampling at the local clients. See also our response on computational complexity to Reviewer *gKSi*.
>
> Through the guaranteed posterior robustness by Theorem 4.11, our prediction accuracy is less affected by outliers which would be detrimental for instance in medical settings; see e.g. Jonker et al. (2024) for FL in medicine.
>
> The server divergence and the potential for robust aggregation at the servers (Sec. 6) without fundamentally changing our approach allows for unique optimisation on a global level not provided by Bayesian FL approaches.
>
> We thank the reviewer for encouraging us to flesh these points out and we will include these in the revised paper.

---

> > ### Comment · Reviewer_RwrC · 2025-04-01
> >
> > Thanks for all the work done on the rebuttals including the added experiments.
> >
> > > FedGVI with the negative log likelihood and KL divergence is equivalent to PVI, i.e. FedGVI(NLL,KLD)=PVI. In the idealised setting where the DGP
> >  is known, PVI with a correctly specified likelihood is preferable to FedGVI(RobustLoss, KLD), i.e. using a robust loss will degrade performance; see Knoblauch et al. (2022), Jewson et al. (2018), Bissiri et al. (2016). Arguably, we rarely have accurate likelihood functions, since these are either too complex to work with or practitioners opt for standard likelihoods without integrating domain expertise.
> >
> > Appreciate the suggested amendment. I think the point made here (FEDGVI(NLL, KLD) == PVI), though obvious, would be useful to make clearer (e.g., when looking the plots, it would be useful if the reader was drawn to this connection always when looking at the results of PVI).
> >
> > > We have made a figure to highlight this, see https://anonymous.4open.science/r/Resources-3CEF/FedGVI.png, which we will include in the paper.
> >
> > Yea sure I understand all this. This comment was probably a bit too harsh initially - I just think that the *main contribution* is other stuff compared to the *framing* of GBI rather than framing of GBI itself - because you get the generalization properties for free really (i.e., any new application of GVI will generalize VI methods that exist for that purpose). Personally, I don't really like the suggested figure. I think that the bottom two thirds of the figure are nice but that the top row is a bit overkill for the purposes of the paper (just a personal opinion - don't feel strongly).
> >
> > > An interesting future direction is to extend FedGVI within personalised FL settings (Kotelevskii et al., 2020) and hierarchical Bayesian FL through latent variables (Kim & Hospedales, 2023), as well as the use of a structured posterior approximation (Hassan et al., 2023; 2024), in order to incorporate client level variations. Incorporating the hierarchical model structures and additional inductive biases from such settings, while maintaining conjugacy and favourable computational complexity, remain as open challenges.
> >
> > Thanks. I think these are useful comments to include. I think if you were ever in the business of chasing SOTA performance, a lot of these methods would likely outperform of the baselines, and regardless, interesting future work!
> >
> > > experimental results
> >
> > The added experiments look good, and in combo with the experiments in the submission, clearly a useful method.
> >
> > > FedGVI can be more computationally efficient than PVI through the use of GBI, as we have shown in Prop. 4.9. The conjugacy with exponential family likelihoods through the robust score matching loss enables faster computation than PVI since it does not require sampling at the local clients. See also our response on computational complexity to Reviewer gKSi.
> >
> > Yea this was an oversight in my initial review; this is a very nice property.
> >
> > Overall, my feeling is that this submission clearly warrants acceptance at this conference.

---

> > > ### Author Response · Authors · 2025-04-02
> > >
> > > Thank you for your fast reply, the constructive feedback, and your score increase.
> > >
> > > We appreciate the remarks on the figure we provided, and will amend it for the paper to the bottom two thirds in order to keep the focus on Federated Learning.
> > > Thanks also for the point on highlighting PVI=FedGVI(NLL, KLD), we will amend the figures/tables accordingly, and expand on this in Sec. 4.1.

---

### Official Review · Reviewer_gKSi · 2025-03-14

**Overall Recommendation:** 4

**Summary:**

The paper presents a new framework, Federated Generalised Variational Inference (FEDGVI), for robust probabilistic federated learning. The authors argue that standard Bayesian and frequentist federated learning methods are vulnerable to model misspecification (e.g., contaminated data, incorrect priors, or mismatched likelihood models). Building on the theory of Generalised Variational Inference (GVI), they propose a method that can (1) systematically handle likelihood and prior misspecifications, (2) generalize existing frameworks like Partitioned Variational Inference (PVI) and FedAvg, and (3) provide stronger robustness guarantees with calibrated uncertainty estimates. Key contributions include:

- A unifying algorithmic framework for GVI in the federated learning (FL) setting, with theoretical convergence guarantees.
- Proofs that FEDGVI is robust to outliers and model misspecification under suitable choices of loss functions and divergences.
- Empirical evaluations on both synthetic tasks (e.g., the 1D “clutter” problem and logistic regression) and real data (CoverType, MNIST, FashionMNIST), demonstrating improved robustness and accuracy compared to baselines such as PVI, DSGLD, and FedAvg.

**Claims And Evidence:**

1. FEDGVI is robust to both likelihood and prior misspecification. The authors develop theoretical results (Section 4) showing that if each client employs a robust (bounded) loss function, the global posterior remains stable even when some portion of data is contaminated (Theorem 4.11).
2. FEDGVI recovers existing federated learning methods (e.g., PVI, FedAvg) as special cases: In Section 4.1, the authors show that by choosing particular divergences and losses, the method reduces to either standard partitioned variational inference (for Bayesian FL) or FedAvg (for frequentist FL).
3. The approach is scalable to real-world tasks and yields superior predictive performance compared to baselines: Empirical studies on real datasets (Cover Type, MNIST, FashionMNIST) demonstrate that FEDGVI obtains higher classification accuracy under mismatched or noisy training.

Overall, the manuscript provides both proofs of conceptual claims and corresponding empirical corroboration.

**Essential References Not Discussed:**

N/A

**Experimental Designs Or Analyses:**

The authors systematically vary contamination rates (e.g., flipping labels randomly on MNIST) and track how well each method recovers clean performance. They compare with classical federated baselines (FedAvg) and Bayesian baselines (PVI, DSGLD, DSVGD, etc.) across varied contamination levels. The data partition is typically homogeneous or split randomly, though the paper briefly remarks it is straightforward to allow non-i.i.d. partitions.

The authors provide clarity that the number of clients (M) or the chosen “damping” parameters can be adapted.
Some additional exploration of the method’s sensitivity to the choice of robust hyperparameters (e.g., $\beta$, $\delta$, $\gamma$) might help readers see how stable the procedure is under default vs. tuned settings.

**Methods And Evaluation Criteria:**

The proposed method relies on Generalised Variational Inference to handle potential model misspecification. Clients use robust local objectives (like Density–Power or Generalised Cross-Entropy losses) while the server employs a chosen divergence-based penalty (often KL or an $\alpha$-Rényi divergence) on the global posterior. Convergence is examined in terms of fixed points—where the global posterior stops changing under repeated local-global updates.

The proposed evaluation criteria seems solid.

**Other Comments Or Suggestions:**

The authors might add small clarifications on the recommended heuristics for selecting the “robustness parameters” (β, γ, etc.) in real-world use, or mention a rule of thumb.

It might be beneficial to discuss whether robust aggregator strategies at the server (like coordinate-wise trimming or medians) could be combined with robust GVI for even stronger resilience against malicious clients.

**Other Strengths And Weaknesses:**

**Strengths**

1. The framework is conceptually simple to implement: each client modifies its update with a robust local objective and the server does a GVI-based merge.
2. The theoretical story on why robust GVI helps in the federated setting is coherent and well-linked with standard references.
3. Strong empirical demonstration that the method truly mitigates outlier influence.

**Weaknesses**

1. Some of the hyperparameter tuning for the robust losses (like $\beta$ or $\gamma$ for L$\beta$/GCE) is not deeply explored in the main text; it might require extra user expertise.
2. Communication cost or runtime overhead is not directly benchmarked: presumably the overhead is comparable to standard PVI or FedAvg, but actual run-time comparisons might be valuable to demonstrate practical feasibility.

**Questions For Authors:**

1. How sensitive is FEDGVI to the choice of the robust loss hyperparameters (e.g., $\delta$ in the generalised cross-entropy or $\beta$ in the density–power loss)? Would small changes degrade performance significantly?

2. In practical setups with many clients ($M>>1$), do the introduced robust loss functions or divergences incur any significant computational overhead relative to standard negative log-likelihood updates?

**Relation To Broader Scientific Literature:**

This work builds on the recognized problem of model misspecification in Bayesian methods, referencing established works like Bissiri et al. (2016) for generalised Bayes, and the robust M-estimation approaches from Ghosh & Basu (2016) or Knoblauch et al. (2022). Overall, the framework is well-situated within existing lines of research but has a distinct novelty in bridging the gap between robust GVI and federated Bayesian methods.

**Theoretical Claims:**

The authors extend existing PVI theory by showing that if a robust generalised Bayesian update is done at each client and a KL-based GVI update at the server, the global solution is itself robust. Key points:
1. Fixed point analysis (Proposition 4.4): Demonstrates that, if the algorithm converges, the final server distribution is a stationary minimizer of a global GVI functional.
2. Equivalence to GBI (Lemma 4.5): With certain parameter choices (e.g., negative log-likelihood as the local “loss,” or a robust alternative + KL at the server), the final solution coincides with the standard or robust GBI posterior.
3. Cavity distribution necessity (Theorem 4.8): Shows that removing the cavity from the client update would lead to systematically biased or overconfident global updates. This result clarifies why each client must “subtract out” the impact of its own data from the global prior, rather than simply using the existing posterior.

I did not uncover any obvious mistakes in the proofs; they seem consistent with prior results on partitioned variational methods. That said, verifying every detail would require deeper familiarity with some of the advanced robust GVI and measure-theoretic derivations, which I do not fully possess.

---

> ### Author Rebuttal · Authors · 2025-04-01
>
> We thank the reviewer for their in-depth, constructive, and positive review.
>
> **Question 1, Weakness 1, and Experiments**
>
> > ...method’s sensitivity to the choice of robust hyperparameters [...] how stable...
>
> > How sensitive is FEDGVI to [...] hyperparameters [...]? Would small changes degrade performance significantly?
>
> We presented results on FashionMNIST while varying $\delta$ with different contaminations in Fig. 2. We now extend these to further contamination levels (0, 0.1, 0.2, 0.4) and $\delta$ (0, 0.4, 0.5, 0.8, 1),
> as well as FedAvg, FedPA, and $\beta$-PredBayes, to demonstrate hyperparameter sensitivity:
>
> https://anonymous.4open.science/r/Resources-3CEF/FashionMNIST.png
>
> We also now offer an ablation study on MNIST for hyperparameter selection: https://anonymous.4open.science/r/Resources-3CEF/ablation_study.png (see also response to reviewer *4k22*).
>
> To show the stability of FedGVI under small perturbations of $\delta$, we fix $\alpha=2.5$, vary $\delta$ around $0.8$, and report accuracies on uncontaminated test data after training on 10% contaminated MNIST data split across 5 clients:
>
> https://anonymous.4open.science/r/Resources-3CEF/Perturbations.png
>
> **Question 2, Weakness 2** Computational Complexity/Overhead
>
> **W2**
> > Communication cost or runtime overhead is not directly benchmarked...
>
> **Q2**
> > do [...] loss functions [...] incur any significant computational overhead [...] updates?
>
> As the number of clients increases, the comp. time at each client is reduced due to less data per client. By Proposition 4.9 we can choose a robust loss at each client that enables conjugate client updates whereas the negative log likelihood might not have closed form. These include intractable likelihood models (Matsubara et al., 2022) where the normalising constant is intractable but we can tractably solve.
>
> The comp. complexity of the divergence is solely dependent on the choice of variational family, prior, and divergence without dependence on number of clients or data per client. The divergence may be more comp. expensive in the case where no closed form solutions exist, e.g. for the Total Variation Distance. However, for exponential family distributions, many robust divergences with closed form solutions exist (Pardo Llorente, 2006; Knoblauch et al., 2022).
>
> The KL and A-R divergences between Gaussians (Sec. 5.5) are available in closed form, scaling only in the number of parameters. The A-R can be computed in $\mathcal{O}(1)$ times the comp. complexity of the KL, driven by the determinant of the covariances; assuming $\Theta\subset\mathbb{R}^d$, and using a Cholesky decomposition this naively takes $\mathcal{O}(d^3)$ for both; assume these are (block)-diagonal, makes this cheaper.
> The discrete losses in the classification setting are non-differentiable. We cannot apply the conjugate score matching loss, and require sampling from the approximation. We can transform the NLL in $\mathcal{O}(1)$ to the GCE, thereby, as the likelihood is used for the GCE and it's log for the NLL, these are of the same order of magnitude.
>
> We will add in Section 5.5:
>
> *FedGVI incurs no additional computational complexity when compared to PVI. This is due to the KL and Alpha-Rényi divergences having closed form solutions between Multivariate Gaussians with complexity of $\mathcal{O}(1)$ in each other, and because we require $\mathcal{O}(1)$ additional, constant operations to get the GCE from the NLL.*
>
> To empirically validate this, we now report average wall-clock times at each client for FedGVI and PVI, see https://anonymous.4open.science/r/Resources-3CEF/runtime.png.
>
> **Comments**
>
> > Heuristics for selecting robustness parameters
>
> These include Knoblauch et al. (2018) and Yonekura &  Sugasawa (2023) for the Density Power loss, with the latter a theoretically principled Sequential Monte Carlo sampler for selecting $\beta$, or Altamirano et al. (2024) for weighted score matching, through cross validation.
>
> We propose to add this in Sections 3.2 and 3.3:
>
> 3.2: *We can use a Sequential Monte Carlo sampler to estimate the $\beta$ or $\gamma$ hyperparameters in $\mathcal{L}_B$ and $\mathcal{L}_G$ (Yonekura &  Sugasawa, 2023) or use cross validation to select optimal parameters (Altamirano et al., 2024).*
>
> 3.3: *Similarly to the losses, we can perform cross validation to select the $\alpha$ parameter, however as demonstrated in the ablation study (Figure TBD) FedGVI performs favourably under a range of $\alpha$ values.*
>
> > Robust Aggregator strategies at the server
>
> This is an area we are actively working on. We are exploring ways in which the summation in Eq. 6 can be replaced by a robust aggregator, such as Nearest Neighbour Mixing (Allouah et al., 2024) or indeed coordinate-wise median and trimmed mean, to achieve Byzantine robustness. We will discuss this in Sec. 6.
>
> **References**
>
> Yonekura, S., Sugasawa, S. Adaptation of the tuning parameter in general Bayesian inference with robust divergence. Stat Comput 33, 39, 2023.

---

### Official Review · Reviewer_KG8f · 2025-03-18

**Overall Recommendation:** 4

**Summary:**

This work presents FedGVI, an extension of partitioned variational inference (PVI) to generalised variational inference (GVI). The core benefit of GVI is that it permits robustness to model misspecification. The authors demonstrate a number of advantageous properties of FedGVI, most notably theoretical results that show it convergeses to GVI approximate posteriors, and is thus robust to model misspecification.

**Claims And Evidence:**

Yes, both theoretical and empirical.

**Essential References Not Discussed:**

No,

**Experimental Designs Or Analyses:**

Yes.

**Methods And Evaluation Criteria:**

Yes.

**Other Comments Or Suggestions:**

N/A.

**Other Strengths And Weaknesses:**

Strengths

* The paper is very well written, not only in presenting FedGVI but also in the presentation of PVI (and extending the current understanding of how it works).
* FedGVI itself is easy to implement (a relatively straightforard extension of PVI).
* The theoretical results are supported by a number of experimantal results demonstrating the superiority of FedGVI relative to baseline methods (including PVI) when data contains outliers.

Weaknesses
* In equation 2 $\beta$ is undefined. Also---and I could be mistaken---should the normalisation constant include a $\beta$ in the integral?
* Whilst I understand the intuition behind 1. in Definition 4.10, I struggle more with the intuition behind points 2. and 3.. If there exists relatively compact intuition, perhaps the paper would benefit from its inclusion here.

**Questions For Authors:**

N/A.

**Relation To Broader Scientific Literature:**

Related to the field of federated probabilistic learning, which has wide spread applications.

**Theoretical Claims:**

No.

---

> ### Author Rebuttal · Authors · 2025-03-31
>
> Thank you for your constructive and positive review that highlights the clarity of our writing, the theoretical results on FedGVI, and its provable robustness. By addressing below the weaknesses you mentioned, we notably improve the clarity of the paper and its ease of understanding for the reader.
>
> **Weakness 1** $\beta$ parameter in Eq. 2 undefined:
>
> Thank you for spotting this typo, it has now been corrected, and we will add the following clarification:
>
> *Here, $\beta\in\mathbb{R}_{>0}$ is a learning rate parameter that determines how much weight we place on the observed data, similar to power posteriors in VI (Grünwald, 2012; Kallioinen et al., 2024).*
>
> The $\beta$ parameter comes from the power/cold/tempered posteriors of e.g. Grünwald (2012), where the likelihood in Bayesian posteriors is raised to some power of $\beta>0$. This was originally done to add some robustness to the posterior, down-weighting observations if $\beta< 1$ and up weighting these for $\beta > 1$. Through a known result (Knoblauch et al., 2022), which we highlight in Lemma B.1 in the Appendix, this is equivalent to having a weighted Kullback-Leibler divergence, $\frac1\beta\mathrm{KL}$. This also allows us to define if we want to trust the prior more $\beta<1$ or less $\beta>1$, since up weighting the data means down weighting the prior and vice versa.
>
> We will add this discussion to Appendix B.1 to give further intuition behind GBI.
>
> **Weakness 2** Intuition behind Definition 4.10 and Conditions 2 and 3:
>
> The three conditions combined allow us to say whether the client posterior (or simply the posterior in a global, 1 Client, GBI setting) derived from such a robust loss is provably robust to Huber contamination.
>
> From *Condition 1* we are able to bound an infinitesimal change in the loss with the contaminating data point $z$ by some auxiliary function $\gamma$, possibly infinite for some values of $\theta$.
>
> *Condition 2* states that the product function, $\gamma(\theta)\pi(\theta)$ has finite uniform norm.
> This ensures that this product under the worst case contamination and the worst parameter $\theta$, is finite and hence it cannot be made arbitrarily bad, which does not hold for the negative log likelihood in general. Alternatively, the prior decays to zero faster than the auxiliary function can diverge to infinity in $\theta$.
>
> *Condition 3* further says that $\gamma(\theta)\pi(\theta)$ is finitely integrable, i.e. that this is in $L^1(\Theta)$. This, in effect, bounds the normalising constant of the contaminated posterior and will ensure that this is finite.
>
> Taking all these conditions together tells us that the product function $\pi(\theta)\gamma(\theta)$ is in $L^1(\Theta)$ and that it is finite everywhere; two conditions that are mutually independent, i.e. one doesn't follow form the other.
>
> These conditions characterise the notion of robustness we use for Theorem 4.11, a derivation of this notion is shown in Appendix B.7, by considering the worst choice for the contamination $z$ and the parameter $\theta$ with respect to small perturbations of the resulting posterior through $\epsilon$.
> The influence of the contamination $z$ and parameter $\theta$ on the posterior is defined as $\frac d{d\epsilon}q_m^{(t)}(\theta;\mathbb{P}_{n_m,\epsilon, z})|_0$ (evaluated at $\epsilon=0$), which is bounded through the conditions. This then implies that the local posterior is *globally bias robust*, i.e. robust to Huber contamination of all $z$ for all parameters $\theta\in\Theta$.
>
>
> Thank you for pointing this out, we will provide the following clarification between Definition 4.10 and Theorem 4.11.
>
> *These conditions ensure that the influence of arbitrary contamination on the local posterior is not arbitrarily bad.
> In particular the auxiliary function $\gamma_m^{(t)}$ ensures that the influence of an adversarial data point $z$ on the posterior over infinitesimal contaminations,
> $$\frac d{d\epsilon} q_m^{(t)}(\theta;\mathbb{P}_{n_m,\epsilon, z}) {\Big|}_0$$
> evaluated at $\epsilon=0$, are finite over all $\theta$ and $z$. Condition 2 ensures the loss increases slowly enough for the local posterior to concentrate around the data, and condition 3 ensures the resulting posterior will be normalisable.*
>
> **References not in Paper**
>
> Kallioinen, N., Paananen, T., Bürkner, P.-C., and Vehtari, A. Detecting and diagnosing prior and likelihood sensitivity with power-scaling. Statistics and Computing, 34(1):57, 2024.

---

### Decision · Program_Chairs · 2025-05-01

**Decision:**

Accept (spotlight poster)

**Comment:**

The submission considers generalised variational inference in the federated settings, extending the partitioned variational inference framework of Ashman et al (2022) to handle potential model misspecification. The theoretical contributions are rigorous and validated by a suite of experiments. All reviewers' questions are comprehensively addressed. I therefore recommend acceptance.